# CLIP BODY AND TAIL SEPARATELY: HIGH PROBABILITY GUARANTEES FOR DPSGD WITH HEAVY TAILS

## ABSTRACT

Differentially Private Stochastic Gradient Descent (DPSGD) is widely utilized to preserve training data privacy in deep learning, which first clips the gradients to a predefined norm and then injects calibrated noise into the training procedure. Existing DPSGD works typically assume the gradients follow sub-Gaussian distributions and design various gradient clipping mechanisms to optimize training performance. However, recent studies have shown that the gradients in deep learning exhibit a heavy-tail phenomenon, that is, the tails of the gradient may have infinite variance, which leads to excessive clipping loss with existing mechanisms. To address this problem, we propose a novel approach, Discriminative Clipping (DC)-DPSGD, with two key designs. First, we introduce a *subspace identification technique* to distinguish between body and tail gradients. Second, we present a *discriminative clipping mechanism* that applies different clipping thresholds for body and tail gradients separately to reduce the clipping loss. Under the non-convex condition and heavy-tailed sub-Weibull gradient noise assumption, DC-DPSGD reduces the empirical risk from $\mathbb{O}\left(\log^{\max(0,\theta-1)}(T/\delta)\log^{2\theta}(\sqrt{T})\right)$ to $\mathbb{O}\left(\log(\sqrt{T})\right)$ with heavy-tailed index $\theta > 1/2$, iterations $T$, and high probability $1-\delta$. Extensive experiments on five real-world datasets demonstrate that our approach outperforms three baselines by up to 9.72% in terms of accuracy.

## 1 INTRODUCTION

DPSGD Abadi et al. (2016), as a mainstream paradigm of privacy-preserving deep learning, has wide applications in areas such as privacy-preserving recommender systems Liu et al. (2023), face recognition Tang et al. (2024), and medical diagnosis Meng et al. (2021); Ji et al. (2022). Essentially, in each iteration of model training, DPSGD clips per-sample gradient under the $L_2$ norm constraint to obtain the maximum divergence between gradient distributions that differ by only one training data and adds random noise within rigorous privacy bounds for unbiased gradient estimation.

Most of existing DPSGD works Bu et al. (2024); Xia et al. (2023); Zhang et al. (2023); Zhu & Blaschko (2023); Koloskova et al. (2023); Li et al. (2022); Fang et al. (2022); Yang et al. (2022) rely on the assumption that the gradient noise follows a sub-Gaussian distribution to devise effective clipping strategies. However, recent studies Zhang et al. (2020b); Simsekli et al. (2019; 2020); Camuto et al. (2021); Barsbey et al. (2021) have shown that SGD gradient noise in deep learning often exhibit heavy-tailed distributions instead of light-tailed distributions (e.g., sub-Gaussian). This occurs even when the dataset originates from a light-tailed distribution, the gradients still diverge to a heavy-tailed distribution with infinite variance Gurbuzbalaban et al. (2021), which may slow down the convergence rate and impair training performance Li & Liu (2022; 2023); Madden et al. (2020); Gorbunov et al. (2020). To cope with this problem in SGD, Li & Liu (2023); Wang et al. (2021); Gorbunov et al. (2020) suggest employing larger clipping thresholds to get rid of the oscillations caused by heavy-tailed gradients on the training trajectory.

Nevertheless, the clipping operation in DPSGD is closely tied to the magnitude of DP noise added to the gradients. Setting the clipping threshold too large can lead to a high-dimensional noise catastrophe Zhou et al. (2021), which negatively impacts model performance and potentially disrupts the convergence of DPSGD algorithms. Therefore, practitioners need to carefully strike a balance between injected noise and clipping loss, as illustrated in Figure 1. The left sub-figure shows the trade-off under the light-tailed assumption. As the clipping threshold increases (i.e., when the red

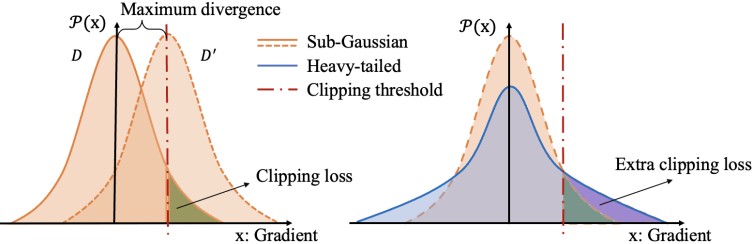

Figure 1: The trade-off between clipping loss and noise magnitude under heavy-tailed distributions.

dotted line moves to the right), the clipping loss decreases, but the maximum divergence between the distributions differing by one clipped gradient increases, leading to more DP noise being added. While in the right sub-figure, under the same noise magnitude, the slower decay rate of the heavy-tailed distribution (blue line) will introduce extra clipping loss. Therefore, we aim to investigate the following key question in this paper: *how to design an effective clipping mechanism under the heavy-tailed assumption to balance the trade-off between clipping loss and DP noise in DPSGD?*

Previous clipping mechanisms for DPSGD Bu et al. (2024); Yang et al. (2022); Xia et al. (2023) have been proposed under the light-tailed assumption, but none of them can be adapted to our problem. Specifically, Bu et al. (2024); Yang et al. (2022); Xia et al. (2023) focus on small-norm gradients (i.e., those near the center of the distribution) and normalize them to be around 1. These approaches reduce the maximum divergence, thereby requiring less noise to be injected. However, they do not account for heavy-tailed gradients and thus cannot optimize the clipping loss. Another line of work directly estimates the actual norm of the per-sample gradient and utilizes it as the clipping threshold to reduce the clipping loss. For instance, Andrew et al. (2021) estimate the true gradient trajectory by collecting the norms of historical gradients. However, this approach requires knowing the upper bound of historical norms for adding noise, which is highly uneconomical under heavy-tailed distributions, as the upper bound for moment generating function (MGF) Vladimirova et al. (2020) can be immeasurable, making the scale of DP noise unbearable and the expectation bounds inapplicable. Moreover, due to the constraints of a finite privacy budget, practical private learning cannot perform indefinite training. Therefore, it is essential to obtain a high probability bound to ensure algorithm performance with the probabilistic nature of privacy noise on single runs.

In this paper, we present high probability bounds with faster convergence rates for DPSGD and propose a novel approach, named **Discriminative Clipping (DC)-DPSGD**, to effectively balance the trade-off between clipping loss and required DP noise under the heavy-tailed assumption. The key idea is to utilize different clipping thresholds for the body gradients and tail gradients respectively, retaining more information from tail gradients that can withstand more severe DP noise. We introduce two techniques to achieve this goal. First, we design a subspace identification technique to identify heavy-tailed gradients with high probability guarantees. We note that the body of heavy-tailed distributions exhibits characteristics similar to those of light-tailed distributions, and the main difference lies in the decay rate at the tails. Therefore, we extract orthogonal random vectors from heavy-tailed distributions (e.g., sub-Weibull distribution) to construct a random projection subspace, and compute the trace of the second moment matrix between gradients and this subspace to distinguish heavy-tailed gradients. Second, we present a discriminative clipping mechanism, which applies a large clipping threshold for the identified heavy-tailed gradients and a smaller one for the remaining light-tailed gradients. We theoretically analyze the choice of the two clipping thresholds and the convergence of DC-DPSGD with a tighter bound. Our contributions are summarized as follows.

- We propose DC-DPSGD with a subspace identification technique and a discriminative clipping mechanism to optimize DPSGD under sub-Weibull gradient noise assumption. To our knowledge, this is the first work to rigorously address heavy tails in DPSGD with high probability guarantees.

- We present a high probability guarantee with best-known rates for the optimization performance of DPSGD, and improve it to faster rates by DC-DPSGD, which shows that the empirical risk is reduced from $\mathbb{O}\left(\log^{\max(0,\theta-1)}(T/\delta)\log^{2\theta}(\sqrt{T})\right)$ to $\mathbb{O}\left(\log(\sqrt{T})\right)$ with heavy-tailed index $\theta > 1/2$, iterations $T$, and high probability $1 - \delta$, under the non-convex condition.

- We conduct extensive experiments on five real-world datasets, where DC-DPSGD consistently outperforms three baselines with up to 9.72% accuracy improvements, demonstrating the effectiveness of our proposed approach.

## 2 RELATED WORK

**Heavy-tailed noise and high probability bounds.** Recently, from the perspective of escaping from stationary points and Langevin dynamics, the noise in neural networks is more inclined to anisotropic and non-Gaussian properties Gurbuzbalaban et al. (2021); Simsekli et al. (2019); Gorbunov et al. (2020); Zhang et al. (2020b), with specific heavy-tailed phenomena discovered and defined in gradient descent in deep neural networks. Several works focused on heavy-tailed convex optimization in privacy-preserving deep learning Lowy & Razaviyayn (2023); Wang et al. (2020); Kamath et al. (2022). Building upon the work of Wang et al. (2020), Kamath et al. (2022) relax the assumption of Lipschitz condition and sub-Exponential distribution to a more general $\alpha$-th moment bounded condition. However, no work has investigated the convergence characteristics of heavy-tailed DPSGD in non-convex settings. Meanwhile, high probability bounds are more frequently discussed in optimization properties such as convex and non-convex learning with SGD, but rarely addressed in the context of private learning. Specifically, with bounded $\alpha$-th moments assumption, Li & Liu (2023) provide a high probability theoretical analysis for variants like clipped SGD with momentum and adaptive step sizes. Nevertheless, these works on optimizing DPSGD rely on expectation bounds, which are unsuitable for heavy-tailed assumptions.

**Projection subspace in DPSGD.** DPSGD has gained wide concerns for its detrimental impact on model accuracy. A series of works leverage projection techniques to improve performance. For instance, Zhou et al. (2021); Yu et al. (2021a;b) confine DPSGD training dynamics to more compact and condensed subspaces through projection. While ensuring the fidelity of training data compression, they decouple the irrelevant relationship between ambient features and DP noise, and reduce the optimization error of DPSGD under stringent privacy constraints. However, existing works rely on the assumption that public datasets are available for designing the techniques Golatkar et al. (2022); Zhou et al. (2021); Yu et al. (2021a); Gu et al. (2023), which is rather strong, especially in sensitive domains. In contrast, our approach does not rely on any public dataset.

**Gradient clipping.** Gradient clipping is a widely adopted technique to ensure the sensitivity of gradients is bounded in both practical implementations and theoretical analysis for DPSGD Chen et al. (2020); Zhang et al. (2020a; 2022); Andrew et al. (2021); Xiao et al. (2023); Wei et al. (2022); Koloskova et al. (2023). Since the tuning parameters in the classical Abadi's clipping function Abadi et al. (2016) are complex, adaptive gradient clipping schemes have been proposed by Bu et al. (2024); Yang et al. (2022). These schemes scale per-sample gradients based on their norms. In particular, gradients with small norms are amplified infinitely. Building upon this, Xia et al. (2023) control the amplification of gradients with small norms in a finite manner. However, no work has specifically optimized gradient clipping under the heavy-tailed assumption of DPSGD. Due to the scale of noise required to achieve differential privacy, trivial clipping methods and analysis are not applicable.

## 3 PRELIMINARIES

### 3.1 NOTATIONS

Let $D$ be a private dataset, which consists of $n$ training data $S = \{z_1, ..., z_n\}$ with a sample domain $Z$ drawn i.i.d. from the underlying distribution $\mathcal{P}$. Since $\mathcal{P}$ is unknown and inaccessible in practice, we minimize the following empirical risk in a differentially private manner:

$$L_S(\mathbf{w}) := \frac{1}{n} \sum_{i=1}^{n} \ell(\mathbf{w}, z_i), \tag{1}$$

where the objective function $\ell(\cdot) : (\mathbf{w} \subseteq W, Z) \to \mathbb{R}$ is possible non-convex and $W \subseteq \mathbb{R}^d$ represents the model parameter space. Then, we denote $\nabla \ell$ as the gradient of $\ell$ with respect to $\mathbf{w}$. Furthermore, we introduce several notations regarding the projection subspace. Let $V_k \in \mathbb{R}^{d \times k}$ denote $k$-dimensional random projection sampled from heavy-tailed distributions. The empirical second moment of $V_k^T \nabla \ell$ is given by $V_k^T \nabla \ell \nabla \ell^T V_k$. The total variance in the empirical projection subspace is generally measured by the trace of the second moment denoted as $\text{tr}(V_k^T \nabla \ell \nabla \ell^T V_k)$.

DPSGD lies in strict mathematical definitions Dwork et al. (2006); Abadi et al. (2016) and composition theorems Kairouz et al. (2015); Mironov (2017); Dong et al. (2022). Definition 3.1 gives a formal definition of differential privacy (DP).

**Definition 3.1** (**Differential Privacy**). *A randomized algorithm $M$ is $(\epsilon, \delta)$-differentially private if for any two neighboring datasets $D$, $D'$ differ in exactly one data point and any event $Y$, we have*

$$\mathbb{P}(M(D) \in Y) \leq \exp(\epsilon) \cdot \mathbb{P}(M(D') \in Y) + \delta, \tag{2}$$

*where $\epsilon$ is the privacy budget and $\delta$ is a small probability.*

### 3.2 Assumptions

A substantial amount of research has shown that even on the simplest MNIST dataset, gradient descent exhibits heavy-tailed behavior Gurbuzbalaban et al. (2021), allowing our theoretical framework to center around a state-of-the-art heavy-tailed distribution, sub-Weibull distribution Vladimirova et al. (2020), which generalizes the sub-Gaussian and sub-Exponential families to potentially heavier-tailed ones. Sub-Weibull distributions are characterized by a positive tail index $\theta$, with $\theta = \frac{1}{2}$ represents sub-Gaussian distributions, $\theta = 1$ represents heavy-tailed sub-Exponential distributions, and $\theta > 1$ represents heavier-tailed ones.

**Assumption 3.1** (**Sub-Weibull Gradient Noise**). *Conditioned on the iterates, we make an assumption that the gradient noise $\nabla\ell(\mathbf{w}_t) - \nabla L(\mathbf{w}_t)$ satisfies $\mathbb{E}[\nabla\ell(\mathbf{w}_t) - \nabla L(\mathbf{w}_t)] = 0$ and $\|\nabla\ell(\mathbf{w}_t) - \nabla L(\mathbf{w}_t)\|_2 \sim subWeibull(\theta, K)$ for some positive $K$, such that $\theta \geq \frac{1}{2}$, and have*

$$\mathbb{E}_t[\exp((\|\nabla\ell(\mathbf{w}_t) - \nabla L(\mathbf{w}_t)\|_2/K)^{\frac{1}{\theta}})] \leq 2.$$

Assumption 3.1 is a relaxed version of gradient noise following sub-Gaussian distributions, that is $\mathbb{E}_t[\exp((\|\nabla\ell(\mathbf{w}_t) - \nabla L(\mathbf{w}_t)\|_2/K)^2)] \leq 2$, which means that finding upper bounds for moment generating function (MGF) under Assumption 3.1 is impracticable by standard tools Vladimirova et al. (2020). Thus, the truncated tail theory Bakhshizadeh et al. (2023) and martingale difference inequality Madden et al. (2020) play a crucial role in our analysis.

**Assumption 3.2** ($\beta$-**Smoothness**). *The loss function $\ell$ is $\beta$-smooth, for any $\mathbf{w}_t, \mathbf{w}_t' \in \mathbb{R}^d$, we have*

$$\|\nabla\ell(\mathbf{w}_t) - \nabla\ell(\mathbf{w}_t')\|_2 \leq \beta\|\mathbf{w}_t - \mathbf{w}_t'\|_2.$$

**Assumption 3.3** (**G-Bounded**). *For any $\mathbf{w} \in \mathbb{R}^d$ and per-sample $z$, there exists positive real numbers $G > 0$, and the expectation gradient satisfies*

$$\|\nabla L(\mathbf{w}_t)\|_2^2 \leq G.$$

Assumption 3.2 is widely used in optimization literature Foster et al. (2018); Zhou et al. (2021); Li & Liu (2022) and is essential for ensuring the convergence of gradients to zero Li & Orabona (2020). Compared to the bounded stochastic gradient assumption Zhou et al. (2021); Li & Liu (2022; 2023), i.e., $\|\nabla\ell(\mathbf{w}_t, z_i)\|_2^2 \leq G$, Assumption 3.3 is milder, with our results being more applicable.

## 4 Heavy-tailed DPSGD with High Probability Bounds

To analyze the performance degradation of DPSGD and the imperative of discriminative clipping in heavy-tailed scenarios, we first present the current optimal optimization error of DPSGD Yang et al. (2022); Bu et al. (2024); Zhou et al. (2021) on expectation bounds and the representative heavy-tailed results with high probability bounds in Table 1. Most works with expectation bounds rely on the assumption of light-tailed distributions, rough clipping analysis, or additional conditions, and cannot be adapted to heavy-tailed DPSGD. Moreover, while high probability bounds are widely adopted in the domain of SGD, applying them to DPSGD is challenging due to the additional unbounded privacy noise introduced by DPSGD. This makes it difficult to provide empirical guidance for determining the clipping threshold under rigorous theoretical guarantees. To fill this gap, we analyze the high probability bound for classical DPSGD on the gradients of empirical risks, denoted as $\|\nabla L_S(\mathbf{w}_t)\|_2$, under the heavy-tailed sub-Weibull assumption, as stated in Theorem 4.1. Consequently, we can use this theorem to establish the relationship between the clipping threshold and the heavy tail index.

**Theorem 4.1** (**Convergence of Heavy-tailed DPSGD**). *Under Assumptions 3.1 and 3.2, let $\mathbf{w}_t$ be the iterate produced by DPSGD with learning rate $\eta_t = \frac{1}{\sqrt{T}}$. Suppose that $T = \mathbb{O}(\frac{n\epsilon}{\sqrt{d\log(1/\delta)}})$,*

Table 1: Summary of state-of-the-art optimization results under non-convex conditions, where 'symmetry' means the gradient noise $\xi$ satisfies $\mathbb{P}(\xi) = \mathbb{P}(-\xi)$, and $\hat{\log}(\cdot) := \log^{\max(0,\theta-1)}(\cdot)$.

| Measure | Method | DPSGD | SGD | Assumption | Clipping |
|---|---|---|---|---|---|
| Expectation | Yang et al. (2022) | $\mathbb{O}\left(\frac{\sqrt[4]{d\log(T/\delta)}}{(n\epsilon)^{\frac{1}{2}}}\right)$ | $\times$ | bounded variance | $\checkmark$ |
| | Bu et al. (2024) | $\mathbb{O}\left(\frac{\sqrt{d}}{n\epsilon}\right)$ | $\mathbb{O}\left(\frac{d}{\sqrt[4]{T}}\right)$ | symmetry | $\checkmark$ |
| | Zhou et al. (2021) | $\mathbb{O}\left(\frac{k}{n\epsilon}\right)$ | $\times$ | public subspace | $\times$ |
| High probability | Madden et al. (2020) | $\times$ | $\mathbb{O}\left(\frac{\sqrt{\log(T)}\log^{\theta}(1/\delta)}{\sqrt{T}} + \frac{\hat{\log}(T/\delta)\log(1/\delta)}{\sqrt{T}}\right)$ | heavy tails | $\times$ |
| | Li & Liu (2022) | $\times$ | $\mathbb{O}\left(\frac{\log^{2\theta}(1/\delta)\log(T)}{\sqrt{T}} + \frac{\hat{\log}(T/\delta)\log(1/\delta)}{\sqrt{T}}\right)$ | heavy tails | $\times$ |
| | Li & Liu (2022) | $\times$ | $\mathbb{O}\left(\frac{\log^{\theta}(T/\delta)\log(T)}{\sqrt{T}} + \frac{\log^{2\theta+1}(T)\log(T/\delta)}{\sqrt{T}}\right)$ | heavy tails | $\checkmark$ |
| | Our DPSGD | $\mathbb{O}\left(d^{\frac{1}{4}}\log^{\frac{5}{4}}(T/\delta)\cdot\frac{\hat{\log}(T/\delta)\log^{2\theta}(\sqrt{T})}{(n\epsilon)^{\frac{1}{2}}}\right)$ | | heavy tails | $\checkmark$ |
| | Our DC-DPSGD | $\mathbb{O}\left(d^{\frac{1}{4}}\log^{\frac{5}{4}}(T/\delta)\cdot\left(p\frac{\hat{\log}(T/\delta)\log^{2\theta}(\sqrt{T})}{(n\epsilon)^{\frac{1}{2}}} + (1-p)(\frac{\log(\sqrt{T})}{(n\epsilon)^{\frac{1}{2}}})\right)\right)$ | | heavy tails | $\checkmark$ |

$T \geq 1$, and $c = \max\left(4K\log^{\theta}(\sqrt{T}), 39K\log^{\theta}(2/\delta)\right)$, where $d$ is the number of model parameters. For any $\delta \in (0,1)$, with probability $1-\delta$, we have:

$$\frac{1}{T}\sum_{t=1}^{T}\min\left\{\|\nabla L_S(\mathbf{w}_t)\|_2, \|\nabla L_S(\mathbf{w}_t)\|_2^2\right\} \leq \mathbb{O}\left(\underbrace{\frac{d^{\frac{1}{4}}\log^{\frac{1}{4}}(T/\delta)}{(n\epsilon)^{\frac{1}{2}}}}_{\text{privacy}}\underbrace{\log(T/\delta)\hat{\log}(T/\delta)}_{\text{tail probability}}\underbrace{\log^{2\theta}(\sqrt{T})}_{\text{clipping}}\right),$$

where $\hat{\log}(T/\delta) := \log^{\max(0,\theta-1)}(T/\delta)$.

*Proof.* The proof is provided in Appendix B due to space limitations. $\square$

In Theorem 4.1, we divide the optimization bound on the gradients of empirical risks into **privacy error**, **high probability tail error**, and **clipping error**. Overall, we can derive that, as $\theta$ ascends, the optimization performance of DPSGD gradually deteriorates, because both $\hat{\log}(T/\delta)$ (appearing when $\theta > 1$) and $\log^{2\theta}(\sqrt{T})$ increase. Next, we compare our heavy-tailed DPSGD result to existing works.

- **Compared to existing DPSGD with expectation bounds.** Our work achieves the current optimal results for classical DPSGD based on weaker assumptions and is extensible to heavy-tailed scenarios. When $\theta = \frac{1}{2}$ (i.e., light-tailed scenarios), the convergence bound becomes $\mathbb{O}(d^{\frac{1}{4}}\log^{\frac{5}{4}}(T/\delta)\log(\sqrt{T})/(n\epsilon)^{\frac{1}{2}})$. It aligns with the current optimal expectation bounds of DPSGD, i.e., $\mathbb{O}(\sqrt[4]{d\log(1/\delta)}/(n\epsilon)^{\frac{1}{2}})$ in Yang et al. (2022), except for an extra high probability term $\log(T/\delta)\log(\sqrt{T})$, while excluding the requirements of bounded variance, symmetric gradients Bu et al. (2024), and public data Zhou et al. (2021).

- **Compared to existing SGD with high probability bounds.** Our high probability term demonstrates improved performance in terms of clipping error. Specifically, the dependency on the confidence parameter $1/\delta$ is logarithmic, similar to the optimal high probability bounds for SGD Li & Liu (2022; 2023); Madden et al. (2020), as shown in Table 1. Moreover, suppose $\sqrt{T} = (n\epsilon)^{\frac{1}{2}}/\sqrt[4]{d\log(1/\delta)}$, our DPSGD result can be transformed to $\mathbb{O}(\log(T/\delta)\hat{\log}(T/\delta)\log^{2\theta}(\sqrt{T})/\sqrt{T})$, improving the clipping error from $\log^{2\theta+1}(T)$ in Li & Liu (2022) to $\log^{2\theta}(\sqrt{T})$.

To our knowledge, we are the first to use the high probability bound as a measure to analyze the optimization performance in heavy-tailed DPSGD.

**Tail-aware clipping mechanism.** We can further observe from Theorem 4.1 that the theoretical value of $c$ is positively correlated to $\theta$, which means the ideal clipping threshold should scale up as the heavy tail index $\theta$ increases. Otherwise, the convergence bound may become sub-optimal and even collapse. Intuitively, using existing empirical guidance for clipping threshold under the heavy-tailed assumption will cause higher clipping losses for tailed gradients with larger $L_2$ norms. This motivates us to design a tail-aware clipping mechanism to improve the performance of DPSGD.

## 5 DISCRIMINATIVE CLIPPING DPSGD

In this section, we present our approach DC-DPSGD that effectively handles heavy-tailed gradients with a novel tail-aware clipping mechanism, as illustrated in Figure 2. The rationale is to divide gradients following a heavy-tailed sub-Weibull distribution into two parts: light body and heavy tail, and employ different clipping thresholds for the two parts respectively, where a small clipping threshold is applied for light body and a larger one for heavy tail to mitigate the extra clipping loss.

Specifically, DC-DPSGD consists of two steps. In the first step, we propose a subspace identification technique to distinguish gradients from light body and heavy tail in a privacy-preserving way. To satisfy differential privacy, noise with scale $\sigma_{\mathrm{tr}}$ is added to this step (Section 5.1). In the second step, we present a discriminative clipping method that utilizes different clipping thresholds for the two parts and adds DP noise with scale $\sigma_{\mathrm{dp}}$ for privacy preservation (Section 5.2). For a fair comparison to existing DPSGD works, the total privacy budget allocated by DC-DPSGD to $\epsilon_{\mathrm{tr}}$ and $\epsilon_{\mathrm{dp}}$ must be equal to the privacy budget $\epsilon$ in DPSGD variants, i.e., $\epsilon = \epsilon_{\mathrm{tr}} + \epsilon_{\mathrm{dp}}$. Algorithm 1 presents the detailed steps of DC-DPSGD, and Theorem 5.1 gives its privacy guarantee.

**Theorem 5.1 (Privacy Guarantee).** *There exist constants $m_1$ and $m_2$ such that for any $\epsilon_{\mathrm{tr}} \leq m_1 q^2 T$, $\epsilon_{\mathrm{dp}} \leq m_1 q^2 T$ and $\delta > 0$, the noise multiplier $\sigma_{\mathrm{tr}}^2 = \frac{m_2 T q^2 \ln \frac{1}{\delta}}{\epsilon_{\mathrm{tr}}^2}$ and $\sigma_{\mathrm{dp}}^2 = \frac{m_2 T q^2 \ln \frac{1}{\delta}}{\epsilon_{\mathrm{dp}}^2}$ over $T$ iterations, where $q = \frac{B}{n}$, and DC-DPSGD is $(\epsilon_{\mathrm{tr}} + \epsilon_{\mathrm{dp}}, \delta)$-differentially private.*

*Proof.* According to the results of trace sorting, we apply two clipping thresholds for gradient perturbation, making it essential to reanalyze the unified privacy guarantees of our composition mechanism. Due to space limitations, we defer the proof to Appendix C for more details. $\square$

### 5.1 SUBSPACE IDENTIFICATION

We note that the heavy tail index $\theta$ reflects the per-sample gradient norm, which means samples drawn from heavier-tailed distributions are more likely to exhibit larger $L_2$ norms, and their subspace eigenvectors differ from those of light-tailed distributions. Due to the high-dimensional nature of gradients, their normalized versions act as mutually orthogonal eigenvectors Wainwright (2019). By measuring the similarity between the empirical normalized gradients and the underlying heavy-tailed subspace, a higher similarity indicates closer alignment with the heavy tail, while a lower similarity implies the light body. Given that the normalized gradients retain directional information with bounded sensitivity $L_2$ norm (equal to 1), this allows for bypassing the unbounded norm of heavy-tailed gradients and identifying different responses of gradients in the heavy-tailed subspace.

Specifically, we first construct a projection matrix composed of $k$ random orthogonal unit vectors $[v_1, ..., v_k]$ consistent with heavy-tailed sub-Weibull distributions ($\theta > \frac{1}{2}$), and then divide gradients

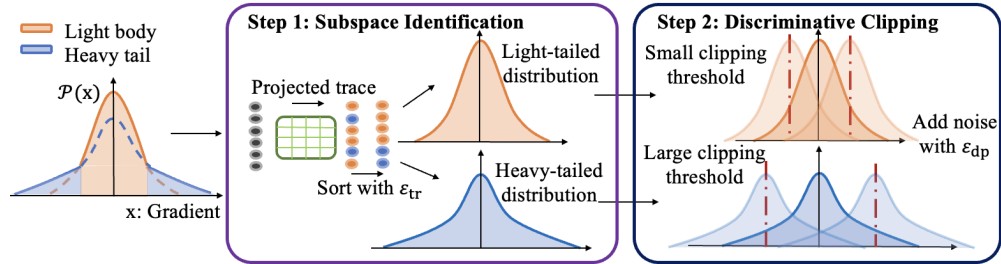

Figure 2: Overview of DC-DPSGD.

---

**Algorithm 1** Discriminative Clipping DPSGD

---

**Input**: Private batch size $B$, heavy-tailed ratio $p$, heavy-tailed clipping threshold $c_1$, light-tailed clipping threshold $c_2$, learning rate $\eta_t$ and subspace dimension $k$.

1: Initialize $\mathbf{w}_0$ randomly.
2: **for** $e \in$ Epochs $E$ **do**
3:     Initialize $V_{t,k}$ to None.
4:     **for** $t \in$ Iterations $T$ **do**
5:         Take a random batch $B$ with sampling ratio $B/n$ and $\mathbf{g}_t(z_i) = \nabla\ell(\mathbf{w}_t, z_i)$.
6:         Extract orthogonal vectors $[v_1, ..., v_k]$ from sub-Weibull distributions and construct projection subspace with $V_{t,k}V_{t,k}^T = \frac{1}{k}\sum_{i=1}^{k} v_i v_i^T$.
7:         Normalize per-sample gradient $\hat{\mathbf{g}}_t(z_i) = \mathbf{g}_t(z_i)/\|\mathbf{g}_t(z_i)\|$.
8:         Calculate the trace $\lambda_{t,i}^{\mathrm{tr}}$ of the projected second moment $V_{t,k}^T \hat{\mathbf{g}}_t(z_i)\hat{\mathbf{g}}_t^T(z_i)V_{t,k}$.
9:         Perturb traces $\widetilde{\lambda}_{t,i}^{\mathrm{tr}} = \lambda_{t,i}^{\mathrm{tr}} + \mathbb{N}(0, \sigma_{\mathrm{tr}}^2)$ and identify top-$pB$ based on sorted $\widetilde{\lambda}_{t,i}^{\mathrm{tr}}$.
10:       Discriminative clipping: clip per-sample gradient and add noise with $c_1$ and $c_2$.
           For heavy tail: $\overline{\mathbf{g}}_t^{\mathrm{tail}}(z_i) = \mathbf{g}_t^{\mathrm{tail}}(z_i)/\max(1, \frac{\|\mathbf{g}_t^{\mathrm{tail}}(z_i)\|_2}{c_1}) + \mathbb{N}(0, c_1^2\sigma_{\mathrm{dp}}^2\mathbb{I}_d)$
           For light body: $\overline{\mathbf{g}}_t^{\mathrm{body}}(z_i) = \mathbf{g}_t^{\mathrm{body}}(z_i)/\max(1, \frac{\|\mathbf{g}_t^{\mathrm{body}}(z_i)\|_2}{c_2}) + \mathbb{N}(0, c_2^2\sigma_{\mathrm{dp}}^2\mathbb{I}_d)$
11:       Weighted average $\widetilde{\mathbf{g}}_t = \frac{1}{B}\left(\sum_{i=1}^{pB}\overline{\mathbf{g}}_t^{\mathrm{tail}}(z_i) + \sum_{i=1}^{(1-p)B}\overline{\mathbf{g}}_t^{\mathrm{body}}(z_i)\right)$.
12:       Update $\mathbf{w}_{t+1} = \mathbf{w}_t - \eta_t\widetilde{\mathbf{g}}_t$.
13:     **end for**
14: **end for**

---

into the light body or heavy tail region according to the projected trace $\lambda_{t,i}^{\mathrm{tr}} = V_{t,k}^T\hat{\mathbf{g}}_t(z_i)\hat{\mathbf{g}}_t^T(z_i)V_{t,k}$, where the larger $\lambda_{t,i}^{\mathrm{tr}}$ indicates a higher similarity between the gradient and the projection subspace, and $V_{t,k}V_{t,k}^T = \frac{1}{k}\sum_{i=1}^{k}v_iv_i^T$ is the approximated second moment. To estimate the utility of the identification, we need to bound the skewing between the empirical second moment and the population second moment, i.e., $\|V_kV_k^T - \mathbb{E}[V_kV_k^T]\|_2$. It is worth noting that in line 9 of Algorithm 1, as the publicly available traces are sorted to identify the top $p\%$ heavy-tailed gradients, which may expose intrinsic preferences, extra noise is injected. According to Ahlswede-Winter Inequality Wainwright (2019), we analyze the error of subspace skewing in a high probability form.

**Theorem 5.2 (Subspace Skewing for Identification).** *Assume that the empirical second moment matrix $M = V_kV_k^T \in \mathbb{R}^{d \times d}$ with $V_k^T V_k = \mathbb{I}_k$ approximates the population second moment matrix $\hat{M} = \hat{V}_k\hat{V}_k^T = \mathbb{E}_{V_k \sim \mathscr{P}}[V_kV_k^T]$, $\lambda_{t,i}^{\mathrm{tr}} = \mathrm{tr}(V_k^T\hat{\mathbf{g}}_t(z_i)\hat{\mathbf{g}}_t^T(z_i)V_k)$ and $\hat{\lambda}_t^{\mathrm{tr}} = \mathrm{tr}(\hat{V}_k^T\hat{\mathbf{g}}_t(z_i)\hat{\mathbf{g}}_t^T(z_i)\hat{V}_k)$, for any gradient $\hat{\mathbf{g}}_t(z_i)$ that satisfies $\|\hat{\mathbf{g}}_t(z_i)\|_2 = 1$, $\zeta_t^{\mathrm{tr}} \sim \mathbb{N}(0, \sigma_{\mathrm{tr}}^2)$, with probability $1 - \delta_m - \delta_{\mathrm{tr}}$:*

$$|\lambda_{t,i}^{\mathrm{tr}} - \hat{\lambda}_t^{\mathrm{tr}} + \zeta_t^{\mathrm{tr}}| \le \frac{4\log(2d/\delta_m)}{k} + \frac{m_2\sqrt{B}\log^{\frac{1}{2}}(1/\delta_{\mathrm{tr}})}{d^{\frac{1}{2}}},$$

*where $\delta_m, \delta_{\mathrm{tr}} \in (0, 1)$ are introduced by concentration inequalities and DP noise respectively.*

By comparing the magnitudes $\log(2d/\delta_m)/k$ and $\log^{\frac{1}{2}}(1/\delta_{\mathrm{tr}})/d^{\frac{1}{2}}$ in Theorem 5.2, it is evident that the first term dominates since $d \gg k$ (please refer to Appendix D for more discussion). Thus, the error is negligible when $k$ is large, indicating that the gradients can be correctly identified with high probability, guaranteed by $1 - \delta'_m$, where $\delta'_m = \delta_{\mathrm{tr}} + \delta_m$.

## 5.2 DISCRIMINATIVE CLIPPING

Assuming that the gradients are classified into the correct heavy tail and light body regions, we then apply two different clipping thresholds (denoted as $c_1$ and $c_2$) in our discriminative clipping method for the tail and body gradients, respectively. This way, we can reduce tail gradients' clipping losses and obtain faster DPSGD convergence, according to the analysis in Section 4.

Specifically, the tail probability $\mathbb{P}(|X| > x) = \exp(-I(x))\ \forall x > 0$ of the sub-Weibull variables $X \sim subW(\theta, K)$ exhibits two different behaviors: **(1) Light body**: for small $x$ values, the tail rate capturing function $I(x)$ decays like a sub-Gaussian tail. **(2) Heavy tail**: for $x$ greater than the normal convergence region, i.e., $x \ge x_{\max}$ is a large deviation region, its decay is slower than that

of the normal distribution, where $x_{\max}$ is a mathematical inflection point related to the population variance of underlying distributions Bakhshizadeh et al. (2023). Existing literature has studied the first region in the optimization analysis for DPSGD Bu et al. (2024); Yang et al. (2022); Xia et al. (2023); Cheng et al. (2022); Xiao et al. (2023); Sha et al. (2023), but they overlook the heavy-tailed behavior for the second region. In this paper, we not only study the optimization performance of each region, but also combine the two regions with discriminative clipping thresholds. To construct a clear convergence boundary for the two regions in heavy-tailed scenarios, we generalize the sharp heavy-tailed concentration Bakhshizadeh et al. (2023) and sub-Weibull Freedman inequality Madden et al. (2020) to truncate the theoretical distribution and find the optimal clipping threshold for each region. As a result, we have the following theorem.

**Theorem 5.3** (**Convergence of Discriminative Clipping**). *Under Assumptions 3.1, 3.2 and 3.3, let $\mathbf{w}_t$ be the iterate produced by DC-DPSGD with $T = \mathbb{O}(\frac{n\epsilon}{\sqrt{d\log(1/\delta)}})$, $T \geq 1$ and $\eta_t = \frac{1}{\sqrt{T}}$. Define $\hat{\log}(T/\delta) := \log^{\max(0,\theta-1)}(T/\delta)$, $\Gamma(x) := \int_0^\infty t^{x-1}e^{-t}dt$, $a = 2$ if $\theta = \frac{1}{2}$, $a = (4\theta)^{2\theta}e^2$ if $\theta \in (\frac{1}{2}, 1]$, and $a = (2^{2\theta+1}+2)\Gamma(2\theta+1) + \frac{2^{3\theta}\Gamma(3\theta+1)}{3}$ if $\theta > 1$, for any $\delta \in (0,1)$:*

*(i).* **In the heavy tail region**:
   *suppose that $c_1 = \max\left(4^\theta 2K\log^\theta(\sqrt{T}), 4^\theta 33K\log^\theta(2/\delta)\right)$, with probability $1-\delta$,*

$$\frac{1}{T}\sum_{t=1}^T \min\left\{\|\nabla L_S(\mathbf{w}_t)\|_2, \|\nabla L_S(\mathbf{w}_t)\|_2^2\right\} \leq \mathbb{O}\left(\frac{d^{\frac{1}{4}}\log^{\frac{5}{4}}(T/\delta)\hat{\log}(T/\delta)\log^{2\theta}(\sqrt{T})}{(n\epsilon)^{\frac{1}{2}}}\right).$$

*(ii).* **In the light body region**:
   *suppose that $c_2 = \max\left(2\sqrt{2a}K\log^{\frac{1}{2}}(\sqrt{T}), 33\sqrt{2a}K\log^{\frac{1}{2}}(2/\delta)\right)$, with probability $1-\delta$,*

$$\frac{1}{T}\sum_{t=1}^T \min\left\{\|\nabla L_S(\mathbf{w}_t)\|_2, \|\nabla L_S(\mathbf{w}_t)\|_2^2\right\} \leq \mathbb{O}\left(\frac{d^{\frac{1}{4}}\log^{\frac{5}{4}}(T/\delta)\log(\sqrt{T})}{(n\epsilon)^{\frac{1}{2}}}\right).$$

*Proof.* We provide a proof sketch below and defer the full proof to Appendix E. In DC-DPSGD, the convergence bounds for the two regions correspond to $c_1$ and $c_2$, respectively. First, we optimize the theoretical tools by transforming the concentration inequalities for the sum of sub-Weibull random variables $X$ into two-region versions distinguished by the tail probability $\mathbb{P}(|X| > x)$, namely sub-Gaussian tail decay rate $\exp(-x^2)$ and heavy-tailed decay rate $\exp(-x^{1/\theta})$, $\theta > \frac{1}{2}$. Then, we analyze the high probability bounds for the gradient noise of DPSGD in each region. In the heavy tail region, we make the inequality $\mathbb{P}(\|\mathbf{g}_t - \nabla L_S(\mathbf{w}_t)\|_2 > c_1) \leq 2\exp(-c_1^{1/\theta})$ hold and derive the dependence of factor $\log^\theta(1/\delta)$ for $c_1$. In the light body region, we have $\mathbb{P}(\|\mathbf{g}_t - \nabla L_S(\mathbf{w}_t)\|_2 > c_2) \leq 2\exp(-c_2^2)$, resulting in the factor $\log^{1/2}(1/\delta)$ of $c_2$. Next, we investigate the high probability error on the unbounded DPSGD privacy noise using Gaussian distribution properties. Finally, we integrate the results regarding gradient noise and privacy noise to determine the optimal clipping thresholds for both regions and achieve faster convergence rates for the optimization performance. □

From Theorem 5.3, we can observe that when gradients fall into the light body region, our result does not contain the heavy-tailed index $\theta$, implying that the optimization performance is not affected by $\theta$ and always converges with respect to the light-tailed sub-Gaussian rate. When the gradients are in the heavy-tailed region, the convergence will be the same as that of classical heavy-tailed DPSGD, which becomes deteriorated as $\theta$ increases. In summary, compared to existing optimization results that fully rely on the heavy-tailed index $\theta$ Li & Liu (2022); Madden et al. (2020), our DC-DPSGD bound only increases with $\theta$ for partial gradients (i.e., heavy-tailed gradients), leading to improved optimization performance, notably when $\theta > 1/2$.

### 5.3 UNIFORM BOUND FOR DC-DPSGD

Notice that in the subspace identification method, we use the trace of the second moment to approximate the population variance of projected gradients, and the approximation error is bounded by a high probability of $1-\delta'_m$ in Theorem 5.2. Thus, we can analyze the convergence by combining Theorems 5.2 and 5.3 to derive the uniform bound for Algorithm 1, as stated in Theorem 5.4.

**Theorem 5.4** (**Uniform Bound for DC-DPSGD**). *Given Assumptions 3.1, 3.2 and 3.3, we can obtain that for any $\delta' \in (0, 1)$, with probability $1 - \delta'$ and $\mathcal{C}_u := \sum_{t=1}^{T} \min\{\|\nabla \hat{L}_S(\mathbf{w}_t)\|_2^2, \|\nabla \hat{L}_S(\mathbf{w}_t)\|_2\}$:*

$$\mathcal{C}_{\mathrm{u}} \leq p * \mathbb{O}\left(\frac{d^{\frac{1}{4}} \log^{\frac{5}{4}}(T/\delta)\hat{\log}(T/\delta) \log^{2\theta}(\sqrt{T})}{(n\epsilon)^{\frac{1}{2}}}\right) + (1 - p) * \mathbb{O}\left(\frac{d^{\frac{1}{4}} \log^{\frac{5}{4}}(T/\delta) \log(\sqrt{T})}{(n\epsilon)^{\frac{1}{2}}}\right),$$

*where $p$ is the ratio of heavy-tailed gradients, $\hat{\log}(T/\delta) = \log^{\max(0,\theta-1)}(T/\delta)$, $\delta' = \delta'_m + \delta$, with $\delta'_m$ being the error of subspace identification, and $\delta$ being the convergence probability of DC-DPSGD.*

Theorem 5.4 indicates that the optimization performance of DC-DPSGD is composed of $p$-weighted average bounds, where the heavy-tailed convergence rate merely accounts for a portion of $p$, with the rest made up of the light body rate. Therefore, our bound minimizes the dependency on $\theta$ from $\hat{\log}(T/\delta) \log^{2\theta}(\sqrt{T})$ to $\log(\sqrt{T})$ with high probability $(1 - p) * (1 - \delta')$, which is tighter than heavy-tailed DPSGD (Theorem 4.1). According to the statistical properties Vershynin (2018); Wainwright (2019), approximately 5%-10% of data points fall into the tail in practice, that is, $p \in [5\%, 10\%]$.

## 6 EXPERIMENTS

### 6.1 EXPERIMENTAL SETUP

**Datasets and models.** We evaluate DC-DPSGD on five real-world datasets, including MNIST, FMNIST, CIFAR10, ImageNette Deng et al. (2009) for image classification, and E2E Dušek et al. (2020) for natural language generation. Moreover, we use two heavy-tailed versions: namely CIFAR10-HT Cao et al. (2019) (a heavy-tailed version of CIFAR10) and ImageNette-HT (modified on Park et al. (2021)) to evaluate the performance under heavy tail assumption.

For MNIST and FMNIST, we use a two-layer CNN model. For CIFAR10 and CIFAR10-HT, we fine-tune SimCLRv2 pre-trained by unlabeled ImageNet and ResNeXt-29 pre-trained by CIFAR100 Tramer & Boneh (2021) with a linear classifier, respectively. For ImageNette and ImageNette-HT, we adopt the same setting as Bu et al. (2024) and ResNet9 without pre-train. For E2E, we use a transformer-based GPT-2 model (163 million parameters) and fine-tune it with the dataset. We evaluate image classification tasks using accuracy that measures the portion of correct predictions, and natural language generation tasks using the BLEU score Papinesi (2002) that measures the quality of generated data with a modified n-gram precision score.

**Baselines.** We compare DC-DPSGD with three differentially private baselines: DPSGD with Abadi's clipping Abadi et al. (2016), Auto-S/NSGD Bu et al. (2024); Yang et al. (2022), DP-PSAC Xia et al. (2023), and a non-private baseline: non-DP ($\epsilon = \infty$).

**Implementation details.** We set $c_2 = 0.1$, $B = 128$, and $\eta = 0.1$ for MNIST and FMNIST. For CIFAR10, we set $c_2 = 0.1$, $B = 256$, and $\eta = 1$. For ImageNette, we set $c_2 = 0.15$, $\eta = 0.0001$ and $B = 1000$. For E2E, we adopt the DPAdam optimizer and use the same settings as Li et al. (2022), where $c_2 = 0.1$. By default, we set $c_1 = 10 * c_2$, and heavy-tailed ratio $p$ is 10%. We implement per-sample clipping in DPSGD by BackPACK Dangel et al. (2020) and allocate the privacy budget equally according to $\epsilon = \epsilon_{\mathrm{tr}+\epsilon_{\mathrm{dp}}}$.

### 6.2 EFFECTIVENESS EVALUATION

Table 2 summarizes the comparison results between DC-DPSGD and baselines. We observe that on normal datasets, DC-DPSGD outperforms DPSGD, Auto-S, and DP-PSAC by up to 4.57%, 5.42%, and 4.99%, respectively. While on heavy-tailed datasets, the corresponding improvements are 8.34%, 9.72%, and 9.55%. The reason is that our approach places a larger clipping threshold for heavy-tailed gradients, thereby preserving more information about them and improving accuracy. Moreover, we demonstrate the trajectories of training accuracy in Figure 3, indicating that the optimization performance of DC-DPSGD is superior to existing clipping mechanisms.

We then evaluate the effects of four parameters on test accuracy, including the subspace-$k$, the allocation of privacy budget $\epsilon$, the heavy tail index sub-Weibull-$\theta$, and the heavy tail ratio $p$, with other parameters kept at default. The results are shown in Table 3. We can see that the test accuracy increases with the value of $k$, which aligns with the theoretical analysis that the trace error is related to $\mathbb{O}(1/k)$ and has a small impact on the results. For the allocation of privacy budget between $\epsilon_{\mathrm{tr}}$ and

Table 2: Effectiveness comparison between DC-DPSGD and baselines.

| Dataset | DP $(\epsilon, \delta)$ | Accuracy % or BLEU % | | | | |
|---|---|---|---|---|---|---|
| | | DPSGD | Auto-S | DP-PSAC | Ours | non-DP |
| MNIST | $(8, 1e^{-5})$ | 97.65±0.09 | 97.55±0.16 | 97.67±0.06 | **98.72±0.02** | 99.10±0.02 |
| FMNIST | $(8, 1e^{-5})$ | 83.23±0.10 | 82.38±0.15 | 82.81±0.18 | **87.80±0.47** | 89.95±0.32 |
| CIFAR10 | $(8, 1e^{-5})$ | 93.31±0.01 | 93.28±0.06 | 93.30±0.03 | **94.05±0.11** | 94.62±0.03 |
| CIFAR10 | $(4, 1e^{-5})$ | 93.06±0.09 | 93.08±0.06 | 93.11±0.08 | **93.42±0.14** | 94.62±0.03 |
| ImageNette | $(8, 1e^{-4})$ | 66.81±0.42 | 65.57±0.85 | 65.68±1.71 | **69.29±0.19** | 71.67±0.49 |
| CIFAR10-HT | $(8, 1e^{-5})$ | 57.98±0.59 | 58.30±0.61 | 57.99±0.58 | **62.57±1.03** | 71.74±0.65 |
| ImageNette-HT | $(8, 1e^{-4})$ | 25.36±1.71 | 23.98±2.00 | 24.15±1.99 | **33.70±0.91** | 39.91±1.46 |
| E2E (full fine-tune) | $(8, 1e^{-5})$ | 63.189 | 63.600 | 63.627 | **65.380** | 69.463 |
| E2E (LoRA fine-tune) | $(8, 1e^{-5})$ | 63.389 | 63.518 | 63.502 | **64.150** | 69.692 |

Table 3: Effects of parameters on test accuracy.

| Dataset | Subspace-$k$ | | | $\epsilon_{tr} + \epsilon_{dp}$ | | | Sub-Weibull-$\theta$ | | | Tail Ratio-$p$ | | |
|---|---|---|---|---|---|---|---|---|---|---|---|---|
| | None | 100 | 200 | 2+6 | 4+4 | 6+2 | 1/2 | 1 | 2 | 5% | 10% | 20% |
| CIFAR10 | 93.07 | 93.82 | 94.05 | 93.92 | 94.05 | 93.37 | 93.88 | 93.99 | 94.05 | 93.90 | 94.05 | 93.63 |
| CIFAR10-HT | 57.27 | 61.60 | 62.57 | 62.54 | 62.57 | 60.07 | 61.58 | 62.28 | 62.57 | 61.12 | 62.57 | 61.70 |

$\epsilon_{dp}$, we find that a balanced allocation strategy can mitigate excessive noise caused by a one-sided small privacy budget. For subspace distribution, since the 'HT' dataset is extracted through sub-Exponential distributions, the gradient exhibits a heavier tail phenomenon. Therefore, the accuracy increases as $\theta$ becomes larger. For the tail ratio, $p = 10\%$ achieves better results. If $p$ is too low, it fails to mitigate clipping loss, while if $p$ is too large, it could introduce additional noise.

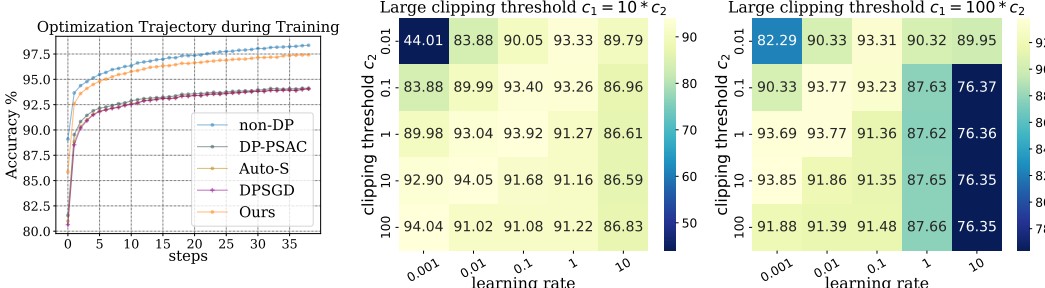

Figure 3: Optimization performance during CIFAR10 Training. Figure 4: Test accuracy heatmap on CIFAR10 with $c_1$, $c_2$ and $\eta$.

### 6.3 GUIDANCE FOR THE LARGE CLIPPING THRESHOLD

We now validate our empirical guidance for the clipping threshold in Theorem 5.3. The results in Figure 4 indicate that the optimal ratio is approximately $c_1 \approx 10c_2$. We note that when $c_1 = 100c_2$, the maximum performance declines noticeably, and when $c_1 = c_2$, it corresponds to classical DPSGD. From a theoretical perspective, given $\delta = 1e^{-5}$, $\eta/B = 0.04$, and $\theta \approx 2$ (following Gurbuzbalaban et al. (2021)), we can obtain $c_1 = \mathbb{O}(\log^{\theta}(1/\delta))$, which is $\sqrt{125}$ times larger than $c_2 = \mathbb{O}(\log^{1/2}(1/\delta))$, that is, $c_1 = \log^{3/2}(1/\delta)c_2$, i.e., $c_1 \approx 10c_2$. In conclusion, the optimal clipping threshold aligns with our empirical guidance.

## 7 CONCLUSION

In this paper, we propose a novel approach DC-DPSGD under the heavy-tailed assumption, which effectively reduces extra clipping loss in the heavy-tailed region. We rigorously analyze the high probability bound of the classic heavy-tailed DPSGD under non-convex conditions and obtain results matching the expectation bounds. Furthermore, we sharpen the weighted average optimization performance of DC-DPSGD. Extensive experiments on five real-world datasets demonstrate that DC-DPSGD outperforms three state-of-the-art clipping mechanisms.

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

## A  PRELIMINARIES

A random variable $X$ called a sub-Weibull random variable with tail parameter $\theta$ and scale factor $K$, which is denoted by $X \sim subW(\theta, K)$. We next introduce the equivalent properties and theoretical tools of sub-Weibull distributions.

### A.1  PROPERTIES

**Definition A.1** (Sub-Weibull Equivalent Properties Vladimirova et al. (2020)). *Let $X$ be a random variable and $\theta \geq 0$, and there exists some constant $K_1, K_2, K_3, K_4$ depending on $\theta$. Then the following characterizations are equivalent:*

1. *The tails of $X$ satisfy*
$$\exists K_1 > 0 \text{ such that } \mathbb{P}(|X| > t) \leq 2\exp(-(t/K_1)^{\frac{1}{\theta}}), \forall t > 0.$$

2. *The moments of $X$ satisfy*
$$\exists K_2 > 0 \text{ such that } \|X\|_p \leq K_2 p^\theta, \forall k \geq 1.$$

3. *The moment generating function (MGF) of $|X|^{\frac{1}{\theta}}$ satisfies*
$$\exists K_3 > 0 \text{ such that } \mathbb{E}[\exp((\lambda|X|)^{\frac{1}{\theta}})] \leq \exp((\lambda K_3)^{\frac{1}{\theta}}), \forall \lambda \in (0, 1/K_3).$$

4. *The MGF of $|X|^{\frac{1}{\theta}}$ is bounded at some point,*
$$\exists K_4 > 0 \text{ such that } \mathbb{E}[\exp((|X|/K_4)^{\frac{1}{\theta}})] \leq 2.$$

### A.2  THEORETICAL TOOLS

Based on the properties of sub-Weibull variables, we have the following high probability bounds and concentration inequalities for heavier tails as theoretical tools. Besides, We define $l_p$ norm as $\|\|_p$, for any $p \geq 1$.

**Lemma A.1.** *Let a variable $X \sim subW(\theta, K)$, for any $\delta \in (0, 1)$, then with probability $(1 - \delta)$ we have*
$$|X| \leq K \log^\theta (2/\delta).$$

*Proof.* Let $K_1 = K$ in Definition A.1, and take $t = K \log^\theta (2/\delta)$, then the inequality holds with probability $1 - \delta$. $\qquad \square$

**Lemma A.2** (Vladimirova et al. (2020); Madden et al. (2020)). *Let $X_1, ..., X_n$ are $subW(\theta, K_i)$ random variables with scale parameters $K_1, ...K_n$. $\forall x \geq 0$, we have*
$$\mathbb{P}(|\sum_{i=1}^{n} X_i| \geq x) \leq 2\exp(-(\frac{x}{g(\theta)\sum_{i=1}^{n} K_i})^{\frac{1}{\theta}})$$

*where $g(\theta) = (4e)^\theta$ for $\theta \leq 1$ and $g(\theta) = 2(2e\theta)^\theta$ for $\theta \geq 1$.*

**Lemma A.3** (Sub-Weibull Freedman Inequality Madden et al. (2020)). *Let $(\Omega, \mathscr{F}, (\mathscr{F}_i), \mathbb{P})$ be a filtered probability space. Let $(\xi_i)$ and $(K_i)$ be adapted to $(\mathscr{F}_i)$. Let $n \in \mathbb{N}$, then $\forall i \in [n]$, assume $K_{i-1} \geq 0$, $\mathbb{E}[\xi_i|\mathscr{F}_{i-1}] = 0$, and $\mathbb{E}[\exp((|\xi_i|/K_{i-1})^{\frac{1}{\theta}})|\mathscr{F}_{i-1}] \leq 2$ where $\theta \geq 1/2$. If $\theta > 1/2$, assume there exists $(m_i)$ such that $K_{i-1} \leq m_i$.*

*if $\theta = 1/2$, let $a = 2$, then $\forall x, \beta \geq 0$, $\alpha > 0$, and $\lambda \in [0, \frac{1}{2\alpha}]$,*
$$\mathbb{P}\left(\bigcup_{k \in [n]} \left\{ \sum_{i=1}^{k} \xi_i \geq x \text{ and } \sum_{i=1}^{k} aK_{i-1}^2 \leq \alpha \sum_{i=1}^{k} \xi_i + \beta \right\}\right) \leq \exp(-\lambda x + 2\lambda^2 \beta), \quad (3)$$

*and $\forall x, \beta, \lambda \geq 0$,*

$$\mathbb{P}\left(\bigcup_{k \in [n]} \left\{\sum_{i=1}^{k} \xi_i \geq x \text{ and } \sum_{i=1}^{k} aK_{i-1}^2 \leq \beta\right\}\right) \leq \exp(-\lambda x + \frac{\lambda^2}{2}\beta). \quad (4)$$

*If $\theta \in (\frac{1}{2}, 1]$, let $a = (4\theta)^{2\theta}e^2$ and $b = (4\theta)^{\theta}e$. $\forall x, \beta \geq 0$, and $\alpha \geq b\max_{i \in [n]} m_i$, and $\lambda \in [0, \frac{1}{2\alpha}]$,*

$$\mathbb{P}\left(\bigcup_{k \in [n]} \left\{\sum_{i=1}^{k} \xi_i \geq x \text{ and } \sum_{i=1}^{k} aK_{i-1}^2 \leq \alpha \sum_{i=1}^{k} \xi_i + \beta\right\}\right) \leq \exp(-\lambda x + 2\lambda^2\beta), \quad (5)$$

*and $\forall x, \beta \geq 0$, and $\lambda \in [0, \frac{1}{b\max_{i \in [n]} m_i}]$,*

$$\mathbb{P}\left(\bigcup_{k \in [n]} \left\{\sum_{i=1}^{k} \xi_i \geq x \text{ and } \sum_{i=1}^{k} aK_{i-1}^2 \leq \beta\right\}\right) \leq \exp(-\lambda x + \frac{\lambda^2}{2}\beta). \quad (6)$$

*If $\theta > 1$, let $\delta \in (0, 1)$. Let $a = (2^{2\theta+1} + 2)\Gamma(2\theta + 1) + 2^{3\theta}\Gamma(3\theta + 1)/3$ and $b = 2\log n/\delta^{\theta-1}$, where $\Gamma(x) = \int_0^{\infty} t^{x-1}e^{-t}dt$. $\forall x, \beta \geq 0$, $\alpha \geq b\max_{i \in [n]} m_i$, and $\lambda \in [0, \frac{1}{2\alpha}]$,*

$$\mathbb{P}\left(\bigcup_{k \in [n]} \left\{\sum_{i=1}^{k} \xi_i \geq x \text{ and } \sum_{i=1}^{k} aK_{i-1}^2 \leq \alpha \sum_{i=1}^{k} \xi_i + \beta\right\}\right) \leq \exp(-\lambda x + 2\lambda^2\beta) + 2\delta, \quad (7)$$

*and $\forall x, \beta \geq 0$, and $\lambda \in [0, \frac{1}{b\max_{i \in [n]} m_i}]$,*

$$\mathbb{P}\left(\bigcup_{k \in [n]} \left\{\sum_{i=1}^{k} \xi_i \geq x \text{ and } \sum_{i=1}^{k} aK_{i-1}^2 \leq \beta\right\}\right) \leq \exp(-\lambda x + \frac{\lambda^2}{2}\beta) + 2\delta. \quad (8)$$

**Lemma A.4** (Zhang (2005)). *Let $z_1, ..., z_n$ be a sequence of randoms variables such that $z_k$ may depend the previous variables $z_1, ..., z_{k-1}$ for all $k = 1, ..., n$. Consider a sequence of functionals $\xi_k(z_1, ..., z_k)$, $k = 1, ..., n$. Let $\sigma_n^2 = \sum_{k=1}^{n} \mathbb{E}_{z_k}[(\xi_k - \mathbb{E}_{z_k}[\xi_k])^2]$ be the conditional variance. Assume $|\xi_k - \mathbb{E}_{z_k}[\xi_k]| \leq b$ for each $k$. Let $\rho \in (0, 1]$ and $\delta \in (0, 1)$. With probability at least $1 - \delta$ we have*

$$\sum_{k=1}^{n} \xi_k - \sum_{k=1}^{n} \mathbb{E}_{z_k}[\xi_k] \leq \frac{\rho\sigma_n^2}{b} + \frac{b\log\frac{1}{\delta}}{\rho}. \quad (9)$$

**Lemma A.5** (Cutkosky & Mehta (2020)). *For any vector $\mathbf{g} \in \mathbb{R}^d$, $\langle \mathbf{g}/\|\mathbf{g}\|_2, \nabla L_S(\mathbf{w})\rangle \geq \frac{\|\nabla L_S(\mathbf{w})\|_2}{3} - \frac{8\|\mathbf{g} - L_S(\mathbf{w})\|_2}{3}$.*

**Lemma A.6** (Madden et al. (2020)). *If $X \sim subW(\theta, K)$, then $\mathbb{E}[|X^p|] \leq 2\Gamma(p\theta + 1)K^p \ \forall p > 0$. In particular, $\mathbb{E}[X^2] \leq 2\Gamma(2\theta + 1)K^2$.*

**Lemma A.7** (Bakhshizadeh et al. (2023)). *Suppose $X_1, ..., X_m \overset{d}{=} X$ are independent and identically distributed random variables whose right tails are captured by an increasing and continuous function $I : \mathbb{R} \to \mathbb{R}^{\geq 0}$ with the property $I(x) = \mathbb{O}(x)$ as $x \to \infty$. Let $X^L = X\mathbb{I}(X \leq L)$, $S_m = \sum_{i=1}^{m} X_i$ and $Z^L := X^L - \mathbb{E}[X]$. Define $x_{\max} := \sup\{x \geq 0 : x \leq \eta v(mx, \eta)\frac{I(mx)}{mx}\}$, then*

$$\mathbb{P}(S_m - \mathbb{E}[S_m] > mx) \leq \begin{cases} \exp(-c_x\eta I(mx)) + m\exp(-I(mx)), & \text{if } x \geq x_{\max}, \\ \\ \exp(-\frac{mx^2}{2v(mx_{\max}, \eta)}) + m\exp(-\frac{mx_{\max}^2(\eta)}{\eta v(mx_{\max}, \eta)}), & \text{if } 0 \leq x \leq x_{\max}, \end{cases} \quad (10)$$

*where $c_x = 1 - \frac{\eta v(mx, \eta)I(mx)}{2mx^2}$ and $v(L, \eta) = \mathbb{E}[(Z^L)^2\mathbb{I}(Z^L \leq 0) + (Z^L)^2\exp(\eta\frac{I(L)}{L}Z^L)\mathbb{I}(Z^L > 0)]$, $\forall \beta \in (0, 1]$.*

**Lemma A.8** (Bakhshizadeh et al. (2023))**.** *Consider the same settings as the ones in Lemma A.7. Assume $\mathbb{E}[X_i] = 0$, then $\forall t \geq 0$ we have*

$$\mathbb{P}(S_m > mt) \leq \exp(-\frac{mt^2}{2v(mt, \eta)}) + \exp(-\eta \max\{c_t, \frac{1}{2}\}I(mt)) + m\exp(-I(mt)). \quad (11)$$

**Lemma A.9** (**Ahlswede-Winter Inequality**)**.** *Let $Y$ be a random, symmetric, positive semi-definite dd matrix such that $\|\mathbb{E}[Y]\|_2 \leq 1$. Suppose $\|Y\|_2 \leq R$ for some fixed scalar $R \geq 1$. Let $Y_1, ..., Y_m$ be independent copies of $Y$ (i.e., independently sampled matrix with the same distribution as $Y$). For any $\mu \in (0, 1)$, we have*

$$\mathbb{P}(\|\frac{1}{m}\sum_{i=1}^{m} Y_i - \mathbb{E}[Y_i]\|_2 > \mu) \leq 2d \cdot \exp(-m\mu^2/4R).$$

### A.3 NOTATIONS

Table 4: Summary of notations

| Definition of Notations | |
| --- | --- |
| $\mathbf{w}$ | the model parameter |
| $d$ | the dimension of model parameters |
| $z$ | the training sample |
| $n$ | the sample size |
| $B$ | the batch sample size |
| $\ell$ | the loss function |
| $D\ D'$ | the neighboring datasets |
| $\epsilon_{\text{dp}}$ | the privacy budget for differential privacy |
| $\epsilon_{\text{tr}}$ | the privacy budget for preserving traces |
| $\sigma_{\text{dp}}$ | the noise multiplier for differential privacy |
| $\sigma_{\text{tr}}$ | the noise multiplier for preserving traces |
| $V_k$ | $k$-dimensional the random projection vector |
| $K$ | the variance-related positive constant |
| $\nabla L(\mathbf{w}_t)$ | $k$-dimensional the random projection vector |
| $T$ | the iterations of training |
| $\eta_t$ | the learning rate in $t$ iteration |
| $\theta$ | the heavy tail index |
| $p$ | the ratio of heavy tail |
| $\lambda_{t,i}^{\text{tr}}$ | the empirical trace of the sample |
| $\hat{\lambda}_t^{\text{tr}}$ | the population trace for dividing heavy tail or light body |

## B  CONVERGENCE OF HEAVY-TAILED DPSGD

**Theorem B.1** (**Convergence of Heavy-tailed DPSGD**). *Under Assumptions 3.1 and 3.2, let $\mathbf{w}_t$ be the iterate produced by Algorithm DPSGD with $T = \mathbb{O}(\frac{n\epsilon}{\sqrt{d\log(1/\delta)}})$, $T \geq 1$, and $\eta_t = \frac{1}{\sqrt{T}}$. Define*

$\hat{\sigma}_{\mathrm{dp}}^2 := m_2 \frac{Tdc^2B^2\log(1/\delta)}{n^2\epsilon^2}$. *If $\theta = \frac{1}{2}$ and $K \leq \hat{\sigma}_{\mathrm{dp}}$, then $c = \max\left(4K\log^\theta(\sqrt{T}), \frac{19K\log^{\frac{1}{2}}(1/\delta)}{12}\right)$. If $\theta = \frac{1}{2}$ and $K \geq \hat{\sigma}_{\mathrm{dp}}$, then $c = \max\left(4K\log^\theta(\sqrt{T}), 39K\log^{\frac{1}{2}}(2/\delta)\right)$. If $\theta > \frac{1}{2}$, then $c = \max\left(4K\log^\theta(\sqrt{T}), 20K\log^\theta(2/\delta)\right)$. For any $\delta \in (0,1)$, with probability $1 - \delta$, we have*

$$\frac{1}{T}\sum_{t=1}^{T}\min\left\{\|\nabla L_S(\mathbf{w}_t)\|_2, \|\nabla L_S(\mathbf{w}_t)\|_2^2\right\} \leq \mathbb{O}\left(\frac{d^{\frac{1}{4}}\log^{\frac{5}{4}}(T/\delta)\hat{\log}(T/\delta)\log^{2\theta}(\sqrt{T})}{(n\epsilon)^{\frac{1}{2}}}\right),$$

*where $\hat{\log}(T/\delta) := \log^{\max(0,\theta-1)}(T/\delta)$.*

*Proof.* We consider two cases: $\nabla L_S(\mathbf{w}_t) \leq c/2$ and $\nabla L_S(\mathbf{w}_t) \geq c/2$. To simplify notation, we omit the subscript of privacy parameters throughout, such as $\epsilon_{\mathrm{dp}}$.

We first consider the case $\nabla L_S(\mathbf{w}_t) \leq c/2$.

$$L_S(\mathbf{w}_{t+1}) - L_S(\mathbf{w}_t) \leq \langle \mathbf{w}_{t+1} - \mathbf{w}_t, \nabla L_S(\mathbf{w}_t)\rangle + \frac{1}{2}\beta\|\mathbf{w}_{t+1} - \mathbf{w}_t\|^2 \tag{12}$$

$$\leq -\eta_t\langle\overline{\mathbf{g}}_t + \zeta_t, \nabla L_S(\mathbf{w}_t)\rangle + \frac{1}{2}\beta\eta_t^2\|\overline{\mathbf{g}}_t + \zeta_t\|^2$$

$$= -\eta_t\langle\overline{\mathbf{g}}_t - \mathbb{E}_t[\overline{\mathbf{g}}_t] + \mathbb{E}_t[\overline{\mathbf{g}}_t] - \nabla L_S(\mathbf{w}_t), \nabla L_S(\mathbf{w}_t)\rangle - \eta_t\langle\zeta_t, \nabla L_S(\mathbf{w}_t)\rangle$$

$$- \eta_t\|\nabla L_S(\mathbf{w}_t)\|^2 + \frac{1}{2}\beta\eta_t^2\|\overline{\mathbf{g}}_t\|^2 + \frac{1}{2}\beta\eta_t^2\|\zeta_t\|^2 + \beta\eta_t^2\langle\overline{\mathbf{g}}_t, \zeta_t\rangle$$

$$= -\eta_t\langle\overline{\mathbf{g}}_t - \mathbb{E}_t[\overline{\mathbf{g}}_t], \nabla L_S(\mathbf{w}_t)\rangle - \eta_t\langle\mathbb{E}_t[\overline{\mathbf{g}}_t] - \nabla L_S(\mathbf{w}_t), \nabla L_S(\mathbf{w}_t)\rangle - \eta_t\langle\zeta_t, \nabla L_S(\mathbf{w}_t)\rangle$$

$$- \eta_t\|\nabla L_S(\mathbf{w}_t)\|^2 + \frac{1}{2}\beta\eta_t^2\|\overline{\mathbf{g}}_t\|^2 + \frac{1}{2}\beta\eta_t^2\|\zeta_t\|^2 + \beta\eta_t^2\langle\overline{\mathbf{g}}_t, \zeta_t\rangle$$

Considering all $T$ iterations, we get

$$\sum_{t=1}^{T}\eta_t\|\nabla L_S(\mathbf{w}_t)\|^2 \leq L_S(\mathbf{w}_1) - L_S(\mathbf{w}_S) + \sum_{t=1}^{T}\frac{1}{2}\beta\eta_t^2c^2 + \underbrace{\sum_{t=1}^{T}\frac{1}{2}\beta\eta_t^2\|\zeta_t\|^2}_{\text{Eq.1}} + \underbrace{\sum_{t=1}^{T}\beta\eta_t^2\langle\overline{\mathbf{g}}_t, \zeta_t\rangle}_{\text{Eq.2}}$$

$$\underbrace{- \sum_{t=1}^{T}\eta_t\langle\zeta_t, \nabla L_S(\mathbf{w}_t)\rangle}_{\text{Eq.3}} \underbrace{- \sum_{t=1}^{T}\eta_t\langle\overline{\mathbf{g}}_t - \mathbb{E}_t[\overline{\mathbf{g}}_t], \nabla L_S(\mathbf{w}_t)\rangle}_{\text{Eq.4}} \underbrace{- \sum_{t=1}^{T}\eta_t\langle\mathbb{E}_t[\overline{\mathbf{g}}_t] - \nabla L_S(\mathbf{w}_t), \nabla L_S(\mathbf{w}_t)\rangle}_{\text{Eq.5}}$$

$$\tag{13}$$

For Eq.1, Eq.2 and Eq.3, since $\zeta_t \sim \mathbb{N}(0, c\sigma_{\mathrm{dp}}\mathbb{I}_d)$, according to sub-Gaussian properties and Lemma A.2, with probability at least $1 - \delta$, we have

$$\sum_{t=1}^{T}\frac{1}{2}\beta\eta_t^2\|\zeta_t\|^2 \leq 2\beta K^2 e\log(2/\delta)\sum_{t=1}^{T}\eta_t^2$$

$$\leq 2\beta m_2 ed\frac{Tc^2B^2\log^2(2/\delta)}{n^2\epsilon^2}\sum_{t=1}^{T}\eta_t^2. \tag{14}$$

Also, with probability at least $1 - \delta$, we get

$$\sum_{t=1}^{T} \beta \eta_t^2 \langle \overline{\mathbf{g}}_t, \zeta_t \rangle \leq \sum_{t=1}^{T} \beta \eta_t^2 \|\overline{\mathbf{g}}_t\| \|\zeta_t\|$$

$$\leq \sum_{t=1}^{T} 2\beta c K \sqrt{e} \log^{\frac{1}{2}}(2/\delta) \eta_t^2$$

$$\leq 2\beta \sqrt{em_2 T d} \frac{c^2 B \log(2/\delta)}{n\epsilon} \sum_{t=1}^{T} \eta_t^2. \tag{15}$$

Due to $\nabla L_S(\mathbf{w}_t) \leq c/2$, for the term $-\sum_{t=1}^{T} \eta_t \langle \zeta_t, \nabla L_S(\mathbf{w}_t) \rangle$, with probability at least $1 - \delta$, we have

$$-\sum_{t=1}^{T} \eta_t \langle \zeta_t, \nabla L_S(\mathbf{w}_t) \rangle \leq \sum_{t=1}^{T} \eta_t \|\zeta_t\| \|\nabla L_S(\mathbf{w}_t)\|$$

$$\leq \sum_{t=1}^{T} 2cK\sqrt{e} \log^{\frac{1}{2}}(2/\delta) \eta_t$$

$$\leq 2\sqrt{em_2 T d} \frac{c^2 B \log(2/\delta)}{n\epsilon} \sum_{t=1}^{T} \eta_t. \tag{16}$$

Since $\mathbb{E}_t[-\eta_t \langle \overline{\mathbf{g}}_t - \mathbb{E}_t[\overline{\mathbf{g}}_t], \nabla L_S(\mathbf{w}_t) \rangle] = 0$, the sequence $(-\eta_t \langle \overline{\mathbf{g}}_t - \mathbb{E}_t[\overline{\mathbf{g}}_t], \nabla L_S(\mathbf{w}_t) \rangle, t \in \mathbb{N})$ is a martingale difference sequence. Applying Lemma A.4, we define $\xi_t = -\eta_t \langle \overline{\mathbf{g}}_t - \mathbb{E}_t[\overline{\mathbf{g}}_t], \nabla L_S(\mathbf{w}_t) \rangle$ and have

$$|\xi_t| \leq \eta_t (\|\overline{\mathbf{g}}_t\|_2 + \|\mathbb{E}_t[\overline{\mathbf{g}}_t]\|_2) \|\nabla L_S(\mathbf{w}_t)\|_2 \leq \eta_t c^2. \tag{17}$$

Applying $\mathbb{E}_t[(\xi_t - \mathbb{E}_t \xi_t)^2] \leq \mathbb{E}_t[\xi_t^2]$, we have

$$\sum_{t=1}^{T} \mathbb{E}_t[(\xi_t - \mathbb{E}_t \xi_t)^2] \leq \sum_{t=1}^{T} \eta_t^2 \mathbb{E}_t[\|\overline{\mathbf{g}}_t - \mathbb{E}_t[\overline{\mathbf{g}}_t]\|_2^2 \|\nabla L_S(\mathbf{w}_t)\|_2^2]$$

$$\leq 4c^2 \sum_{t=1}^{T} \eta_t^2 \|\nabla L_S(\mathbf{w}_t)\|_2^2. \tag{18}$$

Then, with probability $1 - \delta$, we obtain

$$\sum_{t=1}^{T} \xi_t \leq \frac{\rho 4c^2 \sum_{t=1}^{T} \eta_t^2 \|\nabla L_S(\mathbf{w}_t)\|_2^2}{\eta_t c^2} + \frac{\eta_t c^2 \log(1/\delta)}{\rho}. \tag{19}$$

Next, to bound term Eq.5, we have

$$\sum_{t=1}^{T} \eta_t \langle \mathbb{E}_t[\overline{\mathbf{g}}_t] - \nabla L_S(\mathbf{w}_t), \nabla L_S(\mathbf{w}_t) \rangle \leq \frac{1}{2} \sum_{t=1}^{T} \eta_t \|\mathbb{E}_t[\overline{\mathbf{g}}_t] - \nabla L_S(\mathbf{w}_t)\|_2^2 + \frac{1}{2} \sum_{t=1}^{T} \eta_t \|\nabla L_S(\mathbf{w}_t)\|_2^2.$$

Setting $a_t = \mathbb{I}_{\|\mathbf{g}_t\|_2 > c}$ and $b_t = \mathbb{I}_{\|\mathbf{g}_t - \nabla L_S(\mathbf{w}_t)\|_2 > \frac{c}{2}}$, for term $\|\mathbb{E}_t[\overline{\mathbf{g}}_t] - \nabla L_S(\mathbf{w}_t)\|_2$, we have

$$\|\mathbb{E}_t[\overline{\mathbf{g}}_t] - \nabla L_S(\mathbf{w}_t)\|_2 = \|\mathbb{E}_t[(\overline{\mathbf{g}}_t - \mathbf{g}_t)a_t]\|_2$$

$$= \|\mathbb{E}_t[(\mathbf{g}_t(\frac{c}{\|\mathbf{g}_t\|_2} - 1)a_t]\|_2$$

$$\leq \mathbb{E}_t[\|(\mathbf{g}_t(\frac{c}{\|\mathbf{g}_t\|_2} - 1)a_t\|_2]$$

$$\leq \mathbb{E}_t[|\|\mathbf{g}_t\|_2 - c|a_t]$$

$$\leq \mathbb{E}_t[|\|\mathbf{g}_t\|_2 - \|\nabla L_S(\mathbf{w}_t)\|_2|a_t]$$

$$\leq \mathbb{E}_t[|\|\mathbf{g}_t - \nabla L_S(\mathbf{w}_t)\|_2|a_t]$$

$$\leq \mathbb{E}_t[|\|\mathbf{g}_t - \nabla L_S(\mathbf{w}_t)\|_2|b_t]$$

$$\leq \sqrt{\mathbb{E}_t[\|\mathbf{g}_t - \nabla L_S(\mathbf{w}_t)\|_2^2]\mathbb{E}_t b_t^2}. \tag{20}$$

Applying Lemma A.6, we get $\mathbb{E}_t[\|\mathbf{g}_t - \nabla L_S(\mathbf{w}_t)\|_2^2] \le 2K^2\Gamma(2\theta + 1)$. Then, for term $\mathbb{E}_t b_t^2$, with sub-Weibull properties and probability $1 - \delta$ we have

$$\mathbb{E}_t b_t^2 = \mathbb{P}(\|\mathbf{g}_t - \nabla L_S(\mathbf{w}_t)\|_2 > \frac{c}{2}) \le 2\exp(-(\frac{c}{4K})^{\frac{1}{\theta}}) \tag{21}$$

So, we get formula.(20) as

$$\sqrt{\mathbb{E}_t[\|\mathbf{g}_t - \nabla L_S(\mathbf{w}_t)\|_2^2]\mathbb{E}_t b_t^2} \le 2\sqrt{K^2\Gamma(2\theta + 1)\exp(-(\frac{c}{4K})^{\frac{1}{\theta}})}. \tag{22}$$

Thus, for Eq.5, with probability $1 - T\delta$ we finally obtain

$$\sum_{t=1}^{T} \eta_t \langle \mathbb{E}_t[\overline{\mathbf{g}}_t] - \nabla L_S(\mathbf{w}_t), \nabla L_S(\mathbf{w}_t) \rangle$$

$$\le 2K^2\Gamma(2\theta + 1)\sum_{t=1}^{T} \eta_t\exp(-(\frac{c}{4K})^{\frac{1}{\theta}}) + \frac{1}{2}\sum_{t=1}^{T} \eta_t\|\nabla L_S(\mathbf{w}_t)\|_2^2. \tag{23}$$

Combining Eq.1-5 with the inequality (10), with probability $1 - 4\delta - T\delta$, we have

$$\sum_{t=1}^{T} \eta_t\|\nabla L_S(\mathbf{w}_t)\|_2^2 \le L_S(\mathbf{w}_1) - L_S(\mathbf{w}_S) + \sum_{t=1}^{T} \frac{1}{2}\beta\eta_t^2 c^2 + 2\beta m_2 ed\frac{Tc^2 B^2 \log^2(2/\delta)}{n^2\epsilon^2}\sum_{t=1}^{T} \eta_t^2$$

$$+ 2\beta\sqrt{em_2 Td}\frac{c^2 B\log(2/\delta)}{n\epsilon}\sum_{t=1}^{T} \eta_t^2 + 2\sqrt{em_2 Td}\frac{c^2 B\log(2/\delta)}{n\epsilon}\sum_{t=1}^{T} \eta_t + \frac{\eta_t c^2 \log(1/\delta)}{\rho}$$

$$+ \frac{4\rho c^2 \sum_{t=1}^{T} \eta_t^2\|\nabla L_S(\mathbf{w}_t)\|_2^2}{\eta_t c^2} + 2K^2\Gamma(2\theta + 1)\exp(-(\frac{c}{4K})^{\frac{1}{\theta}})\sum_{t=1}^{T} \eta_t + \frac{1}{2}\sum_{t=1}^{T} \eta_t\|\nabla L_S(\mathbf{w}_t)\|_2^2. \tag{24}$$

Setting $\rho = \frac{1}{16}$, $T = \mathbb{O}(\frac{n\epsilon}{\sqrt{d\log(1/\delta)}})$ and $\eta_t = \frac{1}{\sqrt{T}}$, we have

$$\frac{1}{4}\sum_{t=1}^{T} \eta_t\|\nabla L_S(\mathbf{w}_t)\|_2^2 \le L_S(\mathbf{w}_1) - L_S(\mathbf{w}_S) + \frac{1}{2}\beta c^2 + 2\beta m_2 e\frac{d^{\frac{1}{2}}c^2 B^2 \log^{\frac{3}{2}}(2/\delta)}{n\epsilon}$$

$$+ 2\beta\sqrt{em_2}\frac{d^{\frac{1}{4}}c^2 B\log^{\frac{1}{2}}(2/\delta)}{\sqrt{n\epsilon}} + 2\sqrt{em_2}c^2 B\log^{\frac{1}{2}}(2/\delta) + \frac{16d^{\frac{1}{4}}c^2 \log^{\frac{5}{4}}(1/\delta)}{\sqrt{n\epsilon}}$$

$$+ \underbrace{2K^2\Gamma(2\theta + 1)\exp(-(\frac{c}{4K})^{\frac{1}{\theta}})\sqrt{T}}_{\text{Eq.6}}. \tag{25}$$

Then, we pay attention to term Eq.6. If $c \to 0$, then $\exp(-(\frac{c}{4K})^{\frac{1}{\theta}}) \to 1$ and $\sqrt{T}$ will dominate term Eq.6. We know that in classical DPSGD, a small $c$ is regarded as the clipping threshold guide, which will cause the variance term Eq.6 to dominate the entire bound. For this, we will provide guidance on the clipping values of DPSGD under the heavy-tailed assumption.

Let $\exp(-(\frac{c}{4K})^{\frac{1}{\theta}}) \le \frac{1}{\sqrt{T}}$, then we have $c \ge 4K\log^{\theta}(\sqrt{T})$. So, we obtain

$$\sum_{t=1}^{T} \eta_t\|\nabla L_S(\mathbf{w}_t)\|_2^2 \le 4(L_S(\mathbf{w}_1) - L_S(\mathbf{w}_S)) + 2\beta c^2 + 8\beta m_2 e\frac{d^{\frac{1}{2}}c^2 B^2 \log^{\frac{3}{2}}(2/\delta)}{n\epsilon}$$

$$+ 8\beta\sqrt{em_2}\frac{d^{\frac{1}{4}}c^2 B\log^{\frac{1}{2}}(2/\delta)}{\sqrt{n\epsilon}} + 8\sqrt{em_2}c^2 B\log^{\frac{1}{2}}(2/\delta) + \frac{64d^{\frac{1}{4}}c^2 \log^{\frac{5}{4}}(1/\delta)}{\sqrt{n\epsilon}} + 8K^2\Gamma(2\theta + 1). \tag{26}$$

Multiplying $\frac{1}{\sqrt{T}}$ on both sides, we get

$$\frac{1}{\sqrt{T}}\sum_{t=1}^{T}\eta_t\|\nabla L_S(\mathbf{w}_t)\|_2^2 \leq \frac{1}{\sqrt{T}}\left(4(L_S(\mathbf{w}_1)-L_S(\mathbf{w}_S)) + 2\beta c^2 + 8\beta m_2 e\frac{d^{\frac{1}{2}}c^2 B^2 \log^{\frac{3}{2}}(2/\delta)}{n\epsilon}\right.$$

$$\left. +8\beta\sqrt{em_2}\frac{d^{\frac{1}{4}}c^2 B \log^{\frac{1}{2}}(2/\delta)}{\sqrt{n\epsilon}} + 8\sqrt{em_2}c^2 B \log^{\frac{1}{2}}(2/\delta) + \frac{64d^{\frac{1}{4}}c^2 \log^{\frac{5}{4}}(1/\delta)}{\sqrt{n\epsilon}} + 8K^2\Gamma(2\theta+1)\right). \tag{27}$$

Taking $c = 4K\log^{\theta}(\sqrt{T})$, due to $T \geq 1$, we achieve

$$\frac{1}{\sqrt{T}}\sum_{t=1}^{T}\eta_t\|\nabla L_S(\mathbf{w}_t)\|_2^2 \leq \frac{4(L_S(\mathbf{w}_1)-L_S(\mathbf{w}_S))}{\sqrt{T}} + \frac{8K^2\Gamma(2\theta+1)}{\sqrt{T}}$$

$$+ \frac{16K^2\log^{2\theta}(\sqrt{T})\log(2/\delta)}{\sqrt{T}}\left(2\beta + 8\beta m_2 e\frac{d^{\frac{1}{2}}B^2\log^{\frac{1}{2}}(2/\delta)}{n\epsilon}\right.$$

$$\left. +8\beta\sqrt{em_2}\frac{d^{\frac{1}{4}}B\log^{-\frac{1}{2}}(2/\delta)}{\sqrt{n\epsilon}} + 8\sqrt{em_2}B\log^{-\frac{1}{2}}(2/\delta) + \frac{64d^{\frac{1}{4}}\log^{\frac{1}{4}}(1/\delta)}{\sqrt{n\epsilon}}\right)$$

$$\leq \mathbb{O}\left(\frac{\log^{2\theta}(\sqrt{T})\log(1/\delta)}{\sqrt{T}} \cdot \frac{d^{\frac{1}{4}}\log^{\frac{1}{4}}(1/\delta)}{\sqrt{n\epsilon}}\right)$$

$$\leq \mathbb{O}\left(\frac{\log^{2\theta}(\sqrt{T})\log(1/\delta)d^{\frac{1}{4}}\log^{\frac{1}{4}}(1/\delta)}{\sqrt{n\epsilon}}\right). \tag{28}$$

Due to $\frac{1}{T}\sum_{t=1}^{T}\|\nabla L_S(\mathbf{w}_t)\|_2^2 \leq \frac{1}{\sqrt{T}}\sum_{t=1}^{T}\eta_t\|\nabla L_S(\mathbf{w}_t)\|_2^2$, we have

$$\frac{1}{T}\sum_{t=1}^{T}\|\nabla L_S(\mathbf{w}_t)\|_2^2 \leq \mathbb{O}\left(\frac{d^{\frac{1}{4}}\log^{2\theta}(\sqrt{T})\log^{\frac{5}{4}}(1/\delta)}{(n\epsilon)^{\frac{1}{2}}}\right), \tag{29}$$

with probability $1 - T\delta - 4\delta$.

By substitution, with probability $1 - \delta$, we get

$$\frac{1}{T}\sum_{t=1}^{T}\|\nabla L_S(\mathbf{w}_t)\|_2^2 \leq \mathbb{O}\left(\frac{d^{\frac{1}{4}}\log^{2\theta}(\sqrt{T})\log^{\frac{5}{4}}(T/\delta)}{(n\epsilon)^{\frac{1}{2}}}\right). \tag{30}$$

Secondly, we consider the case $\nabla L_S(\mathbf{w}_t) \geq c/2$.

$$L_S(\mathbf{w}_{t+1}) - L_S(\mathbf{w}_t) \leq \langle\mathbf{w}_{t+1}-\mathbf{w}_t, \nabla L_S(\mathbf{w}_t)\rangle + \frac{1}{2}\beta\|\mathbf{w}_{t+1}-\mathbf{w}_t\|_2^2$$

$$\leq \underbrace{-\eta_t\langle\overline{\mathbf{g}}_t + \zeta_t, \nabla L_S(\mathbf{w}_t)\rangle}_{\text{Eq.7}} + \underbrace{\frac{1}{2}\beta\eta_t^2\|\overline{\mathbf{g}}_t + \zeta_t\|_2^2}_{\text{Eq.8}} \tag{31}$$

We have discussed term Eq.8 in the above case, so we focus on Eq.7 here. Setting $s_t^+ = \mathbb{I}_{\|\mathbf{g}_t\|_2 \geq c}$ and $s_t^- = \mathbb{I}_{\|\mathbf{g}_t\|_2 \leq c}$.

$$-\eta_t\langle\overline{\mathbf{g}}_t + \zeta_t, \nabla L_S(\mathbf{w}_t)\rangle$$

$$= -\eta_t\langle\frac{c\mathbf{g}_t}{\|\mathbf{g}_t\|_2}s_t^+ + \mathbf{g}_t s_t^-, \nabla L_S(\mathbf{w}_t)\rangle - \eta_t\langle\zeta_t, \nabla L_S(\mathbf{w}_t)\rangle. \tag{32}$$

Applying Lemma A.5 to term $-\eta_t\langle\frac{c\mathbf{g}_t}{\|\mathbf{g}_t\|_2}s_t^+, \nabla L_S(\mathbf{w}_t)\rangle$, we have

$$-\eta_t\langle\frac{c\mathbf{g}_t}{\|\mathbf{g}_t\|_2}s_t^+, \nabla L_S(\mathbf{w}_t)\rangle \leq -\frac{c\eta_t s_t^+\|\nabla L_S(\mathbf{w}_t)\|_2}{3} + \frac{8c\eta_t\|\mathbf{g}_t - \nabla L_S(\mathbf{w}_t)\|_2}{3}$$

$$\leq -\frac{c\eta_t(1-s_t^-)\|\nabla L_S(\mathbf{w}_t)\|_2}{3} + \frac{8c\eta_t\|\mathbf{g}_t - \nabla L_S(\mathbf{w}_t)\|_2}{3}. \tag{33}$$

For term $-\eta_t \langle \mathbf{g}_t s_t^-, \nabla L_S(\mathbf{w}_t) \rangle$, we obtain

$$
\begin{aligned}
-\eta_t \langle \mathbf{g}_t s_t^-, \nabla L_S(\mathbf{w}_t) \rangle &= -\eta_t s_t^- \left( \langle \mathbf{g}_t - \nabla L_S(\mathbf{w}_t), \nabla L_S(\mathbf{w}_t) \rangle + \|\nabla L_S(\mathbf{w}_t)\|_2^2 \right) \\
&\leq -\eta_t s_t^- \left( -\|\mathbf{g}_t - \nabla L_S(\mathbf{w}_t)\|_2 \|\nabla L_S(\mathbf{w}_t)\|_2 + \|\nabla L_S(\mathbf{w}_t)\|_2^2 \right) \\
&\leq \eta_t \|\mathbf{g}_t - \nabla L_S(\mathbf{w}_t)\|_2 \|\nabla L_S(\mathbf{w}_t)\|_2 - \frac{c}{2} \eta_t s_t^- \|\nabla L_S(\mathbf{w}_t)\|_2 \\
&\leq \eta_t \|\mathbf{g}_t - \nabla L_S(\mathbf{w}_t)\|_2 \|\nabla L_S(\mathbf{w}_t)\|_2 - \frac{c}{3} \eta_t s_t^- \|\nabla L_S(\mathbf{w}_t)\|_2.
\end{aligned} \tag{34}
$$

According to Lemma A.1, with probability at least $1 - \delta$, we have

$$
\|\mathbf{g}_t - \nabla L_S(\mathbf{w}_t)\|_2 \leq K \log^\theta(2/\delta), \tag{35}
$$

then we get

$$
-\eta_t \langle \mathbf{g}_t s_t^-, \nabla L_S(\mathbf{w}_t) \rangle \leq K \log^\theta(2/\delta) \|\nabla L_S(\mathbf{w}_t)\|_2 - \frac{c}{3} \eta_t s_t^- \|\nabla L_S(\mathbf{w}_t)\|_2, \tag{36}
$$

and

$$
-\eta_t \langle \frac{c \mathbf{g}_t}{\|\mathbf{g}_t\|_2} s_t^+, \nabla L_S(\mathbf{w}_t) \rangle \leq -\frac{c \eta_t (1 - s_t^-) \|\nabla L_S(\mathbf{w}_t)\|_2}{3} + \frac{8 c \eta_t K \log^\theta(2/\delta)}{3}. \tag{37}
$$

Using Lemma A.2 to term $-\sum_{t=1}^T \eta_t \langle \zeta_t, \nabla L_S(\mathbf{w}_t) \rangle$, with probability at least $1 - \delta$, we have

$$
-\sum_{t=1}^T \eta_t \langle \zeta_t, \nabla L_S(\mathbf{w}_t) \rangle \leq 4\sqrt{e m_2 T d} \frac{cB \log(2/\delta)}{n\epsilon} \sum_{t=1}^T \eta_t \|\nabla L_S(\mathbf{w}_t)\|_2. \tag{38}
$$

So, combining formula.(35), formula.(37) and formula.(38) with term Eq.7, with probability at least $1 - 2\delta - T\delta$, we obtain

$$
\begin{aligned}
&-\sum_{t=1}^T \eta_t \langle \overline{\mathbf{g}}_t + \zeta_t, \nabla L_S(\mathbf{w}_t) \rangle \leq -\sum_{t=1}^T \frac{c\eta_t}{3} \|\nabla L_S(\mathbf{w}_t)\|_2 + \sum_{t=1}^T \frac{8 c \eta_t K \log^\theta(2/\delta)}{3} \\
&+ K \log^\theta(2/\delta) \sum_{t=1}^T \eta_t \|\nabla L_S(\mathbf{w}_t)\|_2 + 4\sqrt{e m_2 T d} \frac{cB \log(2/\delta)}{n\epsilon} \sum_{t=1}^T \eta_t \|\nabla L_S(\mathbf{w}_t)\|_2 \\
&\leq -\sum_{t=1}^T \frac{c\eta_t}{3} \|\nabla L_S(\mathbf{w}_t)\|_2 + \left( \frac{19}{3} K \log^\theta(2/\delta) + 4\sqrt{e m_2 T d} \frac{cB \log(2/\delta)}{n\epsilon} \right) \sum_{t=1}^T \eta_t \|\nabla L_S(\mathbf{w}_t)\|_2.
\end{aligned} \tag{39}
$$

Next, considering all $T$ iterations and term Eq.8 with $\hat{\sigma}_{\text{dp}}^2 := dc^2 \sigma_{\text{dp}}^2 = m_2 \frac{T dc^2 B^2 \log(1/\delta)}{n^2 \epsilon^2}$ and probability $1 - 4\delta - T\delta$, we have

$$
\begin{aligned}
&\left( \frac{c}{3} - \frac{19}{3} K \log^\theta(2/\delta) - 4\sqrt{e} \hat{\sigma}_{\text{dp}} \log^{\frac{1}{2}}(1/\delta) \right) \sum_{t=1}^T \eta_t \|\nabla L_S(\mathbf{w}_t)\|_2 \leq L_S(\mathbf{w}_1) - L_S(\mathbf{w}_S) \\
&+ \left( 2\beta m_2 ed \frac{T c^2 B^2 \log^2(2/\delta)}{n^2 \epsilon^2} + 2\beta \sqrt{e m_2 T d} \frac{c^2 B \log(2/\delta)}{n\epsilon} + \frac{1}{2} \beta c^2 \right) \sum_{t=1}^T \eta_t^2.
\end{aligned} \tag{40}
$$

If $\theta = \frac{1}{2}$ and $K \geq \hat{\sigma}_{\mathrm{dp}}$, let $\frac{c}{3} \geq \frac{39}{3}K\log^{\frac{1}{2}}(2/\delta)$, i.e. $c \geq 39K\log^{\frac{1}{2}}(2/\delta)$, taking $c = 39K\log^{\frac{1}{2}}(2/\delta)$, $T = \mathbb{O}(\frac{n\epsilon}{\sqrt{d\log(1/\delta)}})$ and $\eta_t = \frac{1}{\sqrt{T}}$, we have

$$
\begin{aligned}
\sum_{t=1}^{T}\eta_t\|\nabla L_S(\mathbf{w}_t)\|_2 &\leq \frac{3}{K\log^{\frac{1}{2}}(2/\delta)}(L_S(\mathbf{w}_1) - L_S(\mathbf{w}_S)) \\
&+ \frac{3\sum_{t=1}^{T}\eta_t^2}{K\log^{\frac{1}{2}}(2/\delta)}\left(2\beta m_2 ed\frac{Tc^2B^2\log^2(2/\delta)}{n^2\epsilon^2} + 2\beta\sqrt{em_2Td}\frac{c^2B\log(2/\delta)}{n\epsilon} + \frac{1}{2}\beta c^2\right) \\
&\leq \frac{L_S(\mathbf{w}_1) - L_S(\mathbf{w}_S) + 2\beta e\hat{\sigma}_{\mathrm{dp}}^2\log(2/\delta) + 2\beta c\sqrt{e}\hat{\sigma}_{\mathrm{dp}}\log^{\frac{1}{2}}(2/\delta) + \frac{39^2}{2}\beta K^2\log(2/\delta)}{\frac{1}{3}K\log^{\frac{1}{2}}(2/\delta)} \\
&\leq \frac{3(L_S(\mathbf{w}_1) - L_S(\mathbf{w}_S))}{K\log^{\frac{1}{2}}(2/\delta)} + 6\beta eK\log^{\frac{1}{2}}(2/\delta) + 6\beta\sqrt{e}\log^{\frac{1}{2}}(2/\delta) + 3\beta\frac{(39)^2}{2}K\log^{\frac{1}{2}}(2/\delta).
\end{aligned}
\tag{41}
$$

Thus, with probability $1 - 4\delta - T\delta$, we have

$$
\frac{1}{T}\sum_{t=1}^{T}\|\nabla L_S(\mathbf{w}_t)\|_2 \leq \frac{1}{\sqrt{T}}\sum_{t=1}^{T}\eta_t\|\nabla L_S(\mathbf{w}_t)\|_2 \leq \mathbb{O}\left(\frac{\log^{\frac{1}{2}}(1/\delta)}{\sqrt{T}}\right) = \mathbb{O}\left(\frac{\log^{\frac{1}{2}}(1/\delta)d^{\frac{1}{4}}\log^{\frac{1}{4}}(1/\delta)}{\sqrt{n\epsilon}}\right),
$$

implying that with probability $1 - \delta$, we have

$$
\frac{1}{T}\sum_{t=1}^{T}\|\nabla L_S(\mathbf{w}_t)\|_2 \leq \mathbb{O}\left(\frac{d^{\frac{1}{4}}\log^{\frac{3}{4}}(T/\delta)}{\sqrt{n\epsilon}}\right).
\tag{42}
$$

If $\theta = \frac{1}{2}$ and $K \leq \hat{\sigma}_{\mathrm{dp}}$, that is, $c \geq \frac{19\log^{\frac{1}{2}}(1/\delta)K}{12}$, thus there exists $T = \mathbb{O}(\frac{n\epsilon}{\sqrt{d\log(1/\delta)}})$, $T \geq 1$ and $\eta_t = \frac{1}{\sqrt{T}}$ that we obtain

$$
\begin{aligned}
\sum_{t=1}^{T}\eta_t\|\nabla L_S(\mathbf{w}_t)\|_2 &\leq \frac{1}{\sqrt{e}\hat{\sigma}_{\mathrm{dp}}\log^{\frac{1}{2}}(1/\delta)}(L_S(\mathbf{w}_1) - L_S(\mathbf{w}_S)) \\
&+ \frac{\sum_{t=1}^{T}\eta_t^2}{\sqrt{e}\hat{\sigma}_{\mathrm{dp}}\log^{\frac{1}{2}}(1/\delta)}\left(2\beta m_2 ed\frac{Tc^2B^2\log^2(2/\delta)}{n^2\epsilon^2} + 2\beta\sqrt{em_2Td}\frac{c^2B\log(2/\delta)}{n\epsilon} + \frac{1}{2}\beta c^2\right) \\
&\leq \frac{1}{\sqrt{e}\hat{\sigma}_{\mathrm{dp}}\log^{\frac{1}{2}}(1/\delta)}(L_S(\mathbf{w}_1) - L_S(\mathbf{w}_S)) \\
&+ \frac{\sum_{t=1}^{T}\eta_t^2}{\sqrt{e}\hat{\sigma}_{\mathrm{dp}}\log^{\frac{1}{2}}(1/\delta)}\left(2\beta e\hat{\sigma}_{\mathrm{dp}}^2\log(2/\delta) + 2\beta\sqrt{e}\hat{\sigma}_{\mathrm{dp}}\log^{\frac{1}{2}}(2/\delta) + \frac{27^2}{2}\beta e\hat{\sigma}_{\mathrm{dp}}^2\log(2/\delta)\right) \\
&\leq \frac{L_S(\mathbf{w}_1) - L_S(\mathbf{w}_S)}{K\log^{\frac{1}{2}}(2/\delta)} + 2\beta eK\log^{\frac{1}{2}}(2/\delta) + 2\beta\sqrt{e}\log^{\frac{1}{2}}(2/\delta) + \beta\frac{(27)^2}{2}K\log^{\frac{1}{2}}(2/\delta).
\end{aligned}
\tag{43}
$$

Therefore, with probability $1 - 4\delta - T\delta$, we have

$$
\frac{1}{T}\sum_{t=1}^{T}\|\nabla L_S(\mathbf{w}_t)\|_2 \leq \mathbb{O}\left(\frac{\log^{\frac{1}{2}}(1/\delta)d^{\frac{1}{4}}\log^{\frac{1}{4}}(1/\delta)}{\sqrt{n\epsilon}}\right),
$$

then, with probability $1 - \delta$, we have

$$
\frac{1}{T}\sum_{t=1}^{T}\|\nabla L_S(\mathbf{w}_t)\|_2 \leq \mathbb{O}\left(\frac{d^{\frac{1}{4}}\log^{\frac{3}{4}}(T/\delta)}{\sqrt{n\epsilon}}\right).
\tag{44}
$$

If $\theta > \frac{1}{2}$, then term $\log^\theta(2/\delta)$ dominates the left-hand inequality, i.e. $\frac{19}{3}K\log^\theta(2/\delta) \geq 4\sqrt{e}\hat{\sigma}_{\mathrm{dp}}\log^{\frac{1}{2}}(1/\delta)$. Let $\frac{c}{3} \geq \frac{20}{3}K\log^\theta(2/\delta)$, $T = \mathbb{O}(\frac{n\epsilon}{\sqrt{d\log(1/\delta)}})$ and $\eta_t = \frac{1}{\sqrt{T}}$, we obtain

$$
\sum_{t=1}^{T}\eta_t\|\nabla L_S(\mathbf{w}_t)\|_2 \leq \frac{3}{K\log^\theta(2/\delta)}(L_S(\mathbf{w}_1) - L_S(\mathbf{w}_S))
$$

$$
+ \frac{3\sum_{t=1}^{T}\eta_t^2}{K\log^\theta(2/\delta)}\left(2\beta m_2 ed\frac{Tc^2 B^2\log^2(2/\delta)}{n^2\epsilon^2} + 2\beta\sqrt{em_2 Td}\frac{c^2 B\log(2/\delta)}{n\epsilon} + \frac{1}{2}\beta c^2\right)
$$

$$
\leq \frac{3(L_S(\mathbf{w}_1) - L_S(\mathbf{w}_S))}{K\log^\theta(2/\delta)} + \frac{19^2}{24}\beta K\log^\theta(2/\delta) + 190\beta K\log^\theta(2/\delta) + 3\beta(20)^2 K\log^\theta(2/\delta).
$$

$$(45)$$

Consequently, with probability $1 - \delta$, we have

$$
\frac{1}{T}\sum_{t=1}^{T}\|\nabla L_S(\mathbf{w}_t)\|_2 \leq \mathbb{O}(\frac{\log^\theta(T/\delta)d^{\frac{1}{4}}\log^{\frac{1}{4}}(T/\delta)}{\sqrt{n\epsilon}}).
$$

$$(46)$$

Integrating the above results, when $\nabla L_S(\mathbf{w}_t) \geq c/2$ we have

$$
\frac{1}{T}\sum_{t=1}^{T}\|\nabla L_S(\mathbf{w}_t)\|_2 \leq \mathbb{O}(\frac{d^{\frac{1}{4}}\log^{\theta+\frac{1}{4}}(T/\delta)}{\sqrt{n\epsilon}}),
$$

$$(47)$$

with probability $1 - \delta$ and $\theta \geq \frac{1}{2}$.

To sum up, covering the two cases, we ultimately come to the conclusion with probability $1 - \delta$, $T = \mathbb{O}(\frac{n\epsilon}{\sqrt{d\log(1/\delta)}})$, $T \geq 1$, and $\eta_t = \frac{1}{\sqrt{T}}$

$$
\frac{1}{T}\sum_{t=1}^{T}\min\left\{\|\nabla L_S(\mathbf{w}_t)\|_2, \|\nabla L_S(\mathbf{w}_t)\|_2^2\right\} \leq \mathbb{O}(\frac{d^{\frac{1}{4}}\log^{\theta+\frac{1}{4}}(T/\delta)}{(n\epsilon)^{\frac{1}{2}}}) + \mathbb{O}(\frac{d^{\frac{1}{4}}\log^{2\theta}(\sqrt{T})\log^{\frac{5}{4}}(T/\delta)}{(n\epsilon)^{\frac{1}{2}}})
$$

$$
\leq \mathbb{O}(\frac{d^{\frac{1}{4}}\log^{\frac{5}{4}}(T/\delta)\left(\log^{\theta-1}(T/\delta) + \log^{2\theta}(\sqrt{T})\right)}{(n\epsilon)^{\frac{1}{2}}})
$$

$$
\leq \mathbb{O}(\frac{d^{\frac{1}{4}}\log^{\frac{5}{4}}(T/\delta)\hat{\log}(T/\delta)\log^{2\theta}(\sqrt{T})}{(n\epsilon)^{\frac{1}{2}}}),
$$

$$(48)$$

where $\hat{\log}(T/\delta) = \log^{\max(0,\theta-1)}(T/\delta)$. If $\theta = \frac{1}{2}$ and $K \leq \hat{\sigma}_{\mathrm{dp}}$, then $c = \max\left(4K\log^\theta(\sqrt{T}), \frac{19K\log^{\frac{1}{2}}(1/\delta)}{12}\right)$. If $\theta = \frac{1}{2}$ and $K \geq \hat{\sigma}_{\mathrm{dp}}$, then $c = \max\left(4K\log^\theta(\sqrt{T}), 39K\log^{\frac{1}{2}}(2/\delta)\right)$. If $\theta > \frac{1}{2}$, then $c = \max\left(4K\log^\theta(\sqrt{T}), 20K\log^\theta(2/\delta)\right)$. $\square$

The proof of Theorem 4.1 is completed.

## C    PRIVACY GUARANTEE

We provide the complete privacy guarantee proof of Theorem 5.1 for our differential private mechanism $M'$: Subsample∘TraceSorting (TS)∘GradientPerturbation (GP). The specific proof process is as follows, and our proof comprehensively encompasses mechanism $M'$:

- **TraceSorting**: We prove that TraceSorting is $(\epsilon_{\mathrm{tr}}, \delta_{\mathrm{tr}})$-DP.
- **TraceSorting∘GradientPerturbation**: We prove that based on the results of TraceSorting, with two different clipping threshold, the unified composition of TraceSorting and GradientPerturbation is $(\epsilon_{\mathrm{tr}} + \epsilon_{\mathrm{dp}}, \delta)$-DP, where $\delta = \delta_{\mathrm{tr}} + \delta_{\mathrm{dp}}$.
- **Subsample∘TraceSorting∘GradientPerturbation**: We prove that, under the premise of subsampling, the privacy amplification effect remains valid for our composition mechanism.

**(1)** Firstly, we show the TS with Gaussian noise here is $(\epsilon_{\mathrm{tr}}, \delta_{\mathrm{tr}})$-DP and follow the proof of Report Noisy Argmax (RNA) in Claim 3.9 Dwork et al. (2014) to clarify that.

*Proof.* Our trace sorting is to choose traces ranked from 1 to $pB$. To prove that this process satisfies differential privacy (DP), we need to demonstrate that the method of Report $i$-th Noisy Argmax for any $i \in \mathbb{Z}^+$ and $i \in (0, m]$ is $(\epsilon_{\mathrm{tr}}, \delta_{\mathrm{tr}})$-DP, where $m$ is sample size. Fix the neighboring datasets $D = D' \cup \{a\}$. Let $\lambda$, respectively $\lambda'$, denote the vector of traces when the dataset is $D$, respectively $D'$. We have discussed the default $L_2$ sensitivity is 1 and use two properties:

1. **Monotonicity of Traces.** For all $j \in [m]$, $\lambda_j \geq \lambda'_j$;

2. **Lipschitz Property.** For all $j \in [m]$, $1 + \lambda'_j \geq \lambda_j$.

Fix any $i \in [m]$. We will bound from above and below the ratio of the probabilities that $i$ is selected with $D$ and with $D'$. Fix $r^+_{-i}$, a set from $\mathrm{Gauss}(1/\epsilon_{\mathrm{tr}})^{m-i}$ used for all the noisy traces greater than the $i$-th trace. Defines $r^-_{-i}$, a set from $\mathrm{Gauss}(1/\epsilon_{\mathrm{tr}})^{i-1}$ used for all the noisy traces less than the $i$-th trace. We will argue for each $r_{-i} = r^+_{-i} \cup r^-_{-i}$ independently. We use the notation $\mathbb{P}[i \mid \xi]$ to mean the probability that the output of the Report Noisy Max algorithm is $i$, conditioned on $\xi$.

We first argue that $\mathbb{P}[i \mid D, r^-_{-i}] \leq e^\epsilon_{\mathrm{tr}} \mathbb{P}[i \mid D', r^-_{-i}] + \delta_{\mathrm{tr}}$. Define

$$r^* = \min_{r_i} : \lambda_i + r_i > \lambda_j + r_j \quad \forall j \in \arg(r^-_{-i}).$$

Note that, having fixed $r^-_{-i}$, $i$ will be the output (the $i$-th argmax noisy trace) when the dataset is $D$ if and only if $r_i \geq r^*$. We have, for all $j \in \arg(r^-_{-i})$:

$$\lambda_i + r^* > \lambda_j + r_j$$

$$\Rightarrow (1 + \lambda'_i) + r^* \geq \lambda_i + r^* > \lambda_j + r_j \geq \lambda'_j + r_j$$

$$\Rightarrow \lambda'_i + (r^* + 1) > \lambda'_j + r_j.$$

Thus, if $r_i \geq r^* + 1$, then the $i$-th trace will be the $i$-th maximum on one side when the dataset is $D'$ and the noise vector is $(r_i, r^-_{-i})$. The probabilities below are over the choice of $r_i \sim \mathrm{Gauss}(1/\epsilon_{\mathrm{tr}})$, then with probability $1 - \delta_{\mathrm{tr}}$:

$$\mathbb{P}[r_i \geq 1 + r^*] \geq e^{-\epsilon_{\mathrm{tr}}} \mathbb{P}[r_i \geq r^*] = e^{-\epsilon_{\mathrm{tr}}} \mathbb{P}[i \mid D, r^-_{-i}]$$

$$\Rightarrow \mathbb{P}[i \mid D', r^-_{-i}] \geq \mathbb{P}[r_i \geq 1 + r^*] \geq e^{-\epsilon_{\mathrm{tr}}} \mathbb{P}[r_i \geq r^*] = e^{-\epsilon_{\mathrm{tr}}} \mathbb{P}[i \mid D, r^-_{-i}],$$

which, after multiplying through by $e^\epsilon_{\mathrm{tr}}$ and adding probability $\delta$ for $\mathbb{P}[r^* - r_i \geq 1] \leq \delta_{\mathrm{tr}}$, yields what we wanted to show:

$$\mathbb{P}[i \mid D, r^-_{-i}] \leq e^\epsilon_{\mathrm{tr}} \mathbb{P}[i \mid D', r^-_{-i}] + \delta_{\mathrm{tr}}.$$

Then, we argue that $\mathbb{P}[i \mid D, r^+_{-i}] \leq e^\epsilon_{\mathrm{tr}} \mathbb{P}[i \mid D', r^+_{-i}] + \delta_{\mathrm{tr}}$. Define

$$r^* = \max_{r_i} : \lambda_i + r_i < \lambda_j + r_j \quad \forall j \in \arg(r^+_{-i}).$$

Note that, having fixed $r^+_{-i}$, $i$ will be the output (the $i$-th argmax noisy trace) when the dataset is $D$ if and only if $r_i \leq r^*$. We have, for all $j \in \arg(r^+_{-i})$:

$$\lambda_i + r^* < \lambda_j + r_j$$

$$\Rightarrow \lambda'_i + r^* \leq \lambda_i + r^* < \lambda_j + r_j \leq (\lambda'_j + 1) + r_j$$

$$\Rightarrow \lambda'_i + (r^* - 1) < \lambda'_j + r_j.$$

Thus, if $r_i \leq r^* - 1$, then the $i$-th trace will be the $i$-th maximum on the other side when the dataset is $D'$ and the noise vector is $(r_i, r^+_{-i})$. The probabilities below are over the choice of $r_i \sim \text{Gauss}(1/\epsilon_{\text{tr}})$, with probability $1 - \delta_{\text{tr}}$, and we have:

$$\mathbb{P}[r_i \leq r^* - 1] \geq e^{-\epsilon_{\text{tr}}} \mathbb{P}[r_i \leq r^*] = e^{-\epsilon_{\text{tr}}} \mathbb{P}[i \mid D, r^+_{-i}]$$

$$\Rightarrow \mathbb{P}[i \mid D', r^+_{-i}] \geq \mathbb{P}[r_i \leq r^* - 1] \geq e^{-\epsilon_{\text{tr}}} \mathbb{P}[r_i \leq r^*] = e^{-\epsilon_{\text{tr}}} \mathbb{P}[i \mid D, r^+_{-i}].$$

After multiplying through by $e^{\epsilon}_{\text{tr}}$ and adding probability $\delta_{\text{tr}}$ for $\mathbb{P}[r_i - r^* \geq -1] \leq \delta$, we get:

$$\mathbb{P}[i \mid D, r^+_{-i}] \leq e^{\epsilon}_{\text{tr}} \mathbb{P}[i \mid D', r^+_{-i}] + \delta_{\text{tr}}.$$

Overall, combing the both cases with $\delta_{\text{tr}} = 2\delta_{\text{tr}}$, we have

$$e^{\epsilon_{\text{tr}}}(\mathbb{P}[i \mid D', r^+_{-i}] + \mathbb{P}[i \mid D', r^-_{-i}]) + \delta_{\text{tr}} \geq \mathbb{P}[i \mid D, r^+_{-i}] + \mathbb{P}[i \mid D, r^-_{-i}]$$

$$e^{\epsilon_{\text{tr}}} \mathbb{P}[i \mid D', r_{-i}] + \delta_{\text{tr}} \geq \mathbb{P}[i \mid D, r_{-i}],$$

more precisely, we can explicitly bound $\delta_{\text{tr}}$ to $\mathbb{O}(\frac{1}{pB})$ by refering to Zhu & Wang (2020).

Using the same approach, we can prove that

$$e^{\epsilon_{\text{tr}}} \mathbb{P}[i \mid D, r_{-i}] + +\delta_{\text{tr}} \geq \mathbb{P}[i \mid D', r_{-i}].$$

$\square$

Thus, TraceSorting with Gaussian noise satisfies $(\epsilon_{\text{tr}}, \delta_{\text{tr}})$-DP.

**(2)** Secondly, we prove the unified composition of TraceSorting∘GradientPerturbation is $(\epsilon_{\text{tr}} + \epsilon_{\text{dp}}, \delta)$-DP. Based on the results of TraceSorting, we employ two different clipping thresholds for GradientPerturbation.

*Proof.* We define the clipping threshold vector $c$ for per-sample gradient by TraceSorting, for example, with $B = 3$ and $p = 1/3$, if heavy tailed indicator $\lambda = [1, 0, 0]$ then $c = [c_1, c_2, c_2]$.

$$\mathbb{P}[M(D) = Y] = \mathbb{P}[\text{TraceSorting=index } i \text{ AND GP}|D]$$

$$= \int_{-\infty}^{\infty} \mathbb{P}[i|D, r_{-i}] \cdot \mathbb{P}[\text{GP with heavy tailed samples } i] dr$$

$$= \int_{-\infty}^{\infty} \int_{-\infty}^{\infty} \mathbb{P}[i|D, r_{-i}] \cdot \mathbb{P}[\frac{1}{B}(\sum_{j}^{B \in D} g_j + c_j \zeta_j) = Y|c] dr d\zeta$$

$$= \int_{-\infty}^{\infty} \int_{-\infty}^{\infty} \mathbb{P}[i|D, r_{-i}] \cdot \mathbb{P}[f(D) = Y|c] \cdot \mathbb{P}[\zeta = c_j \zeta_j/B] dr d\zeta = *,$$

where $r \sim \text{Gauss}(1/\epsilon_{\text{tr}})$ and $\zeta \sim \text{Gauss}(1/\epsilon_{\text{dp}})$. We define $f(\cdot) = \text{GradientDiscent}$ and $\Delta f = \|f(D) - f(D')\|_2 = \frac{1}{B}(pBc_1 + (1-p)Bc_2) = pc_1 + (1-p)c_2$. With $1 - (\delta_{\text{tr}} + \delta_{\text{dp}})$, we have

$$* = \int_{-\infty}^{\infty} \int_{-\infty}^{\infty} \exp(\epsilon_{\text{tr}}) \mathbb{P}[i|D', r_{-i}] \cdot \mathbb{P}[\frac{1}{B}(\sum_{j}^{B \in D'} g_j + c_j \zeta_j) = Y|c] dr d\zeta$$

$$= \int_{-\infty}^{\infty} \int_{-\infty}^{\infty} \exp(\epsilon_{\text{tr}}) \mathbb{P}[i|D', r_{-i}] \cdot \mathbb{P}[f(D') + c_j \zeta_j/B = Y + \Delta f|c] dr d\zeta$$

$$= \int_{-\infty}^{\infty} \int_{-\infty}^{\infty} \exp(\epsilon_{\text{tr}}) \mathbb{P}[i|D', r_{-i}] \cdot \mathbb{I}[f(D') = Y] \cdot \mathbb{P}[\zeta = c_j \zeta_j/B - \Delta f|c] dr d\zeta$$

$$\leq \int_{-\infty}^{\infty} \int_{-\infty}^{\infty} \exp(\epsilon_{\text{tr}}) \mathbb{P}[i|D', r_{-i}] \cdot \mathbb{I}[f(D') = Y] \cdot \exp(\epsilon_{\text{dp}}) \mathbb{P}[\zeta = c_j \zeta_j/B|c] dr d\zeta$$

$$\leq \exp(\epsilon_{\text{tr}} + \epsilon_{\text{dp}}) \mathbb{P}[M(D') = Y],$$

where we have taken into account the randomness of $c$ through $r$ with $\lambda$, then the first inequality comes from TraceSorting satisfying DP, and the penultimate inequality is derived from the basic Gaussian-based DP mechanism. Thus, define $\delta = \delta_{tr} + \delta_{dp}$, TraceSorting∘GradientPerturbation is $(\epsilon_{tr} + \epsilon_{dp}, \delta)$-DP. $\qquad\square$

**(3)** Thirdly, we provide the proof that privacy amplification with subsampling still holds with the mechanism $M$: TraceSorting∘GradientPerturbation.

*Proof.* We use $B \subseteq \{1, ..., n\}$ to denote the identities of the $B$-subsampled samples from $D = \{z_1, \ldots, z_n\}$. Note that the randomness of $M'$ includes both the randomness of the random sample $B$ and the random coins of $M$. Let $D_B$ (or $D'_B$) be a subsample from $D$ (or $D'$). Let $Y$ be an arbitrary output range. For convenience, define $q = B/n$.

To show $(q(e^{\epsilon_{tr}+\epsilon_{dp}} - 1), q\delta)$-DP, we have to bound the ratio with $D' = D \cup i$:

$$\frac{\mathbb{P}[M'(D) = Y] - q\delta}{\mathbb{P}[M'(D') = Y]} = \frac{q\mathbb{P}[M(D_B) = Y \mid i \in B] + (1-q)\mathbb{P}[M(D_B) = Y \mid i \notin B] - q\delta}{q\mathbb{P}[M(D'_B) = Y \mid i \in B] + (1-q)\mathbb{P}[M(D'_B) = Y \mid i \notin B]}$$

by $e^{q(e^{\epsilon_{tr}+\epsilon_{dp}}-1)}$. For convenience, define the quantities:

$$C = \mathbb{P}[M(D_B) = Y \mid i \in B]$$
$$C' = \mathbb{P}[M(D'_B) = Y \mid i \in B]$$
$$E = \mathbb{P}[M(D_B) = Y \mid i \notin B] = \mathbb{P}[M(D'_B) = Y \mid i \notin B]$$

We can rewrite the ratio as:

$$\frac{\mathbb{P}[M'(D) = Y] - q\delta}{\mathbb{P}[M'(D') = Y]} = \frac{qC + (1-q)E - q\delta}{qC' + (1-q)E}$$

Now we use the fact that, by $(\epsilon_{tr}+\epsilon_{dp}, \delta)$-DP, $C \le e^{\epsilon_{tr}+\epsilon_{dp}} \min\{C', E\}+\delta$. The rest is a calculation:

$$
\begin{aligned}
qC + (1-q)E - q\delta &\le q(e^{\epsilon_{tr}+\epsilon_{dp}} \min\{C', E\} + \delta) + (1-q)E - q\delta \\
&= q(\min\{C', E\} + (e^{\epsilon_{tr}+\epsilon_{dp}} - 1) \min\{C', E\} + \delta) + (1-q)E - q\delta \\
&\le q(\min\{C', E\} + (e^{\epsilon_{tr}+\epsilon_{dp}} - 1) \min\{C', E\} + \delta) + (1-q)E - q\delta \\
&\le q(C' + (e^{\epsilon_{tr}+\epsilon_{dp}} - 1)(qC' + (1-q)E) + \delta) + (1-q)E - q\delta \\
&\le q(C' + (e^{\epsilon_{tr}+\epsilon_{dp}} - 1)(qC' + (1-q)E) + \delta) + (1-q)E \\
&\le (1 + q(e^{\epsilon_{tr}+\epsilon_{dp}} - 1))(qC' + (1-q)E).
\end{aligned}
$$

Thus, we have:

$$\frac{\mathbb{P}[M'(D) = Y] - q\delta}{\mathbb{P}[M'(D') = Y]} \le q(e^{\epsilon_{tr}+\epsilon_{dp}} - 1) \cdot \frac{\mathbb{P}[M(D) = Y]}{\mathbb{P}[M(D') = Y]},$$

and we can derive the simpler conclusion $(\mathbb{O}(q\epsilon_{tr} + q\epsilon_{dp}), \mathbb{O}(q\delta))$-DP for mechanism $M'$, i.e Subsample∘TraceSorting∘GradientPerturbation is $(\mathbb{O}(q\epsilon_{tr} + q\epsilon_{dp}), \mathbb{O}(\delta))$-DP. Furthermore, according to RenyiDP Mironov (2017) and tCDP Bun et al. (2018), we can calculate the corresponding noise multiplier $\sigma_{tr,dp} = \mathbb{O}(\frac{q\sqrt{T \log(1/\delta)}}{\epsilon})$ with $\epsilon = \epsilon_{tr}, \epsilon_{dp}$ for the composition of iterations in model training. $\qquad\square$

To sum up, Theorem 5.1 is proven.

## D  SUBSPACE SKEWING FOR IDENTIFICATION

**Theorem D.1 (Subspace Skewing for Identification).** *Assume that the empirical second moment matrix $M = V_k V_k^T \in \mathbb{R}^{d \times d}$ with $V_k^T V_k = \mathbb{I}_k$ approximates the population second moment matrix $\hat{M} = \hat{V}_k \hat{V}_k^T = \mathbb{E}_{V_k \sim \mathscr{P}}[V_k V_k^T]$, $\lambda_{t,i}^{tr} = \text{tr}(V_k^T \hat{\mathbf{g}}_t(z_i) \hat{\mathbf{g}}_t^T(z_i) V_k)$ and $\hat{\lambda}_t^{tr} = \text{tr}(\hat{V}_k^T \hat{\mathbf{g}}_t(z_i) \hat{\mathbf{g}}_t^T(z_i) \hat{V}_k)$, for any gradient $\hat{\mathbf{g}}_t(z_i)$ that satisfies $\|\hat{\mathbf{g}}_t(z_i)\|_2 = 1$, $\zeta_t^{tr} \sim \mathbb{N}(0, \sigma_{tr}^2)$, with probability $1 - \delta_m - \delta_{tr}$, we have*

$$|\lambda_{t,i}^{tr} - \hat{\lambda}_t^{tr} + \zeta_t^{tr}| \leq \frac{4 \log(2d/\delta_m)}{k} + \frac{m_2 \sqrt{B} \log^{\frac{1}{2}}(1/\delta_{tr})}{d^{\frac{1}{2}}}.$$

*Proof.* For simplicity, we abbreviate $\hat{\mathbf{g}}_t(z_i)$ as $\hat{\mathbf{g}}_t$. Due to the Fact.1, $V_k^T V_k = \mathbb{I}$ and $\hat{V}_k^T \hat{V}_k = \mathbb{I}$, we omit subscripts of expectation and have

$$\begin{aligned}
|\lambda_{t,i}^{tr} - \hat{\lambda}_t^{tr}| &:= |\text{tr}(V_k^T \hat{\mathbf{g}}_t \hat{\mathbf{g}}_t^T V_k) - \text{tr}(\hat{V}_k^T \hat{\mathbf{g}}_t \hat{\mathbf{g}}_t^T \hat{V}_k)| \\
&= |\|V_k^T \hat{\mathbf{g}}_t\|_2^2 - \|\hat{V}_k^T \hat{\mathbf{g}}_t\|_2^2| \\
&= |\|V_k V_k^T \hat{\mathbf{g}}_t\|_2^2 - \|\hat{V}_k \hat{V}_k^T \hat{\mathbf{g}}_t\|_2^2| \\
&\leq \|V_k V_k^T \hat{\mathbf{g}}_t - \hat{V}_k \hat{V}_k^T \hat{\mathbf{g}}_t\|_2^2 \\
&\leq \|V_k V_k^T - \hat{V}_k \hat{V}_k^T\|_2^2 \|\hat{\mathbf{g}}_t\|_2^2
\end{aligned} \tag{49}$$

To bound $\mathbb{E}\|V_k V_k^T - \hat{V}_k \hat{V}_k^T\|_2^2$, we need to bound the gap between the sum of the random positive semidefinite matrix $M := V_k V_k^T = \frac{1}{k} \sum_{i=1}^k v_i v_i^T$ and the expectation $\hat{M} := \hat{V}_k \hat{V}_k^T = \mathbb{E}[V_k V_k^T]$.

Due to $\|v_j\|_2 = 1$, we can easily get

$$\begin{aligned}
\|M\|_2 = \|\frac{1}{k} \sum_{i=1}^k v_i v_i^T\|_2 &\leq \frac{1}{k} \sum_{i=1}^k \|v_i v_i^T\|_2 \\
&= \sup_{x:\|x\|_2=1} \frac{1}{k} \sum_{i=1}^k x^T v_i v_i^T x \\
&= \sup_{x:\|x\|_2=1} \frac{1}{k} \sum_{i=1}^k \langle x, v_i \rangle \\
&\leq \frac{1}{k} \sum_{i=1}^k \|x\|_2 \|v_i\|_2 \\
&= 1
\end{aligned} \tag{50}$$

Thus, $\|M\|_2 \leq 1$ and $\|\mathbb{E}M\|_2 = \|M \cdot \mathbb{P}(M)\|_2 \leq 1$ because of $\mathbb{P}(M) \leq 1$.

Then, according to Ahlswede-Winter Inequality with $R = 1$ and $m = k$, we have for any $\mu \in (0, 1)$

$$\mathbb{P}(\|M - \hat{M}\|_2 > \mu) \leq 2d \cdot \exp(\frac{-k\mu^2}{4}), \tag{51}$$

where $d$ is dimension of gradients. The inequality shows that the bounded spectral norm of random matrix $\|M\|_2$ concentrates around its expectation with high probability $1 - 2d \cdot \exp(-k\mu^2/4)$.

Since $\|M\|_2 \in [0, 1]$ and $\|\mathbb{E}M\|_2 \in [0, 1]$, $\|M - \hat{M}\|_2$ is always bounded by 1. Therefore, for $\mu \geq 1$, $\|M - \hat{M}\|_2 > u$ holds with probability 0. So that for any $\mu > 0$, we have

$$\mathbb{P}(\|M - \hat{M}\|_2 > 2\sqrt{\frac{\log 2d}{k}}\mu) \leq \exp(-\mu^2). \tag{52}$$

Based on the inequality above, with probability $1 - \delta_m$, we have

$$\|M - \hat{M}\|_2 \leq 2 \frac{\log^{\frac{1}{2}}(2d/\delta_m)}{\sqrt{k}}. \tag{53}$$

Next, considering that we have implicitly normalized the term $\|\hat{\mathbf{g}}_t\|_2^2$ by the threshold 1, the upper bound of $\|\hat{\mathbf{g}}_t\|_2^2$ is 1. As a result, we obtain

$$
\begin{aligned}
|\lambda_{t,i}^{\mathrm{tr}} - \hat{\lambda}_t^{\mathrm{tr}}| &\le \|V_k V_k^T - \hat{V}_k \hat{V}_k^T\|_2^2 \|\hat{\mathbf{g}}_t\|_2^2 \\
&\le \|V_k V_k^T - \hat{V}_k \hat{V}_k^T\|_2^2 \\
&\le \|M - \hat{M}\|_2^2 \\
&\le \frac{4\log\left(2d/\delta_m\right)}{k},
\end{aligned}
\tag{54}
$$

with probability $1 - \delta_m$.

Due to the shared random subspace of per-sample gradient, the exposed trace may pose potential privacy risks. Thus, we add the noise that satisfies differential privacy to the trace $\lambda_{t,i}^{\mathrm{tr}}$, i.e. $\lambda_{t,i}^{\mathrm{tr}} + \zeta_t^{\mathrm{tr}}$. The upper bound of the trace for per-sample gradient is limited to 1, because we normalize per-sample gradient in advance. So, the sensitivity in differential privacy can be regarded as 1, which in fact means $\zeta_t^{\mathrm{tr}} \sim \mathbb{N}(0, \sigma_{\mathrm{tr}}^2 \mathbb{I}_1)$. Then, applying Gaussian properties, with probability $1 - \delta_m - \delta_{\mathrm{tr}}$, we have

$$
\begin{aligned}
|\lambda_{t,i}^{\mathrm{tr}} - \hat{\lambda}_t^{\mathrm{tr}} + \zeta_t^{\mathrm{tr}}| &\le |\lambda_{t,i}^{\mathrm{tr}} - \hat{\lambda}_t^{\mathrm{tr}}| + |\zeta_t^{\mathrm{tr}}| \\
&\le \frac{4\log\left(2d/\delta_m\right)}{k} + \sigma_{\mathrm{tr}} \log^{\frac{1}{2}}\left(2/\delta_{\mathrm{tr}}\right).
\end{aligned}
\tag{55}
$$

Regarding to $\sigma_{\mathrm{tr}} = \frac{m_2\sqrt{TB\log(1/\delta)}}{n\epsilon_{\mathrm{tr}}}$, we take $T$ as $\frac{n\epsilon_{\mathrm{tr}}}{\sqrt{d\log(1/\delta)}}$ to maintain consistency with the context and have

$$
\begin{aligned}
|\lambda_{t,i}^{\mathrm{tr}} - \hat{\lambda}_t^{\mathrm{tr}} + \zeta_t^{\mathrm{tr}}| &\le \frac{4\log\left(2d/\delta_m\right)}{k} + \frac{m_2\sqrt{B}\log^{\frac{3}{4}}\left(1/\delta_{\mathrm{tr}}\right)}{d^{\frac{1}{4}}\sqrt{n\epsilon_{\mathrm{tr}}}} \\
&\le \frac{4\log\left(2d/\delta_m\right)}{k} + \frac{m_2\sqrt{B}\log^{\frac{1}{2}}\left(1/\delta_{\mathrm{tr}}\right)}{d^{\frac{1}{2}}},
\end{aligned}
$$

where the last inequality holds due to $T \ge 1$.

Intuitively, the conclusion tells us that, since $\lambda_{t,i}^{\mathrm{tr}}$ is a constant, the scale $\sigma_{\mathrm{tr}}\mathbb{I}_1$ of noise added is actually small compared to the noise $\sigma_{\mathrm{dp}}\mathbb{I}_d$ added to gradients, where the latter has a tricky dependence on the dimension space $d$. Concretely, comparing the first term $\frac{4\log(2d/\delta_m)}{k}$, we observe that in the second term $\frac{m_2\sqrt{B}\log^{\frac{1}{2}}(1/\delta_{\mathrm{tr}})}{\sqrt{d}}$, the model parameter $d \gg k$, we concerned in private learning and coupled with noise scale, is in the denominator, which is far better than the factor $\log(d)$ in the numerator of the first term. Therefore the term $\frac{4\log\left(2d/\delta_m\right)}{k}$ will dominate the error of subspace skewing, and we can control this part of the error by adopting a larger $k$.

In conclusion, for the per-sample trace, there is a high probability $1 - \delta_m'$, where $\delta_m' = \delta_m + \delta_{\mathrm{tr}}$, that we can accurately identify heavy-tailed samples within a finite and minor error dependent on the factor $\mathbb{O}(\frac{1}{k})$.

$\square$

The proof of Theorem 5.2 is completed.

# E CONVERGENCE OF DISCRIMINATIVE CLIPPING

**Theorem E.1** (**Convergence of Discriminative Clipping**). *Under Assumptions 3.1, 3.2 and 3.3, let $\mathbf{w}_t$ be the iterate produced by Algorithm Discriminative Clipping DPSGD with $T = \mathbb{O}(\frac{n\epsilon}{\sqrt{d\log(1/\delta)}})$, $T \geq 1$ and $\eta_t = \frac{1}{\sqrt{T}}$. Define $\hat{\log}(T/\delta) = \log^{\max(0,\theta-1)}(T/\delta)$, $\hat{\sigma}_{\mathrm{dp}}^2 = m_2 \frac{Tc^2 dB^2 \log(1/\delta)}{n^2\epsilon^2}$, $a = 2$ if $\theta = 1/2$, $a = (4\theta)^{2\theta}e^2$ if $\theta \in (1/2, 1]$ and $a = (2^{2\theta+1}+2)\Gamma(2\theta+1) + \frac{2^{3\theta}\Gamma(3\theta+1)}{3}$ if $\theta > 1$, for any $\delta \in (0,1)$, with probability $1 - \delta$, then we have:*

*(i). **In the heavy tail region** ($c = c_1$):*

$$\frac{1}{T}\sum_{t=1}^T \min\left\{\|\nabla L_S(\mathbf{w}_t)\|_2, \|\nabla L_S(\mathbf{w}_t)\|_2^2\right\} \leq \mathbb{O}\left(\frac{d^{\frac{1}{4}}\log^{\frac{5}{4}}(T/\delta)\hat{\log}(T/\delta)\log^{2\theta}(\sqrt{T})}{(n\epsilon)^{\frac{1}{2}}}\right).$$

*(1) If $\theta = \frac{1}{2}$ and $K \leq \hat{\sigma}_{\mathrm{dp}}$, then $c_1 = \max\left(4K\log^{\frac{1}{2}}(\sqrt{T}), \frac{16aK\log^{\frac{1}{2}}(1/\delta)}{12}\right)$. (2) If $\theta = \frac{1}{2}$ and $K \geq \hat{\sigma}_{\mathrm{dp}}$, then $c_1 = \max\left(4K\log^{\frac{1}{2}}(\sqrt{T}), 33\sqrt{2a}K\log^{\frac{1}{2}}(2/\delta)\right)$. (3) If $\theta > \frac{1}{2}$, then $c_1 = \max\left(4^\theta 2K\log^\theta(\sqrt{T}), 17K\log^\theta(2/\delta)\right)$.*

*(ii). **In the light body region** ($c = c_2$):*

$$\frac{1}{T}\sum_{t=1}^T \min\left\{\|\nabla L_S(\mathbf{w}_t)\|_2, \|\nabla L_S(\mathbf{w}_t)\|_2^2\right\} \leq \mathbb{O}\left(\frac{d^{\frac{1}{4}}\log^{\frac{5}{4}}(T/\delta)\log(\sqrt{T})}{(n\epsilon)^{\frac{1}{2}}}\right).$$

*(1) If $K \leq \hat{\sigma}_{\mathrm{dp}}$, then $c_2 = \max\left(2\sqrt{2a}K\log^{\frac{1}{2}}(\sqrt{T}), \frac{16aK\log^{\frac{1}{2}}(1/\delta)}{12}\right)$. (2) If $K \geq \hat{\sigma}_{\mathrm{dp}}$, then $c_2 = \max\left(2\sqrt{2a}K\log^{\frac{1}{2}}(\sqrt{T}), 33\sqrt{2a}K\log^{\frac{1}{2}}(2/\delta)\right)$.*

*Proof.* We review two cases in Discriminative Clipping DPSGD: $\nabla L_S(\mathbf{w}_t) \leq c/2$ and $\nabla L_S(\mathbf{w}_t) \geq c/2$. To simplify notation, we write $\epsilon_{\mathrm{dp}}$ as $\epsilon$, omitting the subscript throughout.

Firstly, in the case $\nabla L_S(\mathbf{w}_t) \leq c/2$:

$$L_S(\mathbf{w}_{t+1}) - L_S(\mathbf{w}_t) \leq \langle \mathbf{w}_{t+1} - \mathbf{w}_t, \nabla L_S(\mathbf{w}_t)\rangle + \frac{1}{2}\beta\|\mathbf{w}_{t+1} - \mathbf{w}_t\|^2$$

$$\leq -\eta_t\langle \overline{\mathbf{g}}_t - \mathbb{E}_t[\overline{\mathbf{g}}_t], \nabla L_S(\mathbf{w}_t)\rangle - \eta_t\langle \mathbb{E}_t[\overline{\mathbf{g}}_t] - \nabla L_S(\mathbf{w}_t), \nabla L_S(\mathbf{w}_t)\rangle - \eta_t\langle \zeta_t, \nabla L_S(\mathbf{w}_t)\rangle$$

$$- \eta_t\|\nabla L_S(\mathbf{w}_t)\|^2 + \frac{1}{2}\beta\eta_t^2\|\overline{\mathbf{g}}_t\|^2 + \frac{1}{2}\beta\eta_t^2\|\zeta_t\|^2 + \beta\eta_t^2\langle \overline{\mathbf{g}}_t, \zeta_t\rangle$$

Applying the properties of Gaussian tails and Lemma A.2 to $\zeta_t$, Lemma A.4 to term $\sum_{t=1}^T \eta_t\langle\overline{\mathbf{g}}_t - \mathbb{E}_t[\overline{\mathbf{g}}_t], \nabla L_S(\mathbf{w}_t)\rangle$, with probability $1 - 4\delta$, we have

$$\sum_{t=1}^T \eta_t\|\nabla L_S(\mathbf{w}_t)\|_2^2 \leq L_S(\mathbf{w}_1) - L_S(\mathbf{w}_S) + \sum_{t=1}^T \frac{1}{2}\beta\eta_t^2 c^2 + 2\beta m_2 ed\frac{Tc^2 B^2 \log^2(2/\delta)}{n^2\epsilon^2}\sum_{t=1}^T \eta_t^2$$

$$+ 2\beta\sqrt{em_2 Td}\frac{c^2 B\log(2/\delta)}{n\epsilon}\sum_{t=1}^T \eta_t^2 + 2\sqrt{em_2 Td}\frac{c^2 B\log(2/\delta)}{n\epsilon}\sum_{t=1}^T \eta_t + \frac{\eta_t c^2 \log(1/\delta)}{\rho}$$

$$+ \frac{4\rho c^2 \sum_{t=1}^T \eta_t^2 \|\nabla L_S(\mathbf{w}_t)\|_2^2}{\eta_t c^2} - \underbrace{\sum_{t=1}^T \eta_t\langle \mathbb{E}_t[\overline{\mathbf{g}}_t] - \nabla L_S(\mathbf{w}_t), \nabla L_S(\mathbf{w}_t)\rangle}_{\text{Eq.9}}. \tag{56}$$

We will consider a truncated version of term Eq.9 in the following. Similarly,

$$\sum_{t=1}^T \eta_t\langle \mathbb{E}_t[\overline{\mathbf{g}}_t] - \nabla L_S(\mathbf{w}_t), \nabla L_S(\mathbf{w}_t)\rangle \leq \frac{1}{2}\sum_{t=1}^T \eta_t\|\mathbb{E}_t[\overline{\mathbf{g}}_t] - \nabla L_S(\mathbf{w}_t)\|_2^2 + \frac{1}{2}\sum_{t=1}^T \eta_t\|\nabla L_S(\mathbf{w}_t)\|_2^2.$$

For term $\|\mathbb{E}_t[\overline{\mathbf{g}}_t] - \nabla L_S(\mathbf{w}_t)\|_2$, we also define $a_t = \mathbb{I}_{\|\mathbf{g}_t\|_2 > c}$ and $b_t = \mathbb{I}_{\|\mathbf{g}_t - \nabla L_S(\mathbf{w}_t)\|_2 > \frac{c}{2}}$, and have

$$
\begin{aligned}
\|\mathbb{E}_t[\overline{\mathbf{g}}_t] - \nabla L_S(\mathbf{w}_t)\|_2 &= \|\mathbb{E}_t[(\overline{\mathbf{g}}_t - \mathbf{g}_t)a_t]\|_2 \\
&\leq \mathbb{E}_t[\|(\mathbf{g}_t(\frac{c - \|\mathbf{g}_t\|_2}{\|\mathbf{g}_t\|_2})a_t\|_2] \\
&\leq \mathbb{E}_t[\|\|\mathbf{g}_t\|_2 - \|\nabla L_S(\mathbf{w}_t)\|_2|a_t] \\
&\leq \mathbb{E}_t[\|\|\mathbf{g}_t - \nabla L_S(\mathbf{w}_t)\|_2|b_t] \\
&\leq \sqrt{\mathbb{E}_t[\|\mathbf{g}_t - \nabla L_S(\mathbf{w}_t)\|_2^2]\mathbb{E}_t b_t^2}.
\end{aligned}
\tag{57}
$$

Due to $\mathbb{E}[\mathbf{g}_t - \nabla L_S(\mathbf{w}_t)] = 0$, applying Lemma A.7 and A.8 with

$$
m = 1
$$
$$
\sup_{\eta \in (0,1]} \{v(L, \eta)\} = aK^2
$$
$$
x_{\max} = \frac{\eta I(x)}{x} aK^2
$$
$$
c_t \in [\frac{1}{2}, 1]
$$
$$
\eta = \frac{1}{2}.
$$

In the light body region, i.e. $x \geq x_{\max}$, we have

$$
\begin{aligned}
\mathbb{P}(\|\mathbf{g}_t - \nabla L_S(\mathbf{w}_t)\|_2 > x) &\leq \exp(-c_t \eta I(x)) + \exp(-I(x)) \\
&\leq \exp(-\frac{1}{4}I(x)) + \exp(-I(x)) \\
&\leq 2\exp(-\frac{1}{4}I(x)).
\end{aligned}
\tag{58}
$$

Then, in the heavy tail region, i.e. $0 \leq x \leq x_{\max}$, the inequality

$$
\begin{aligned}
\mathbb{P}(\|\mathbf{g}_t - \nabla L_S(\mathbf{w}_t)\|_2 > x) &\leq \exp(-\frac{x^2}{2v(x_{\max}, \eta)}) + m\exp(-\frac{x_{\max}^2(\eta)}{\eta v(x_{\max}, \eta)}) \\
&\leq 2\exp(-\frac{x^2}{2v(x_{\max}, \eta)}) \\
&\leq 2\exp(-\frac{x^2}{2aK^2})
\end{aligned}
\tag{59}
$$

holds.

Therefore, when $0 \leq x \leq x_{\max}$, we have the follow-up truncated conclusions:

If $\theta = \frac{1}{2}$, $\forall \alpha > 0$ and $a = 2$, we have the following inequality with probability at least $1 - \delta$

$$
\|\mathbf{g}_t - \nabla L_S(\mathbf{w}_t)\|_2 \leq 2K \log^{\frac{1}{2}}(2/\delta).
$$

If $\theta \in (\frac{1}{2}, 1]$, let $a = (4\theta)^{2\theta} e^2$, we have the following inequality with probability at least $1 - \delta$

$$
\|\mathbf{g}_t - \nabla L_S(\mathbf{w}_t)\|_2 \leq \sqrt{2}e(4\theta)^\theta K \log^{\frac{1}{2}}(2/\delta).
$$

If $\theta > 1$, let $a = (2^{2\theta+1} + 2)\Gamma(2\theta + 1) + \frac{2^{3\theta}\Gamma(3\theta+1)}{3}$, we have the following inequality with probability at least $1 - \delta$

$$
\|\mathbf{g}_t - \nabla L_S(\mathbf{w}_t)\|_2 \leq \sqrt{2(2^{2\theta+1} + 2)\Gamma(2\theta + 1) + \frac{2^{3\theta}\Gamma(3\theta + 1)}{3}} K \log^{\frac{1}{2}}(2/\delta).
$$

When $x \geq x_{\max}$, let $I(x) = (x/K)^{\frac{1}{\theta}}$, $\forall \theta \in (\frac{1}{2}, 1]$, with probability at least $1 - \delta$, then we have

$$\|\mathbf{g}_t - \nabla L_S(\mathbf{w}_t)\|_2 \leq 4^\theta K \log^\theta(2/\delta).$$

Apply the truncated corollary above, when $0 \leq x \leq x_{\max}$, we have

$$\mathbb{E}_t[\|\mathbf{g}_t - \nabla L_S(\mathbf{w}_t)\|_2] \leq \sqrt{2a}K \qquad (60)$$

and with probability $1 - \delta$,

$$\mathbb{E}_t b_t^2 = \mathbb{P}(\|\mathbf{g}_t - \nabla L_S(\mathbf{w}_t)\|_2 > \frac{c}{2}) \leq 2\exp(-(\frac{c}{2\sqrt{2a}K})^2) \qquad (61)$$

where $a = 2$ if $\theta = 1/2$, $a = (4\theta)^{2\theta}e^2$ if $\theta \in (1/2, 1]$ and $a = (2^{2\theta+1} + 2)\Gamma(2\theta + 1) + \frac{2^{3\theta}\Gamma(3\theta+1)}{3}$ if $\theta > 1$.

When $x \geq x_{\max}$, the inequalities

$$\mathbb{E}_t[\|\mathbf{g}_t - \nabla L_S(\mathbf{w}_t)\|_2] \leq 4^\theta K \qquad (62)$$

and

$$\mathbb{E}_t b_t^2 = \mathbb{P}(\|\mathbf{g}_t - \nabla L_S(\mathbf{w}_t)\|_2 > \frac{c}{2}) \leq 2\exp(-\frac{1}{4}(\frac{c}{2K})^{\frac{1}{\theta}}) \qquad (63)$$

hold with probability $1 - \delta$, where $\theta \geq \frac{1}{2}$.

Thus, with probability $1 - T\delta$, we get

$$\sum_{t=1}^{T} \eta_t \langle \mathbb{E}_t[\overline{\mathbf{g}}_t] - \nabla L_S(\mathbf{w}_t), \nabla L_S(\mathbf{w}_t) \rangle \leq 2aK^2 \sum_{t=1}^{T} \eta_t \exp(-(\frac{c}{2\sqrt{2a}K})^2) + \frac{1}{2}\sum_{t=1}^{T} \eta_t \|\nabla L_S(\mathbf{w}_t)\|_2^2, \qquad (64)$$

when $0 \leq x \leq x_{\max}$.

With probability $1 - T\delta$, we obtain

$$\sum_{t=1}^{T} \eta_t \langle \mathbb{E}_t[\overline{\mathbf{g}}_t] - \nabla L_S(\mathbf{w}_t), \nabla L_S(\mathbf{w}_t) \rangle \leq 4^{2\theta}K^2 \sum_{t=1}^{T} \eta_t \exp(-\frac{1}{4}(\frac{c}{2K})^{\frac{1}{\theta}}) + \frac{1}{2}\sum_{t=1}^{T} \eta_t \|\nabla L_S(\mathbf{w}_t)\|_2^2, \qquad (65)$$

when $x \geq x_{\max}$.

By setting $\rho = \frac{1}{16}$, $T = \mathbb{O}(\frac{n\epsilon}{\sqrt{d\log(1/\delta)}})$ and $\eta_t = \frac{1}{\sqrt{T}}$, with probability $1 - 4\delta - T\delta$, we have

$$\frac{1}{4}\sum_{t=1}^{T} \eta_t \|\nabla L_S(\mathbf{w}_t)\|_2^2 \leq L_S(\mathbf{w}_1) - L_S(\mathbf{w}_S) + \frac{1}{2}\beta c^2 + 2\beta m_2 e \frac{d^{\frac{1}{2}}c^2 B^2 \log^{\frac{3}{2}}(2/\delta)}{n\epsilon}$$

$$+ 2\beta\sqrt{em_2}\frac{d^{\frac{1}{4}}c^2 B \log^{\frac{1}{2}}(2/\delta)}{\sqrt{n\epsilon}} + 2\sqrt{em_2}c^2 B \log^{\frac{1}{2}}(2/\delta) + \frac{16d^{\frac{1}{4}}c^2 \log^{\frac{5}{4}}(1/\delta)}{\sqrt{n\epsilon}}$$

$$+ \text{Eq.10} \begin{cases} 2aK^2 \sum_{t=1}^{T} \eta_t \exp(-(\frac{c}{2\sqrt{2a}K})^2), & \text{if } 0 \leq x \leq x_{\max}, \\ \\ 4^{2\theta}K^2 \sum_{t=1}^{T} \eta_t \exp(-\frac{1}{4}(\frac{c}{2K})^{\frac{1}{\theta}}), & \text{if } x \geq x_{\max}. \end{cases} \qquad (66)$$

Let the term Eq.10 $\leq \frac{1}{\sqrt{T}}$, and we have $c \geq 2\sqrt{2a}K \log^{\frac{1}{2}}(\sqrt{T})$ if $0 \leq x \leq x_{\max}$ and $c \geq 4^\theta 2K \log^\theta(\sqrt{T})$ if $x \geq x_{\max}$.

In the light body region that $0 \leq x \leq x_{\max}$, by taking $c_2 = c = 2\sqrt{2a}K\log^{\frac{1}{2}}(\sqrt{T})$ we achieve

$$
\begin{aligned}
\frac{1}{\sqrt{T}}\sum_{t=1}^{T}\eta_t\|\nabla L_S(\mathbf{w}_t)\|_2^2 \leq{}& \frac{4(L_S(\mathbf{w}_1) - L_S(\mathbf{w}_S))}{\sqrt{T}} + \frac{2aK^2}{\sqrt{T}} \\
&+ \frac{8aK^2\log(\sqrt{T})\log(2/\delta)}{\sqrt{T}}\left(2\beta + 8\beta m_2 e B^2(\frac{d^{\frac{1}{4}}\log^{\frac{1}{4}}(2/\delta)}{\sqrt{n\epsilon}})^2\right. \\
&+ 8\beta\sqrt{em_2}\frac{d^{\frac{1}{4}}B\log^{-\frac{1}{2}}(2/\delta)}{\sqrt{n\epsilon}} + 8\sqrt{em_2}B\log^{-\frac{1}{2}}(2/\delta) + \left.\frac{64d^{\frac{1}{4}}\log^{\frac{1}{4}}(1/\delta)}{\sqrt{n\epsilon}}\right) \\
\leq{}& \mathbb{O}(\frac{\log(\sqrt{T})\log(1/\delta)}{\sqrt{T}} \cdot \frac{d^{\frac{1}{4}}\log^{\frac{1}{4}}(1/\delta)}{\sqrt{n\epsilon}}) \\
\leq{}& \mathbb{O}(\frac{\log(\sqrt{T})d^{\frac{1}{4}}\log^{\frac{5}{4}}(1/\delta)}{\sqrt{n\epsilon}}).
\end{aligned}
\tag{67}
$$

In the heavy tail region that $x \geq x_{\max}$, by taking $c_1 = c = 4^{\theta}2K\log^{\theta}(\sqrt{T})$ we achieve

$$
\begin{aligned}
\frac{1}{\sqrt{T}}\sum_{t=1}^{T}\eta_t\|\nabla L_S(\mathbf{w}_t)\|_2^2 \leq{}& \frac{4(L_S(\mathbf{w}_1) - L_S(\mathbf{w}_S))}{\sqrt{T}} + \frac{2aK^2}{\sqrt{T}} \\
&+ \frac{4^{2\theta+1}\log^{2\theta}(\sqrt{T})\log(2/\delta)}{\sqrt{T}}\left(2\beta + 8\beta m_2 e B^2(\frac{d^{\frac{1}{4}}\log^{\frac{1}{4}}(2/\delta)}{\sqrt{n\epsilon}})^2\right. \\
&+ 8\beta\sqrt{em_2}\frac{d^{\frac{1}{4}}B\log^{-\frac{1}{2}}(2/\delta)}{\sqrt{n\epsilon}} + 8\sqrt{em_2}B\log^{-\frac{1}{2}}(2/\delta) + \left.\frac{64d^{\frac{1}{4}}\log^{\frac{1}{4}}(1/\delta)}{\sqrt{n\epsilon}}\right) \\
\leq{}& \mathbb{O}(\frac{\log^{2\theta}(\sqrt{T})\log(1/\delta)}{\sqrt{T}} \cdot \frac{d^{\frac{1}{4}}\log^{\frac{1}{4}}(1/\delta)}{\sqrt{n\epsilon}}) \\
\leq{}& \mathbb{O}(\frac{\log^{2\theta}(\sqrt{T})d^{\frac{1}{4}}\log^{\frac{5}{4}}(1/\delta)}{\sqrt{n\epsilon}}).
\end{aligned}
\tag{68}
$$

Secondly, we pay extra attention to the bound in the case $\nabla L_S(\mathbf{w}_t) \geq c/2$.

$$
\begin{aligned}
L_S(\mathbf{w}_{t+1}) - L_S(\mathbf{w}_t) &\leq \langle \mathbf{w}_{t+1} - \mathbf{w}_t, \nabla L_S(\mathbf{w}_t)\rangle + \frac{1}{2}\beta\|\mathbf{w}_{t+1} - \mathbf{w}_t\|_2^2 \\
&\leq \underbrace{-\eta_t\langle \overline{\mathbf{g}}_t + \zeta_t, \nabla L_S(\mathbf{w}_t)\rangle}_{\text{Eq.11}} + \frac{1}{2}\beta\eta_t^2\|\overline{\mathbf{g}}_t + \zeta_t\|_2^2.
\end{aligned}
\tag{69}
$$

We revisit term Eq.11 in the case and also set $s_t^+ = \mathbb{I}_{\|\mathbf{g}_t\|_2 \geq c}$ and $s_t^- = \mathbb{I}_{\|\mathbf{g}_t\|_2 \leq c}$.

$$
-\eta_t\langle \overline{\mathbf{g}}_t + \zeta_t, \nabla L_S(\mathbf{w}_t)\rangle = -\eta_t\langle \frac{c\mathbf{g}_t}{\|\mathbf{g}_t\|_2}s_t^+ + \mathbf{g}_t s_t^-, \nabla L_S(\mathbf{w}_t)\rangle - \eta_t\langle \zeta_t, \nabla L_S(\mathbf{w}_t)\rangle.
\tag{70}
$$

For term $-\sum_{t=1}^{T} \eta_t \langle \mathbf{g}_t s_t^-, \nabla L_S(\mathbf{w}_t) \rangle$, we obtain

$$
\begin{aligned}
-\sum_{t=1}^{T} \eta_t \langle \mathbf{g}_t s_t^-, \nabla L_S(\mathbf{w}_t) \rangle &= -\sum_{t=1}^{T} \eta_t s_t^- \left( \langle \mathbf{g}_t - \nabla L_S(\mathbf{w}_t), \nabla L_S(\mathbf{w}_t) \rangle + \|\nabla L_S(\mathbf{w}_t)\|_2^2 \right) \\
&\leq -\sum_{t=1}^{T} \eta_t s_t^- \langle \mathbf{g}_t - \nabla L_S(\mathbf{w}_t), \nabla L_S(\mathbf{w}_t) \rangle - \sum_{t=1}^{T} \eta_t s_t^- \|\nabla L_S(\mathbf{w}_t)\|_2^2 \\
&\leq -\sum_{t=1}^{T} \eta_t s_t^- \langle \mathbf{g}_t - \nabla L_S(\mathbf{w}_t), \nabla L_S(\mathbf{w}_t) \rangle - \frac{c}{2} \sum_{t=1}^{T} \eta_t s_t^- \|\nabla L_S(\mathbf{w}_t)\|_2^2 \\
&\leq \underbrace{-\sum_{t=1}^{T} \eta_t s_t^- \langle \mathbf{g}_t - \nabla L_S(\mathbf{w}_t), \nabla L_S(\mathbf{w}_t) \rangle}_{\text{Eq.12}} - \frac{c}{3} \sum_{t=1}^{T} \eta_t s_t^- \|\nabla L_S(\mathbf{w}_t)\|_2^2.
\end{aligned}
$$

$$(71)$$

Let consider the term Eq.12. Since $\mathbb{E}_t[\eta_t s_t^- \langle \mathbf{g}_t - \nabla L_S(\mathbf{w}_t), \nabla L_S(\mathbf{w}_t) \rangle] = 0$, the sequence $(-\eta_t s_t^- \langle \mathbf{g}_t - \nabla L_S(\mathbf{w}_t), \nabla L_S(\mathbf{w}_t) \rangle, t \in \mathbb{N})$ is a martingale difference sequence. In addition, the term $\mathbf{g}_t - \nabla L_S(\mathbf{w}_t)$ is a $subW(\theta, K)$ random variable, thus we apply sub-Weibull Freedman inequality with Lemma A.3 and concentration inequality with Lemma A.7 and A.8 to bound it.

In Lemma A.3, Define

$$
v(L, \eta) := \mathbb{E}\left[ (X^L - \mathbb{E}[X])^2 \mathbb{I}(X^L \leq \mathbb{E}[X]) \right] + \mathbb{E}\left[ (X^L - \mathbb{E}[X])^2 \exp\left( \eta(X^L - \mathbb{E}[X]) \right) \mathbb{I}(X^L > \mathbb{E}[X]) \right],
$$

and make $\beta = kv(L, \eta)$, then we have $\sup_{\eta \in (0,1]} \{ kv(L, \eta) \} = a \sum_{i=1}^{k} K_i^2$ based on Lemma A.7 and A.8 in Bakhshizadeh et al. (2023) and obtain

$$
\begin{aligned}
\mathbb{P}\left( \bigcup_{k \in \mathbb{N}} \left\{ \sum_{i=1}^{k} \xi_i \geq kx \text{ and } \sum_{i=1}^{k} aK_{i-1}^2 \leq \beta \right\} \right) &\leq \exp(-\lambda kx + \frac{\lambda^2}{2} \beta) \\
&= \exp(-\lambda kx + kv(L, \eta) \frac{\lambda^2}{2}).
\end{aligned}
$$

$$(72)$$

Subsequently, we define the inflection point $x_{\max} := \frac{\eta I(kx)}{kx} a \sum_{i=1}^{k} K_i^2$ and have

1. In the light body region where $x \geq x_{\max}$, we choose $L = kx$ and $\lambda = \frac{\eta I(kx)}{kx}$, that is $\frac{x}{v(kx, \eta)} \geq \frac{x_{\max}}{v(kx, \eta)} = \frac{\eta I(kx)}{kx}$. Then the inequality achieves

$$
\begin{aligned}
\mathbb{P}\left( \bigcup_{k \in \mathbb{N}} \left\{ \sum_{i=1}^{k} \xi_i \geq kx \text{ and } \sum_{i=1}^{k} aK_{i-1}^2 \leq \beta \right\} \right) &\leq \exp(-\eta I(kx) + v(L, \eta) \frac{\eta^2 I^2(kx)}{2kx^2}) \\
&\leq \exp(-\eta I(kx)(1 - v(L, \eta) \frac{\eta I(kx)}{2kx^2})) \\
&\leq \exp(-\eta c_x I(kx)) \\
&\leq \exp(-\frac{1}{2} \eta I(kx)),
\end{aligned}
$$

$$(73)$$

   where $c_x = 1 - \frac{\eta v(kx, \eta) I(kx)}{2kx^2}$ and the last inequality holds due to $c_x \geq \frac{1}{2}$.

2. In the heavy tail region where $x \leq x_{\max}$, we choose $L = kx_{\max}$ and $\lambda = \frac{x}{v(L, \eta)} \leq \frac{x_{\max}}{v(L, \eta)} = \frac{\eta I(L)}{L}$. Then, we get

$$
\begin{aligned}
\mathbb{P}\left( \bigcup_{k \in \mathbb{N}} \left\{ \sum_{i=1}^{k} \xi_i \geq kx \text{ and } \sum_{i=1}^{k} aK_{i-1}^2 \leq \beta \right\} \right) &\leq \exp(-\frac{kx^2}{v(L, \eta)} + \frac{kx^2}{2v(L, \eta)}) \\
&\leq \exp(-\frac{kx^2}{2v(L, \eta)}).
\end{aligned}
$$

$$(74)$$

Implementing the above inferences and propositions with

$$\xi_t = \eta_t \langle \mathbf{g}_t - \nabla L_S(\mathbf{w}_t), \nabla L_S(\mathbf{w}_t) \rangle$$

$$\Lambda := -\sum_{i=1}^{T} \eta_t s_t^- \langle \mathbf{g}_t - \nabla L_S(\mathbf{w}_t), \nabla L_S(\mathbf{w}_t) \rangle$$

$$K_{t-1} = \eta_t K \|\nabla L_S(\mathbf{w}_t)\|_2$$

$$m_t = \eta_t K G$$

$$k = T$$

$$\eta = 1/2$$

If $\theta = \frac{1}{2}$, $\forall \alpha > 0$ and $a = 2$, when $x \le x_{\max}$ we have the following inequality with probability at least $1 - \delta$

$$-\sum_{t=1}^{T} \eta_t s_t^- \langle \mathbf{g}_t - \nabla L_S(\mathbf{w}_t), \nabla L_S(\mathbf{w}_t) \rangle \le \sqrt{2Tv(L,\eta)} \log^{\frac{1}{2}}(1/\delta)$$

$$\le \sqrt{2a \sum_{t=1}^{T} K_t^2} \log^{\frac{1}{2}}(1/\delta)$$

$$\le 2\sqrt{\sum_{t=1}^{T} \eta_t^2 K^2 \|\nabla L_S(\mathbf{w}_t)\|_2^2} \log^{\frac{1}{2}}(1/\delta)$$

$$\le 2KG\sqrt{\sum_{t=1}^{T} \eta_t^2} \log^{\frac{1}{2}}(1/\delta), \tag{75}$$

when $x \ge x_{\max}$, with $I(Tx) = (Tx/\sum_{i=1}^{T} K_i)^2$, we have

$$-\sum_{t=1}^{T} \eta_t s_t^- \langle \mathbf{g}_t - \nabla L_S(\mathbf{w}_t), \nabla L_S(\mathbf{w}_t) \rangle \le 4^{\frac{1}{2}} \frac{1}{T} \sum_{t=1}^{T} K_t \log^{\frac{1}{2}}(1/\delta)$$

$$\le 2\frac{KG}{T} \sum_{t=1}^{T} \eta_t \log^{\frac{1}{2}}(1/\delta). \tag{76}$$

If $\theta \in (\frac{1}{2}, 1]$, let $a = (4\theta)^{2\theta} e^2$, when $x \le x_{\max}$ we have the following inequality with probability at least $1 - \delta$

$$-\sum_{t=1}^{T} \eta_t s_t^- \langle \mathbf{g}_t - \nabla L_S(\mathbf{w}_t), \nabla L_S(\mathbf{w}_t) \rangle \le \sqrt{2a \sum_{t=1}^{T} K_t^2} \log^{\frac{1}{2}}(1/\delta)$$

$$\le \sqrt{2}(4\theta)^{\theta} eKG\sqrt{\sum_{t=1}^{T} \eta_t^2} \log^{\frac{1}{2}}(1/\delta), \tag{77}$$

when $x \ge x_{\max}$, let $I(Tx) = (Tx/\sum_{i=1}^{T} K_i)^{\frac{1}{\theta}}$, $\forall \theta \in (\frac{1}{2}, 1]$, then we have

$$-\sum_{t=1}^{T} \eta_t s_t^- \langle \mathbf{g}_t - \nabla L_S(\mathbf{w}_t), \nabla L_S(\mathbf{w}_t) \rangle \le \frac{4^{\theta}}{T} \sum_{t=1}^{T} K_t \log^{\frac{1}{2}}(1/\delta)$$

$$\le \frac{4^{\theta} KG}{T} \sum_{t=1}^{T} \eta_t \log^{\theta}(1/\delta). \tag{78}$$

If $\theta > 1$, let $a = (2^{2\theta+1} + 2)\Gamma(2\theta + 1) + \frac{2^{3\theta}\Gamma(3\theta+1)}{3}$, when $x \leq x_{\max}$ we have the following inequality with probability at least $1 - 3\delta$

$$-\sum_{t=1}^{T} \eta_t s_t^- \langle \mathbf{g}_t - \nabla L_S(\mathbf{w}_t), \nabla L_S(\mathbf{w}_t) \rangle \leq \sqrt{2a \sum_{t=1}^{T} K_t^2 \log^{\frac{1}{2}}(1/\delta)}$$

$$\leq \sqrt{2(2^{2\theta+1} + 2)\Gamma(2\theta + 1) + \frac{2^{3\theta}\Gamma(3\theta+1)}{3}} KG \sqrt{\sum_{t=1}^{T} \eta_t^2 \log^{\frac{1}{2}}(1/\delta)}, \tag{79}$$

when $x \geq x_{\max}$, let $I(Tx) = (Tx/\sum_{i=1}^{T} K_i)^{\frac{1}{\theta}}$, $\forall \theta > 1$, then we have

$$-\sum_{t=1}^{T} \eta_t s_t^- \langle \mathbf{g}_t - \nabla L_S(\mathbf{w}_t), \nabla L_S(\mathbf{w}_t) \rangle \leq \frac{4^\theta}{T} \sum_{t=1}^{T} K_t \log^{\frac{1}{2}}(1/\delta)$$

$$\leq \frac{4^\theta KG}{T} \sum_{t=1}^{T} \eta_t \log^\theta(1/\delta). \tag{80}$$

To continue the proof, employing Lemma A.5 in term $-\eta_t \langle \frac{c\mathbf{g}_t}{\|\mathbf{g}_t\|_2} s_t^+, \nabla L_S(\mathbf{w}_t) \rangle$ and covering all $T$ iterations, we have

$$-\sum_{t=1}^{T} \eta_t \langle \frac{c\mathbf{g}_t}{\|\mathbf{g}_t\|_2} s_t^+, \nabla L_S(\mathbf{w}_t) \rangle \leq -\frac{c\sum_{t=1}^{T} \eta_t s_t^+ \|\nabla L_S(\mathbf{w}_t)\|_2}{3} + \frac{8c\sum_{t=1}^{T} \eta_t \|\mathbf{g}_t - \nabla L_S(\mathbf{w}_t)\|_2}{3}$$

$$\leq -\frac{c\sum_{t=1}^{T} \eta_t (1 - s_t^-) \|\nabla L_S(\mathbf{w}_t)\|_2}{3}$$

$$+ \frac{16\sum_{t=1}^{T} \eta_t \|\mathbf{g}_t - \nabla L_S(\mathbf{w}_t)\|_2 \|\nabla L_S(\mathbf{w}_t)\|_2}{3}. \tag{81}$$

With the truncated corollaries above, we have

1. If $0 \leq x \leq x_{\max}$, with probability at least $1 - 3\delta$

$$-\sum_{t=1}^{T} \eta_t \langle \frac{c\mathbf{g}_t}{\|\mathbf{g}_t\|_2} s_t^+, \nabla L_S(\mathbf{w}_t) \rangle \leq -\frac{c\sum_{t=1}^{T} \eta_t (1 - s_t^-) \|\nabla L_S(\mathbf{w}_t)\|_2}{3}$$

$$+ \frac{16\sum_{t=1}^{T} \eta_t \|\nabla L_S(\mathbf{w}_t)\|_2}{3} \begin{cases} 2K \log^{\frac{1}{2}}(2/\delta), & \text{if } \theta = \frac{1}{2}, \\ \sqrt{2}e(4\theta)^\theta K \log^{\frac{1}{2}}(2/\delta), & \text{if } \theta \in (\frac{1}{2}, 1], \\ \sqrt{2(2^{2\theta+1} + 2)\Gamma(2\theta + 1) + \frac{2^{3\theta}\Gamma(3\theta+1)}{3}} K \log^{\frac{1}{2}}(2/\delta) & \text{if } \theta > 1. \end{cases} \tag{82}$$

2. If $x \geq x_{\max}$ and $\theta \geq \frac{1}{2}$, with probability at least $1 - 3\delta$

$$-\sum_{t=1}^{T} \eta_t \langle \frac{c\mathbf{g}_t}{\|\mathbf{g}_t\|_2} s_t^+, \nabla L_S(\mathbf{w}_t) \rangle \leq -\frac{c\sum_{t=1}^{T} \eta_t (1 - s_t^-) \|\nabla L_S(\mathbf{w}_t)\|_2}{3}$$

$$+ \frac{16\sum_{t=1}^{T} \eta_t \|\nabla L_S(\mathbf{w}_t)\|_2}{3} 4^\theta K \log^\theta(2/\delta). \tag{83}$$

Then, according to Lemma A.1, combining the truncated results of $-\sum_{t=1}^{T} \eta_t \langle \mathbf{g}_t s_t^-, \nabla L_S(\mathbf{w}_t) \rangle$ and $-\sum_{t=1}^{T} \eta_t \langle \frac{c\mathbf{g}_t}{\|\mathbf{g}_t\|_2} s_t^+, \nabla L_S(\mathbf{w}_t) \rangle$, we have the inequality:

1. If $0 \leq x \leq x_{\max}$, with probability at least $1 - 3\delta - T\delta$

$$-\sum_{t=1}^{T} \eta_t \langle \overline{\mathbf{g}}_t, \nabla L_S(\mathbf{w}_t) \rangle \leq -\frac{c \sum_{t=1}^{T} \eta_t \|\nabla L_S(\mathbf{w}_t)\|_2}{3}$$

$$+ \begin{cases} 2KG\sqrt{\sum_{t=1}^{T} \eta_t^2} \log^{\frac{1}{2}}(1/\delta), & \text{if } \theta = \frac{1}{2}, \\[2mm] \sqrt{2}(4\theta)^\theta eKG\sqrt{\sum_{t=1}^{T} \eta_t^2} \log^{\frac{1}{2}}(1/\delta), & \text{if } \theta \in (\frac{1}{2}, 1], \\[2mm] \sqrt{2(2^{2\theta+1}+2)\Gamma(2\theta+1) + \frac{2^{3\theta}\Gamma(3\theta+1)}{3}} KG\sqrt{\sum_{t=1}^{T} \eta_t^2} \log^{\frac{1}{2}}(1/\delta) & \text{if } \theta > 1. \end{cases}$$

$$+ \frac{16 \sum_{t=1}^{T} \eta_t \|\nabla L_S(\mathbf{w}_t)\|_2}{3} \begin{cases} 2K \log^{\frac{1}{2}}(2/\delta), & \text{if } \theta = \frac{1}{2}, \\[2mm] \sqrt{2}e(4\theta)^\theta K \log^{\frac{1}{2}}(2/\delta), & \text{if } \theta \in (\frac{1}{2}, 1], \\[2mm] \sqrt{2(2^{2\theta+1}+2)\Gamma(2\theta+1) + \frac{2^{3\theta}\Gamma(3\theta+1)}{3}} K \log^{\frac{1}{2}}(2/\delta) & \text{if } \theta > 1. \end{cases}$$

$$\tag{84}$$

2. If $x \geq x_{\max}$ and $\theta \geq \frac{1}{2}$, with probability at least $1 - 3\delta - T\delta$

$$-\sum_{t=1}^{T} \eta_t \langle \overline{\mathbf{g}}_t, \nabla L_S(\mathbf{w}_t) \rangle \leq -\frac{c \sum_{t=1}^{T} \eta_t \|\nabla L_S(\mathbf{w}_t)\|_2}{3} + \frac{4^\theta KG}{T} \sum_{t=1}^{T} \eta_t \log^\theta(1/\delta)$$

$$+ \frac{16 \sum_{t=1}^{T} \eta_t \|\nabla L_S(\mathbf{w}_t)\|_2}{3} 4^\theta K \log^\theta(2/\delta). \tag{85}$$

Therefore, we refer to formula.(12) and formula.(13), and apply Lemma A.2 due to $\zeta_t \sim \mathbb{N}(0, c\sigma_{\mathrm{dp}}\mathbb{I}_d)$. Then, to simplify the notation, we define $\hat{\sigma}_{\mathrm{dp}}^2 = dc^2\sigma_{\mathrm{dp}}^2$. With $\hat{\sigma}_{\mathrm{dp}}^2 = m_2 \frac{Tc^2 dB^2 \log(1/\delta)}{n^2\epsilon^2}$ and probability $1 - 6\delta - T\delta$, if $0 \leq x \leq x_{\max}$, we have

$$(\frac{c}{3} - \frac{16}{3} aK \log^{\frac{1}{2}}(2/\delta) - 4\sqrt{e}\hat{\sigma}_{\mathrm{dp}} \log^{\frac{1}{2}}(1/\delta)) \sum_{t=1}^{T} \eta_t \|\nabla L_S(\mathbf{w}_t)\|_2 \leq L_S(\mathbf{w}_1) - L_S(\mathbf{w}_S)$$

$$+ (2\beta m_2 ed \frac{Tc^2 B^2 \log^2(2/\delta)}{n^2\epsilon^2} + 2\beta\sqrt{em_2 Td} \frac{c^2 B \log(2/\delta)}{n\epsilon} + \frac{1}{2}\beta c^2) \sum_{t=1}^{T} \eta_t^2$$

$$+ \sqrt{2a} KG\sqrt{\sum_{t=1}^{T} \eta_t^2} \log^{\frac{1}{2}}(1/\delta), \tag{86}$$

if $x \leq x_{\max}$, we have

$$(\frac{c}{3} - \frac{16}{3} aK \log^\theta(2/\delta) - 4\sqrt{e}\hat{\sigma}_{\mathrm{dp}} \log^{\frac{1}{2}}(1/\delta)) \sum_{t=1}^{T} \eta_t \|\nabla L_S(\mathbf{w}_t)\|_2 \leq L_S(\mathbf{w}_1) - L_S(\mathbf{w}_S)$$

$$+ (2\beta m_2 ed \frac{Tc^2 B^2 \log^2(2/\delta)}{n^2\epsilon^2} + 2\beta\sqrt{em_2 Td} \frac{c^2 B \log(2/\delta)}{n\epsilon} + \frac{1}{2}\beta c^2) \sum_{t=1}^{T} \eta_t^2$$

$$+ \sqrt{2a} KG\sqrt{\sum_{t=1}^{T} \eta_t^2} \log^\theta(1/\delta), \tag{87}$$

where $a = 2$ if $\theta = 1/2$, $a = (4\theta)^{2\theta} e^2$ if $\theta \in (1/2, 1]$ and $a = (2^{2\theta+1} + 2)\Gamma(2\theta+1) + \frac{2^{3\theta}\Gamma(3\theta+1)}{3}$ if $\theta > 1$.

Afterwards,

1. In case of light body, when $0 \leq x \leq x_{\max}$ and $\theta \geq \frac{1}{2}$:

   If $K \geq \hat{\sigma}_{\mathrm{dp}}$, let $\frac{c}{3} \geq \frac{33}{3}\sqrt{2a}K\log^{\frac{1}{2}}(2/\delta)$, $T = \mathbb{O}(\frac{n\epsilon}{\sqrt{d\log(1/\delta)}})$ and $\eta_t = \frac{1}{\sqrt{T}}$, we obtain

$$
\sum_{t=1}^{T}\eta_t\|\nabla L_S(\mathbf{w}_t)\|_2 \leq \frac{3}{\sqrt{2a}K\log^{\frac{1}{2}}(2/\delta)}(L_S(\mathbf{w}_1) - L_S(\mathbf{w}_S)) + \frac{3\sqrt{2a}KG\sqrt{\sum_{t=1}^{T}\eta_t^2}\log^{\frac{1}{2}}(1/\delta)}{\sqrt{2a}K\log^{\frac{1}{2}}(2/\delta)}
$$
$$
+ \frac{3\sum_{t=1}^{T}\eta_t^2}{\sqrt{2a}K\log^{\frac{1}{2}}(2/\delta)}\left(2\beta m_2 ed\frac{Tc^2 B^2\log^2(2/\delta)}{n^2\epsilon^2} + 2\beta\sqrt{em_2 Td}\frac{c^2 B\log(2/\delta)}{n\epsilon} + \frac{1}{2}\beta c^2\right)
$$
$$
\leq \frac{3(L_S(\mathbf{w}_1) - L_S(\mathbf{w}_S))}{\sqrt{2a}K\log^{\frac{1}{2}}(2/\delta)} + \frac{3\sqrt{2a}KG\log^{\frac{1}{2}}(1/\delta)}{\sqrt{2a}K\log^{\frac{1}{2}}(2/\delta)}
$$
$$
+ \frac{6\beta ea^2 K^2\log(2/\delta)}{\sqrt{2a}K\log^{\frac{1}{2}}(2/\delta)} + \frac{6\beta\sqrt{e}\sqrt{2a}K\log^{\frac{1}{2}}(2/\delta)}{\sqrt{2a}K\log^{\frac{1}{2}}(2/\delta)} + \frac{3\beta(33\sqrt{2a}K\log^{\frac{1}{2}}(2/\delta))^2}{2\sqrt{2a}K\log^{\frac{1}{2}}(2/\delta)}. \tag{88}
$$

   Therefore, with probability at least $1 - 6\delta - T\delta$, we have

$$
\frac{1}{T}\sum_{t=1}^{T}\|\nabla L_S(\mathbf{w}_t)\|_2 \leq \mathbb{O}(\frac{d^{\frac{1}{4}}\log^{\frac{3}{4}}(1/\delta)}{\sqrt{n\epsilon}}),
$$

   then, with probability $1 - \delta$, we have

$$
\frac{1}{T}\sum_{t=1}^{T}\|\nabla L_S(\mathbf{w}_t)\|_2 \leq \mathbb{O}(\frac{d^{\frac{3}{4}}\log^{\frac{3}{4}}(T/\delta)}{\sqrt{n\epsilon}}). \tag{89}
$$

   If $K \leq \hat{\sigma}_{\mathrm{dp}}$, let $\frac{c}{3} \geq 9\sqrt{e}\hat{\sigma}_{\mathrm{dp}}\log^{\frac{1}{2}}(1/\delta)$, that is, $c \geq 27\sqrt{e}\hat{\sigma}_{\mathrm{dp}}\log^{\frac{1}{2}}(1/\delta)$, thus there exists $T = \mathbb{O}(\frac{n\epsilon}{\sqrt{d\log(1/\delta)}})$, $T \geq 1$ and $\eta_t = \frac{1}{\sqrt{T}}$ that we obtain

$$
\sum_{t=1}^{T}\eta_t\|\nabla L_S(\mathbf{w}_t)\|_2 \leq \frac{1}{\sqrt{e}\hat{\sigma}_{\mathrm{dp}}\log^{\frac{1}{2}}(1/\delta)}(L_S(\mathbf{w}_1) - L_S(\mathbf{w}_S)) + \frac{\sqrt{2a}KG\sqrt{\sum_{t=1}^{T}\eta_t^2}\log^{\frac{1}{2}}(1/\delta)}{\sqrt{e}\hat{\sigma}_{\mathrm{dp}}\log^{\frac{1}{2}}(1/\delta)}
$$
$$
+ \frac{\sum_{t=1}^{T}\eta_t^2}{\sqrt{e}\hat{\sigma}_{\mathrm{dp}}\log^{\frac{1}{2}}(1/\delta)}\left(2\beta m_2 ed\frac{Tc^2 B^2\log^2(2/\delta)}{n^2\epsilon^2} + 2\beta\sqrt{em_2 Td}\frac{c^2 B\log(2/\delta)}{n\epsilon} + \frac{1}{2}\beta c^2\right)
$$
$$
\leq \frac{L_S(\mathbf{w}_1) - L_S(\mathbf{w}_S)}{\sqrt{e}\hat{\sigma}_{\mathrm{dp}}\log^{\frac{1}{2}}(2/\delta)} + \frac{\sqrt{2a}KG}{\sqrt{e}\hat{\sigma}_{\mathrm{dp}}} + 2\beta eK\log^{\frac{1}{2}}(2/\delta) + 2\beta\sqrt{e}\log^{\frac{1}{2}}(2/\delta) + \beta\frac{(27)^2}{2}K\log^{\frac{1}{2}}(2/\delta). \tag{90}
$$

   Therefore, with probability $1 - 6\delta - T\delta$, we have

$$
\frac{1}{T}\sum_{t=1}^{T}\|\nabla L_S(\mathbf{w}_t)\|_2 \leq \mathbb{O}(\frac{d^{\frac{1}{4}}\log^{\frac{3}{4}}(1/\delta)}{\sqrt{n\epsilon}}),
$$

   then, with probability $1 - \delta$, we have

$$
\frac{1}{T}\sum_{t=1}^{T}\|\nabla L_S(\mathbf{w}_t)\|_2 \leq \mathbb{O}(\frac{d^{\frac{1}{4}}\log^{\frac{3}{4}}(T/\delta)}{\sqrt{n\epsilon}}). \tag{91}
$$

2. In case of heavy tail, when $x \geq x_{\max}$:

If $\theta = \frac{1}{2}$ and $K \geq \hat{\sigma}_{\mathrm{dp}}$, let $\frac{c}{3} \geq \frac{33}{3}\sqrt{2a}K\log^{\frac{1}{2}}(2/\delta)$, $T = \mathbb{O}(\frac{n\epsilon}{\sqrt{d\log(1/\delta)}})$ and $\eta_t = \frac{1}{\sqrt{T}}$, we obtain

$$
\begin{aligned}
\sum_{t=1}^{T} \eta_t \|\nabla L_S(\mathbf{w}_t)\|_2 &\leq \frac{3}{\sqrt{2a}K\log^{\frac{1}{2}}(2/\delta)}(L_S(\mathbf{w}_1) - L_S(\mathbf{w}_S)) + \frac{3\sqrt{2a}KG\sqrt{\sum_{t=1}^{T}\eta_t^2}\log^{\frac{1}{2}}(1/\delta)}{\sqrt{2a}K\log^{\frac{1}{2}}(2/\delta)} \\
&\quad + \frac{3\sum_{t=1}^{T}\eta_t^2}{\sqrt{2a}K\log^{\frac{1}{2}}(2/\delta)}\left(2\beta m_2 ed\frac{Tc^2B^2\log^2(2/\delta)}{n^2\epsilon^2} + 2\beta\sqrt{em_2Td}\frac{c^2B\log(2/\delta)}{n\epsilon} + \frac{1}{2}\beta c^2\right) \\
&\leq \frac{3(L_S(\mathbf{w}_1) - L_S(\mathbf{w}_S))}{\sqrt{2a}K\log^{\frac{1}{2}}(2/\delta)} + \frac{3\sqrt{2a}KG\log^{\frac{1}{2}}(1/\delta)}{\sqrt{2a}K\log^{\frac{1}{2}}(2/\delta)} \\
&\quad + \frac{6\beta ea^2K^2\log(2/\delta)}{\sqrt{2a}K\log^{\frac{1}{2}}(2/\delta)} + \frac{6\beta\sqrt{e}\sqrt{2a}K\log^{\frac{1}{2}}(2/\delta)}{\sqrt{2a}K\log^{\frac{1}{2}}(2/\delta)} + \frac{3\beta(33\sqrt{2a}K\log^{\frac{1}{2}}(2/\delta))^2}{2\sqrt{2a}K\log^{\frac{1}{2}}(2/\delta)}. \tag{92}
\end{aligned}
$$

Therefore, with probability at least $1 - 6\delta - T\delta$, we have

$$
\frac{1}{T}\sum_{t=1}^{T}\|\nabla L_S(\mathbf{w}_t)\|_2 \leq \mathbb{O}\left(\frac{d^{\frac{1}{4}}\log^{\frac{3}{4}}(1/\delta)}{\sqrt{n\epsilon}}\right),
$$

then, with probability $1 - \delta$, we have

$$
\frac{1}{T}\sum_{t=1}^{T}\|\nabla L_S(\mathbf{w}_t)\|_2 \leq \mathbb{O}\left(\frac{d^{\frac{3}{4}}\log^{\frac{3}{4}}(T/\delta)}{\sqrt{n\epsilon}}\right). \tag{93}
$$

If $\theta = \frac{1}{2}$ and $K \leq \hat{\sigma}_{\mathrm{dp}}$, that is, $c \geq \frac{16aK\log^{\frac{1}{2}}(1/\delta)}{12}$, thus there exists $T = \mathbb{O}(\frac{n\epsilon}{\sqrt{d\log(1/\delta)}})$, $T \geq 1$ and $\eta_t = \frac{1}{\sqrt{T}}$ that we obtain

$$
\begin{aligned}
\sum_{t=1}^{T} \eta_t \|\nabla L_S(\mathbf{w}_t)\|_2 &\leq \frac{1}{\sqrt{e}\hat{\sigma}_{\mathrm{dp}}\log^{\frac{1}{2}}(1/\delta)}(L_S(\mathbf{w}_1) - L_S(\mathbf{w}_S)) + \frac{\sqrt{2a}KG\sqrt{\sum_{t=1}^{T}\eta_t^2}\log^{\frac{1}{2}}(1/\delta)}{\sqrt{e}\hat{\sigma}_{\mathrm{dp}}\log^{\frac{1}{2}}(1/\delta)} \\
&\quad + \frac{\sum_{t=1}^{T}\eta_t^2}{\sqrt{e}\hat{\sigma}_{\mathrm{dp}}\log^{\frac{1}{2}}(1/\delta)}\left(2\beta m_2 ed\frac{Tc^2B^2\log^2(2/\delta)}{n^2\epsilon^2} + 2\beta\sqrt{em_2Td}\frac{c^2B\log(2/\delta)}{n\epsilon} + \frac{1}{2}\beta c^2\right) \\
&\leq \frac{L_S(\mathbf{w}_1) - L_S(\mathbf{w}_S)}{\sqrt{e}\hat{\sigma}_{\mathrm{dp}}\log^{\frac{1}{2}}(2/\delta)} + \frac{\sqrt{2a}KG}{\sqrt{e}\hat{\sigma}_{\mathrm{dp}}} + 2\beta eK\log^{\frac{1}{2}}(2/\delta) + 2\beta\sqrt{e}\log^{\frac{1}{2}}(2/\delta) + \beta\frac{(27)^2}{2}K\log^{\frac{1}{2}}(2/\delta). \tag{94}
\end{aligned}
$$

Therefore, with probability $1 - 6\delta - T\delta$, we have

$$
\frac{1}{T}\sum_{t=1}^{T}\|\nabla L_S(\mathbf{w}_t)\|_2 \leq \mathbb{O}\left(\frac{d^{\frac{1}{4}}\log^{\frac{3}{4}}(1/\delta)}{\sqrt{n\epsilon}}\right),
$$

then, with probability $1 - \delta$, we have

$$
\frac{1}{T}\sum_{t=1}^{T}\|\nabla L_S(\mathbf{w}_t)\|_2 \leq \mathbb{O}\left(\frac{d^{\frac{1}{4}}\log^{\frac{3}{4}}(T/\delta)}{\sqrt{n\epsilon}}\right). \tag{95}
$$

If $\theta > \frac{1}{2}$, then term $\log^\theta(2/\delta)$ dominates the inequality. Let $\frac{c}{3} \geq \frac{17}{3}K\log^\theta(2/\delta)$, $T = \mathbb{O}(\frac{n\epsilon}{\sqrt{d\log(1/\delta)}})$ and $\eta_t = \frac{1}{\sqrt{T}}$, we obtain

$$
\sum_{t=1}^T \eta_t \|\nabla L_S(\mathbf{w}_t)\|_2 \leq \frac{3}{\sqrt{2a}K\log^\theta(2/\delta)}(L_S(\mathbf{w}_1) - L_S(\mathbf{w}_S)) + \frac{3\sqrt{2a}KG\sqrt{\sum_{t=1}^T \eta_t^2}\log^\theta(1/\delta)}{\sqrt{2a}K\log^\theta(2/\delta)}
$$

$$
+ \frac{3\sum_{t=1}^T \eta_t^2}{\sqrt{2a}K\log^\theta(2/\delta)}\left(2\beta m_2 ed\frac{Tc^2 B^2 \log^2(2/\delta)}{n^2\epsilon^2} + 2\beta\sqrt{em_2 Td}\frac{c^2 B\log(2/\delta)}{n\epsilon} + \frac{1}{2}\beta c^2\right)
$$

$$
\leq \frac{3(L_S(\mathbf{w}_1) - L_S(\mathbf{w}_S))}{\sqrt{2a}K\log^\theta(2/\delta)} + 3G + \frac{16^2}{24}\beta K\log^\theta(2/\delta) + 136\beta K\log^\theta(2/\delta) + 3\beta(17)^2 K\log^\theta(2/\delta).
$$
(96)

As a result, with probability $1 - \delta$, we have

$$
\frac{1}{T}\sum_{t=1}^T \|\nabla L_S(\mathbf{w}_t)\|_2 \leq \mathbb{O}(\frac{\log^\theta(T/\delta)d^{\frac{1}{4}}\log^{\frac{1}{4}}(T/\delta)}{\sqrt{n\epsilon}}).
$$
(97)

Consequently, integrate the above results on the condition that $\nabla L_S(\mathbf{w}_t) \geq c/2$.

For light body, we have

$$
\frac{1}{T}\sum_{t=1}^T \|\nabla L_S(\mathbf{w}_t)\|_2 \leq \mathbb{O}(\frac{d^{\frac{1}{4}}\log^{\frac{3}{4}}(T/\delta)}{\sqrt{n\epsilon}}),
$$
(98)

For heavy tail, we have

$$
\frac{1}{T}\sum_{t=1}^T \|\nabla L_S(\mathbf{w}_t)\|_2 \leq \mathbb{O}(\frac{d^{\frac{1}{4}}\log^{\theta+\frac{1}{4}}(T/\delta)}{\sqrt{n\epsilon}}),
$$
(99)

with probability $1 - \delta$ and $\theta \geq \frac{1}{2}$.

In a word, covering the two cases, we ultimately come to the conclusion with probability $1 - \delta$, $T = \mathbb{O}(\frac{n\epsilon}{\sqrt{d\log(1/\delta)}})$, $T \geq 1$ and $\eta_t = \frac{1}{\sqrt{T}}$:

**1. In the heavy tail region**:

$$
\frac{1}{T}\sum_{t=1}^T \min\left\{\|\nabla L_S(\mathbf{w}_t)\|_2, \|\nabla L_S(\mathbf{w}_t)\|_2^2\right\} \leq \mathbb{O}(\frac{d^{\frac{1}{4}}\log^{\theta+\frac{1}{4}}(T/\delta)}{(n\epsilon)^{\frac{1}{2}}}) + \mathbb{O}(\frac{d^{\frac{1}{4}}\log^{2\theta}(\sqrt{T})\log^{\frac{5}{4}}(T/\delta)}{(n\epsilon)^{\frac{1}{2}}})
$$

$$
\leq \mathbb{O}(\frac{d^{\frac{1}{4}}\log^{\frac{1}{4}}(T/\delta)\left(\log^\theta(T/\delta) + \log^{2\theta}(\sqrt{T})\log(T/\delta)\right)}{(n\epsilon)^{\frac{1}{2}}})
$$

$$
\leq \mathbb{O}(\frac{d^{\frac{1}{4}}\log^{\frac{5}{4}}(T/\delta)\hat{\log}(T/\delta)\log^{2\theta}(\sqrt{T})}{(n\epsilon)^{\frac{1}{2}}}),
$$
(100)

where $\hat{\log}(T/\delta) = \log^{\max(0,\theta-1)}(T/\delta)$. If $\theta = \frac{1}{2}$ and $K \leq \hat{\sigma}_{\mathrm{dp}}$, then $c_1 = \max\left(4^\theta 2K\log^\theta(\sqrt{T}), \frac{16aK\log^{\frac{1}{2}}(1/\delta)}{12}\right)$. If $\theta = \frac{1}{2}$ and $K \geq \hat{\sigma}_{\mathrm{dp}}$, then $c_1 = \max\left(4^\theta 2K\log^\theta(\sqrt{T}), 33\sqrt{2a}K\log^{\frac{1}{2}}(2/\delta)\right)$. If $\theta > \frac{1}{2}$, then $c_1 = \max\left(4^\theta 2K\log^\theta(\sqrt{T}), 17K\log^\theta(2/\delta)\right)$.

**2. In the light body region**:

$$
\frac{1}{T}\sum_{t=1}^T \min\left\{\|\nabla L_S(\mathbf{w}_t)\|_2, \|\nabla L_S(\mathbf{w}_t)\|_2^2\right\} \leq \mathbb{O}(\frac{d^{\frac{1}{4}}\log^{\frac{3}{4}}(T/\delta)}{(n\epsilon)^{\frac{1}{2}}}) + \mathbb{O}(\frac{d^{\frac{1}{4}}\log(\sqrt{T})\log^{\frac{5}{4}}(T/\delta)}{(n\epsilon)^{\frac{1}{2}}})
$$

$$
\leq \mathbb{O}(\frac{d^{\frac{1}{4}}\log^{\frac{1}{4}}(T/\delta)\left(\log^{\frac{1}{2}}(T/\delta) + \log(\sqrt{T})\log(T/\delta)\right)}{(n\epsilon)^{\frac{1}{2}}})
$$

$$
\leq \mathbb{O}(\frac{d^{\frac{1}{4}}\log^{\frac{5}{4}}(T/\delta)\log(\sqrt{T})}{(n\epsilon)^{\frac{1}{2}}}),
$$
(101)

where $\hat{\log}(T/\delta) = \log^{\max(0,\theta-1)}(T/\delta)$. If $\theta = \frac{1}{2}$ and $K \le \hat{\sigma}_{\mathrm{dp}}$, then $c_2 = \max\left(2\sqrt{2a}K\log^{\theta}(\sqrt{T}), \frac{16aK\log^{\frac{1}{2}}(1/\delta)}{12}\right)$. If $\theta = \frac{1}{2}$ and $K \ge \hat{\sigma}_{\mathrm{dp}}$, then $c_2 = \max\left(2\sqrt{2a}K\log^{\theta}(\sqrt{T}), 33\sqrt{2a}K\log^{\frac{1}{2}}(2/\delta)\right)$. If $\theta > \frac{1}{2}$, then $c_2 = \max\left(2\sqrt{2a}K\log^{\theta}(\sqrt{T}), 17K\log^{\theta}(2/\delta)\right)$.

$\square$

The proof of Theorem 5.3 is completed.

## F UNIFORM BOUND FOR DISCRIMINATIVE CLIPPING DPSGD

**Theorem F.1** (**Uniform Bound for Discriminative Clipping DPSGD**). *Under Assumptions 3.1, 3.2 and 3.3, combining Theorem 2 and Theorem 3, for any $\delta' \in (0,1)$, with probability $1 - \delta'$, we have*

$$\frac{1}{T} \sum_{t=1}^{T} \min \left\{ \|\nabla L_S(\mathbf{w}_t)\|_2, \|\nabla L_S(\mathbf{w}_t)\|_2^2 \right\} \leq p * \mathbb{O}\left(\frac{d^{\frac{1}{4}} \log^{\frac{5}{4}}(T/\delta) \hat{\log}(T/\delta) \log^{2\theta}(\sqrt{T})}{(n\epsilon)^{\frac{1}{2}}}\right)$$

$$+ (1-p) * \mathbb{O}\left(\frac{d^{\frac{1}{4}} \log^{\frac{5}{4}}(T/\delta) \log(\sqrt{T})}{(n\epsilon)^{\frac{1}{2}}}\right),$$

*where $\delta' = \delta'_m + \delta$, $\hat{\log}(T/\delta) = \log^{\max(0,\theta-1)}(T/\delta)$ and $p$ is ratio of heavy-tailed samples.*

*Proof.* We combine the subspace skewing error (Theorem 5.2) with the optimization bound of Discriminative Clipping DPSGD (Theorem 5.3) in this section to align with our algorithm outline. We have already discussed the error of traces in previous chapters and considered the condition of additional noise that satisfies DP, obtaining an upper bound on the error that depends on the factor $\mathbb{O}(\frac{1}{k})$. This conclusion means that, under the high probability guarantee of $1 - \delta'_m$, we can accurately identify the trace of the per-sample gradient with minimal error, and classify gradients into the light body and heavy tail based on the metric.

Specifically, based on statistical characteristics, approximately 5% -10% of the data will fall into the tail part. Thus, we select the top $p\%$ samples in the trace ranking as the tailed samples, where $p \in [5\%, 10\%]$. Although a subsampling strategy is used, uniform sampling does not change the proportion of tail samples in the batch. Furthermore, based on the relationship between trace and variance, the $pB$-th of sorted trace $\lambda_t^{\mathrm{tr},p}$ can be seen as the inflection point $x_{\max}$ of distribution defined in truncated theories A.7 and A.8, which corresponds to the empirical sample results with theoretical population variance and the approximation error has bounded in Theorem 5.2. Therefore, in discriminative clipping DPSGD, we can accurately partition the sample into the heavy-tailed convergence bound with a high probability of $(1 - \delta'_m) * p$, and exactly induce the sample to the bound of light bodies with a high probability of $(1 - \delta'_m) * (1 - p)$, while there is a discrimination error with probability $\delta'_m$. Accordingly, we have

$$\mathcal{C}_{\mathrm{u}}(c_1, c_2) := \frac{1}{T} \sum_{t=1}^{T} \min \left\{ \|\nabla L_S(\mathbf{w}_t)\|_2, \|\nabla L_S(\mathbf{w}_t)\|_2^2 \right\}$$

$$= (1 - \delta'_m) * p * \mathcal{C}_{\mathrm{tail}}(c_1) + (1 - \delta'_m) * (1 - p) * \mathcal{C}_{\mathrm{body}}(c_2) + \delta'_m * |\mathcal{C}_{\mathrm{tail}}(c_1) - \mathcal{C}_{\mathrm{body}}(c_2)|. \tag{102}$$

where $\mathcal{C}_{\mathrm{tail}}(c_1)$ means the convergence bound of $\frac{1}{T} \sum_{t=1}^{T} \min \left\{ \|\nabla L_S(\mathbf{w}_t)\|_2, \|\nabla L_S(\mathbf{w}_t)\|_2^2 \right\}$ when $\lambda_t^{\mathrm{tr},i} \geq \lambda_t^{\mathrm{tr},p}$, i.e. $\mathbb{O}\left(\frac{d^{\frac{1}{4}} \log^{\frac{5}{4}}(T/\delta) \hat{\log}(1/\delta) \log^{2\theta}(\sqrt{T})}{(n\epsilon)^{\frac{1}{2}}}\right)$, $\mathcal{C}_{\mathrm{body}}(c_2)$ denotes the bound of $\frac{1}{T} \sum_{t=1}^{T} \min \left\{ \|\nabla L_S(\mathbf{w}_t)\|_2, \|\nabla L_S(\mathbf{w}_t)\|_2^2 \right\}$ when $0 \leq \lambda_t^{\mathrm{tr},i} \leq \lambda_t^{\mathrm{tr},p}$ i.e. $\mathbb{O}\left(\frac{d^{\frac{1}{4}} \log^{\frac{5}{4}}(T/\delta) \log(\sqrt{T})}{(n\epsilon)^{\frac{1}{2}}}\right)$, with $c_1 = 4^\theta 2K \log^\theta(\sqrt{T})$ and $c_2 = 2\sqrt{2a}K \log^{\frac{1}{2}}(\sqrt{T})$.

If $\theta = \frac{1}{2}$, then $\mathcal{C}_{\mathrm{tail}}(c_1) = \mathcal{C}_{\mathrm{body}}(c_2)$ and $\delta'_m \to 0$, thus we have

$$\mathcal{C}_{\mathrm{u}}(c_1, c_2) = \mathcal{C}_{\mathrm{tail}}(c_1) = \mathbb{O}\left(\frac{d^{\frac{1}{4}} \log^{\frac{5}{4}}(T/\delta) \log(\sqrt{T})}{(n\epsilon)^{\frac{1}{2}}}\right). \tag{103}$$

If $\theta > \frac{1}{2}$, then $\mathcal{C}_{\mathrm{tail}}(c_1) \geq \mathcal{C}_{\mathrm{body}}(c_2)$, and we need to proof that $\mathcal{C}_{\mathrm{tail}}(c_1) \geq \mathcal{C}_u(c_1, c_2)$, i.e.

$$\mathcal{C}_{\mathrm{tail}}(c_1) \geq \mathcal{C}_{\mathrm{u}}(c_1, c_2)$$

$$\geq (1 - \delta'_m) * p * \mathcal{C}_{\mathrm{tail}}(c_1) + (1 - \delta'_m) * (1 - p) * \mathcal{C}_{\mathrm{body}}(c_2) + \delta'_m * |\mathcal{C}_{\mathrm{tail}}(c_1) - \mathcal{C}_{\mathrm{body}}(c_2)|.$$

By transposition, we have

$$(1 - \delta'_m)(1 - p) * \mathcal{C}_{\mathrm{tail}}(c_1) + \delta'_m * \mathcal{C}_{\mathrm{body}}(c_2) \geq (1 - \delta'_m) * (1 - p) * \mathcal{C}_{\mathrm{body}}(c_2).$$

Then, we have

$$C_{\text{tail}}(c_1) \geq C_{\text{body}}(c_2) - \frac{\delta'_m}{(1 - \delta'_m) * (1 - p)} C_{\text{body}}(c_2), \tag{104}$$

due to $\frac{\delta'_m}{(1-\delta'_m)*(1-p)} \geq 0$, it is proved that $C_{\text{tail}}(c_1) \geq C_u(c_1, c_2)$.

From another perspective, for $C_{\text{u}}(c_1, c_2)$, with probability $1 - \delta'_m$, we have

$$C_{\text{u}}(c_1, c_2) = p * C_{\text{tail}}(c_1) + *(1 - p) * C_{\text{body}}(c_2). \tag{105}$$

In other words, for the formula.(102), we define $\delta' = \delta'_m + \delta$. Then, with probability $1 - \delta'$, we have

$$\frac{1}{T} \sum_{t=1}^{T} \min \left\{ \|\nabla L_S(\mathbf{w}_t)\|_2, \|\nabla L_S(\mathbf{w}_t)\|_2^2 \right\} \leq p * \mathbb{O}\left( \frac{d^{\frac{1}{4}} \log^{\frac{5}{4}}(T/\delta) \hat{\log}(T/\delta) \log^{2\theta}(\sqrt{T})}{(n\epsilon)^{\frac{1}{2}}} \right)$$

$$+ (1 - p) * \mathbb{O}\left( \frac{d^{\frac{1}{4}} \log^{\frac{5}{4}}(T/\delta) \log(\sqrt{T})}{(n\epsilon)^{\frac{1}{2}}} \right) \tag{106}$$

where $\hat{\log}(T/\delta) = \log^{\max(0, \theta - 1)}(T/\delta)$.

$\square$

The proof of Theorem 5.4 is completed.

# G SUPPLEMENTAL EXPERIMENTS

## G.1 IMPLEMENTATION DETAILS AND CODEBASE

All experiments are conducted on a server with an Intel(R) Xeon(R) E5-2640 v4 CPU at 2.40GHz and a NVIDIA Tesla P40 GPU running on Ubuntu. By default, we uniformly set subspace dimension $k = 200$, $\epsilon = \epsilon_{tr} + \epsilon_{dp}$ with $\epsilon_{tr} = \epsilon_{dp}$, $p = 10\%$ and sub-Weibull index $\theta = 2$ for any datasets. In particular, we use the LDAM Cao et al. (2019) loss function for heavy-tailed tasks.

1. **MNIST**: MNIST has ten categories, 60,000 training samples and 10.000 testing samples. We construct a two-layer CNN network and replace the BatchNorm of the convolutional layer with GroupNorm. We set 40 epochs, 128 batchsize, 0.1 small clipping threshold, 1 large clipping threshold, and 1 learning rate.

2. **FMNIST**: FMNIST has ten categories, 60,000 training samples and 10.000 testing samples. we use the same two-layer CNN architecture, and the other hyperparameters are the same as MNIST.

3. **CIFAR10**: CIFAR10 has 50,000 training samples and 10,000 testing. We set 50 epoch, 256 batchsize, 0.1 small clipping threshold and 1 large clipping threshold with model SimCLRv2 Tramer & Boneh (2021) pre-trained by unlabeled ImageNet. We refer the code for pre-trained SimCLRv2 to `https://github.com/ftramer/Handcrafted-DP`.

4. **CIFAR10-HT**: CIFAR10-HT contains 32×32 pixel 12,406 training data and 10,000 testing data, and the proportion of 10 classes in training data is as follows: [0:5000, 1:2997, 2:1796, 3:1077, 4:645, 5:387, 6:232, 7:139, 8:83, 9:50]. We train CIFAR10-HT on model ResNeXt-29 Xie et al. (2017) pre-trained by CIFAR100 with the same parameters as CIFAR10. We can see pre-trained ResNeXt in `https://github.com/ftramer/Handcrafted-DP` and CIFAR10-HT with LDAM-DRW loss function in `https://github.com/kaidic/LDAM-DRW`.

5. **ImageNette**: ImageNette is a 10-subclass set of ImageNet and contains 9469 training examples and 3925 testing examples. We train on model ResNet-9 He et al. (2016) without pre-train and set 1000 batchsize, 0.15 small clipping threshold, 1.5 large clipping threshold and 0.0001 learning rate with 50 runs.

6. **ImageNette-HT**: We construct the heavy-tailed version of ImageNette by the method in Cao et al. (2019). ImageNette-HT contains 2345 trainging data and 3925 testing data, which is difficult to train, and proportion of 10 classes in training data follows: [0:946, 1:567, 2:340, 3:204, 4:122, 5:73, 6:43, 7:26, 8:15, 9:9]. The other settings are the same as ImageNette. Our ResNet-9 refers to `https://github.com/cbenitez81/Resnet9/` with 2.5M network parameters.

7. **E2E**: We have conducted experiments on transform-based NLP tasks for the dataset E2E with BLEU metric and GPT-2 model, which generates natural language from tabular data in the catering industry. We adopt the DPAdam optimizer and use the same settings as **?**, where small clipping threshold $c_2 = 0.1$ and large clipping threshold $c_1 = 10 * c_2$.

Moreover, we open our source code and implementation details for discriminative clipping on the following link: `https://anonymous.4open.science/r/DC-DPSGD-N-25C9/`.

### G.2 EFFECTS OF PARAMETERS ON TEST ACCURACY

Due to space limitations, we place the remaining ablation study on MNIST, FMNIST, ImageNette and ImageNette-HT in Table 5 and Table 6. We acknowledge that since ImageNette-HT has only 2,345 training data, which is one-fifth of ImageNette, it is difficult to support the convergence of the model. In the future, we will improve this aspect in our work.

Table 5: Effects of parameters on test accuracy with MNIST and FMNIST.

| Dataset | Subspace-$k$ | | | | $\epsilon_{tr} + \epsilon_{dp}$ | | | sub-Weibull-$\theta$ | | |
|---|---|---|---|---|---|---|---|---|---|---|
| | None | 100 | 150 | 200 | 2+6 | 4+4 | 6+2 | 1/2 | 1 | 2 |
| MNIST | 98.16 | 98.48 | 98.66 | 98.72 | 98.78 | 98.72 | 98.42 | 98.61 | 98.69 | 98.72 |
| FMNIST | 85.78 | 87.61 | 87.71 | 87.80 | 87.70 | 87.80 | 87.26 | 87.40 | 87.55 | 87.80 |

Table 6: Effects of parameters on test accuracy with ImageNette and ImageNette-HT.

| Dataset | Subspace-$k$ | | | | $\epsilon_{tr} + \epsilon_{dp}$ | | | sub-Weibull-$\theta$ | | |
|---|---|---|---|---|---|---|---|---|---|---|
| | None | 100 | 150 | 200 | 2+6 | 4+4 | 6+2 | 1/2 | 1 | 2 |
| ImageNette | 66.08 | 68.34 | 69.00 | 69.29 | 68.54 | 69.29 | 68.12 | 67.91 | 68.87 | 69.29 |
| ImageNette-HT | 29.33 | 31.44 | 33.17 | 33.70 | 34.25 | 33.70 | 31.13 | 33.05 | 33.37 | 33.70 |

To investigate the effect of $p$, we have added a set of new experiments by varying $p \in [1\%, 20\%]$. The results are presented in Table 7. We observe that the test accuracy is minimally affected when $p$ is less than 10%, but shows a negative impact at around 20%. We believe that the proportion of heavy-tailed samples aligns with statistical expectations. Assigning larger clipping thresholds to more light-body samples introduces more noise, while conservatively estimating heavy-tails does not fully exploit the algorithm's potential.

Table 7: Effects of parameter on $p$.

| Dataset | Heavy tail ratio-$p$ | | | | |
|---|---|---|---|---|---|
| | 20% | 10% | 5% | 2% | 1% |
| ImageNette | 66.82 | 69.29 | 68.44 | 68.45 | 68.75 |

