}_{\text{tail}}(c_1) + (1 - \delta'_m) * (1 - p) * \mathcal{C}_{\text{body}}(c_2) + \delta'_m * |\mathcal{C}_{\text{tail}}(c_1) - \mathcal{C}_{\text{body}}(c_2)|. \tag{102}$$

where $\mathcal{C}_{\text{tail}}(c_1)$ means the convergence bound of $\frac{1}{T} \sum_{t=1}^{T} \min\left\{\|\nabla L_S(\mathbf{w}_t)\|_2, \|\nabla L_S(\mathbf{w}_t)\|_2^2\right\}$ when $\lambda_t^{\text{tr},i} \geq \lambda_t^{\text{tr},p}$, i.e. $\mathbb{O}\left(\frac{d^{\frac{1}{4}} \log^{\frac{5}{4}}(T/\delta) \hat{\log}(1/\delta) \log^{2\theta}(\sqrt{T})}{(n\epsilon)^{\frac{1}{2}}}\right)$, $\mathcal{C}_{\text{body}}(c_2)$ denotes the bound of $\frac{1}{T} \sum_{t=1}^{T} \min\left\{\|\nabla L_S(\mathbf{w}_t)\|_2, \|\nabla L_S(\mathbf{w}_t)\|_2^2\right\}$ when $0 \leq \lambda_t^{\text{tr},i} \leq \lambda_t^{\text{tr},p}$ i.e. $\mathbb{O}\left(\frac{d^{\frac{1}{4}} \log^{\frac{5}{4}}(T/\delta) \log(\sqrt{T})}{(n\epsilon)^{\frac{1}{2}}}\right)$, with $c_1 = 4^\theta 2K \log^\theta(\sqrt{T})$ and $c_2 = 2\sqrt{2a}K \log^{\frac{1}{2}}(\sqrt{T})$.

If $\theta = \frac{1}{2}$, then $\mathcal{C}_{\text{tail}}(c_1) = \mathcal{C}_{\text{body}}(c_2)$ and $\delta'_m \to 0$, thus we have

$$\mathcal{C}_{\text{u}}(c_1, c_2) = \mathcal{C}_{\text{tail}}(c_1) = \mathbb{O}\left(\frac{d^{\frac{1}{4}} \log^{\frac{5}{4}}(T/\delta) \log(\sqrt{T})}{(n\epsilon)^{\frac{1}{2}}}\right). \tag{103}$$

If $\theta > \frac{1}{2}$, then $\mathcal{C}_{\text{tail}}(c_1) \geq \mathcal{C}_{\text{body}}(c_2)$, and we need to proof that $\mathcal{C}_{\text{tail}}(c_1) \geq \mathcal{C}_u(c_1, c_2)$, i.e.

$$\mathcal{C}_{\text{tail}}(c_1) \geq \mathcal{C}_{\text{u}}(c_1, c_2)$$

$$\geq (1 - \delta'_m) * p * \mathcal{C}_{\text{tail}}(c_1) + (1 - \delta'_m) * (1 - p) * \mathcal{C}_{\text{body}}(c_2) + \delta'_m * |\mathcal{C}_{\text{tail}}(c_1) - \mathcal{C}_{\text{

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

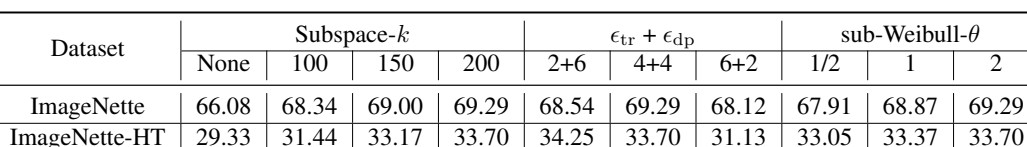

| Dataset | Heavy tail ratio-$p$ | | | | |
|---|---|---|---|---|---|
| | 20% | 10% | 5% | 2% | 1% |
| ImageNette | 66.82 | 69.29 | 68.44 | 68.45 | 68.75 |