# OpenReview forum: "Clip Body and Tail Separately: High Probability Guarantees for DP-SGD with Heavy Tails"
_ICLR.cc/2025/Conference — ICLR 2025 Conference Withdrawn Submission_

### Official Review · Reviewer_ScBt · 2024-10-26

**Soundness:** 1
**Presentation:** 1
**Contribution:** 2
**Rating:** 3
**Confidence:** 4

**Summary:**

This paper proposes an algorithmic method to improve the performances of differentially private training algorithm.

This method involves projecting the per-sample gradients on two sub-spaces, dubbed as body and tail gradients, and then applying a discriminative clipping mechanism to them, to enhance the performances and facilitate the optimization.

The paper proves upper bounds on the empirical risk as a function of the training iterations.

**Strengths:**

This paper studies an important problem, as it looks at improving the performances of differentially private (DP) optimization. The angle is also interesting, as the authors aim to improve the clipping mechanism differentiating it in two subspaces where the phenomenology should be qualitatively different. Considering different sub-spaces of the parameter space has already been considered to improve DP optimization, but, to the best of my knowledge, not with the perspective of the heavy tails of the distribution of the gradients.

**Weaknesses:**

The presentation, in my opinion, is everything but clear, from the very beginning of the paper. The authors quickly refer to the "gradient noise" without defining it. Considering that this work aims to the community of DP, I believe the average reader would think the authors refer to the injection of noise typical of DP-SGD, which is Gaussian and therefore not heavy tailed, generating the first confusion in this regard. Later it gets progressively clear that the authors refer to the randomness induced by the sub-sampling, and therefore the statistical discrepancy between the estimated (mini-batch) gradient with respect to the true empirical loss gradient. I invite the authors in clearly stating this from the very beginning, and to remark what randomness is addressed (and not) in this case.

The choice of references is also of dubious help. In line 46 Wang et. al (2021) does not seem to suggest using larger clipping threshold; while Gorbunov et al. (2020) looks at the clipping of the mini-batch average gradient, which is different from the per-sample clipping used to provide DP guarantees. The first one is (to the best of my knowledge) instead used for optimization purposes and to facilitate convergence, and does not provide the privacy guarantees of the per-sample clipping. This, again, does not help the reader. I invite the authors to review their choice of references (I did not loop over all of them), and in case some specific statement is taken from a paper, to also point the section in the citation.

The motivation of this work is also not clear. Multiple empirical works [1, 2, 3] (see, e.g., the second paragraph at page 6 of [2]) show that arbitrarily small clipping constant can provide optimal utility in private training. This paper aims to provide a control on the clipping constants assuming that the per-sample gradients might be heavy-tailed in the probability space of the data, but why is this a necessity in light of the experimental evidence from [1, 2, 3]? Is the setting different? Would their consideration fail in some setting that this paper aims to tackle? If yes, is there any experimental evidence of this?

In the main contributions the authors show an informal upper bound on the empirical risk, which reads $\log(\sqrt T)$ (which is $\Theta(\log T)$) - see line 103. How is this bound meaningful? If the labels are of constant order, trivial guess would provide an upperbound of $O(1)$ both on the empirical and population loss (while also guaranteeing privacy), which is better than the one given by the authors. What is the effective importance of this bound, and why is it informative? Can the authors elaborate on this?

The baselines are not completely fair. I would recommend using [2] as a better updated version for baselines on DP-SGD. Besides, the results obtaind by the authors look relatively strong, but no code is available and provided in the supplementary material. The algorithm is not as straightforward as DP-SGD and no auditing has been implemented to verify the effective guarantee of the algorithm. I remark that this is stated as a weakness given my opinion of the method not being carefully explained. Nevertheless, I do not consider this for my score, which is mainly due to the points I raised above.


- - - -

[1] https://arxiv.org/abs/2201.12328

[2] https://arxiv.org/abs/2204.13650

[3] https://arxiv.org/abs/2110.05679

**Questions:**

What do the authors mean by:

"the tuning parameters in the classical Abadi's clipping function are complex" in line 137?

"denote $k$-dimensional random projection sampled from heavy-tailed distributions." in line 156? Which distributions? In particular, I find the definition of $V$ rather obscure, and consequently the corresponding step in Algorithm 1. What does "extract orthogonal vectors" in the 6th line of the Algorithm mean? Can the authors be more precise?

"their normalized versions act as mutually orthogonal eigenvectors" in line 305? Independence in high dimension indeed generates approximate orthogonality, but this requires precise assumptions. As a counter-example, if the per-sample gradients where deterministic (i.e. concentrated to their non-0 expectation), Assumption 3.1 would still hold, while the gradients would all be parallel.

---

> ### Author Response · Authors · 2024-11-17
>
> Thank you for your comments.
>
> $\textbf{Presentation}$: We will prioritize the definition of `gradient noise' and more clearly emphasize that the core of our problem focuses on the heavy-tail assumption under sampling randomness.
>
> $\textbf{Reference}$: We list these references to demonstrate that the clipping thresholds in SGD tend to be larger compared to those in DPSGD and to highlight the gap in the application of gradient clipping between SGD and DPSGD. We will revisit the correctness of our references.
>
> $\textbf{Motivation}$: Our motivation is to divide gradient clipping into two parts under the heavy-tailed assumption. We validated the effectiveness of our approach using the experimental setup in [2], demonstrating that relying solely on a sufficiently small clipping threshold does not achieve optimal performance. In fact, if DPSGD operates without gradient clipping, its convergence can be significantly improved. However, small clipping thresholds inevitably introduce considerable clipping bias. Our proposed method seeks to strike a more precise balance between clipping bias and utility of DPSGD, particularly in heavy-tailed scenarios.
>
> $\textbf{Contribution}$: Our contribution is not limited to the logarithmic terms. We first propose a tight high-probability bound for DPSGD, where the logarithmic terms are close to the current SOTA high-probability bounds for SGD. We highlight that optimizing the logarithmic terms in high-probability bounds is not easy. Additionally, our DC-DPSGD (Theorem 5.4) improves the original DPSGD bound (Theorem 4.1), with an exponential improvement in both the logarithmic terms and the broken probability $\delta$, which represents a significant contribution in high-probability bounds.
>
> $\textbf{Baseline and Code}$: More experimental details and code links can be found in Appendix G.1. We will also include additional experiments related to the baseline papers you mentioned.

---

> > ### Comment · Reviewer_ScBt · 2024-11-21
> >
> > I thank the authors for their rebuttal. I believe the paper would highly benefit from a clearer presentation, a better discussion on the relevance of the theoretical bounds (see my original point in the review, not directly addressed in the rebuttal), and a more open presentation of the experimental results, i.e. with public implementation and eventually a privacy auditing scheme to indeed verify also numerically that this method is sound.
> >
> > I confirm my score.

---

### Official Review · Reviewer_wp42 · 2024-10-31

**Soundness:** 3
**Presentation:** 3
**Contribution:** 3
**Rating:** 8
**Confidence:** 2

**Summary:**

This work proposes a discriminative clipping (DC) method for training non-convex smooth models with DP-SGD when gradients are in a class of heavy-tailed sub-Weibull distributions. The method, DC-DPSGD, first employs a subspace identification technique to categorize the per-sample gradient of each batch into a light-body or heavy tail, and then uses a smaller clipping threshold for light-body gradients and a larger threshold for heavy-tailed gradients. The method is found to achieve a better balance between clipping loss and required DP noise, which results in performance improvement at least poly(logT) i.t.o. high-probability excess empirical risks. The work is well-motivated, and the idea of using different clipping thresholds is interesting. Moreover, numerical experiments suggest the method is promising, and insights into the impacts of hyperparameters are interesting.

**Strengths:**

1. interesting idea of discriminative clipping, supported with solid theoretical analysis
2. models and assumptions are sufficiently general and cover a lot of existing settings as special cases.
3. extensive numerical experiments demonstrating the effectiveness of the proposed method; impacts of hyperparameters on performance might be of practical interest.

**Weaknesses:**

It comes to my attention that the subspace identification technique (step 9 of alg 1) seems not a true classifier that classifies each gradient into a light body or a heavy-tailed. Instead, it simply selects a portion of gradients whose linear transformation $V_{t, k}^\top \widehat{\mathbf{g}}_t(z_i)$ are ranked top-p% by squared l2 norm (because $tr(xx^\top)=|x|_2^2$). It would be good if the authors could reply to my this concern.

**Questions:**

1. line 237, $c$ is not defined. I presume $c$ is the clipping threshold. However, because this is the first time $c$ appears and clipping threshold plays a key role, it would be good to clearly define it before using it.
2. regarding the weakness mentioned earlier, can we just privately select top-p% gradients having larger l2 norm, without projecting into a subspace? Any issue with this simple idea?
3. line 357, you may need a subscript $i$ for the notation $\hat{\lambda}^{tr}_{t}$?
4. Theorem 5.4, $C_u$ seems different from the performance measure in Theorem 5.3. And $\nabla \hat{L}_s$ is not defined.
5. Theorem 5.4 appears to be a bit confusing for me. As $p$ is a hyperparameter between 0 and 1, if we take $p\rightarrow 0$, will the result degenerate to that for a case with only light-tailed gradients? In other words, it seems we can fully control the performance by choosing the value of $p$?
6. typos around line 472, 2360.

---

> ### Author Response · Authors · 2024-11-17
>
> Thank you for your appreciation of our work and detailed comments. We will response your concerns.
>
> Weaknesses 1 and Question 2:
> We employ subspace identification to differentiate between light-body and heavy-tail gradients. This involves performing private sorting based on the deviation of normalized per-sample gradients (capturing directional information) from a heavy-tailed distribution, which allows us to characterize the samples. When the underlying distribution of $\widehat{\mathbf{g}}$ aligns more closely with that of $V_{t,k}$ (which can be viewed as eigenvectors), their trace (the sum of inner products) becomes larger. The intuition is analogous to how principal components in PCA maximize the eigenvalue~(inner product) of the original vector. We recognize this as a heuristic method, and we provide a theoretical upper bound on the error of the approach in Section 5.1. We acknowledge that there is potential for improvement in developing more precise discriminative methods and would welcome further discussion on this topic.
>
> Regarding Question 2, in heavy-tailed scenarios, the $l_2$ norm of per-sample gradients may be unbounded. Performing private sorting in such cases would require infinite sensitivity, which makes noise addition infeasible and impractical. In contrast, our subspace identification method, which relies on normalized eigen sorting, effectively mitigates this issue and ensures feasibility under differential privacy constraints.
>
> Question 1\& Question 4: We thank the reviewer for pointing out these issues. In the revised version, we will define these variables and provide clearer and more consistent explanations.
>
> Question3: We will add the subscript $i$ to better illustrate the discrimination for per-sample gradients.
>
> Question5: We appreciate the reviewer’s insightful question. When $p$ approaches 0, it represents a special case under the light-tail assumption, which aligns with the gradient noise assumption used by mainstream DPSGD approaches and their corresponding theoretical results for high-probability bounds. However, under the heavy-tail assumption, $p$ actually depends on the true heavy-tailed magnitude of the data distribution, which generally lies around 10\%. In other words, the improvement depends on the actual heavy-tailed nature of the data distribution.
>
> Question6: We thank the reviewer for the careful observation, and we will correct this in the revised version.

---

> > ### Comment · Reviewer_wp42 · 2024-11-18
> >
> > I want to thank the authors for their response. It helps me understand that the algorithm is recognized as a heuristic method.
> >
> > After reading other reviewer's comments and the authors' responses, I realize I might misunderstand some part of the algorithm. Therefore, I would like to lower my confidence of my assessment.

---

### Official Review · Reviewer_2ETm · 2024-11-03

**Soundness:** 1
**Presentation:** 1
**Contribution:** 1
**Rating:** 1
**Confidence:** 5

**Summary:**

The paper proposes a novel approach, Discriminative Clipping (DC)-DPSGD,
with two key designs. First, it introduces a subspace identification technique to
distinguish between body and tail gradients. Second, it presents a discriminative
clipping mechanism that applies different clipping thresholds for body and tail
gradients separately to reduce the clipping loss. Under the non-convex condition
and heavy-tailed sub-Weibull gradient noise assumption, DC-DPSGD reduces
the empirical risk.

**Strengths:**

1. Proposing DC-DPSGD with a subspace identification technique and a discriminative clipping
mechanism to optimize DPSGD under sub-Weibull gradient noise assumption. To our knowledge,
this is the first work to rigorously address heavy tails in DPSGD with high probability guarantees.

2 Presenting a high probability guarantee with best-known rates for the optimization performance
of DPSGD, and improve it to faster rates by DC-DPSGD

3. DC-DPSGD consistently outperforms three baselines with up to 9.72% accuracy improvements, demonstrating the effectiveness
of our proposed approach.

**Weaknesses:**

1. The theoretical contribution is very limited: Although they mentioned there is some improvement. However, such improvement is just logarithmic terms. Note that for non-convex ERM, the best-known result for the gradient norm is O(\frac{d^1/2}{(n\epsilon)^2/3). Thus, I do not know why the authors only mention a comparison with DP-SGD. If you use DP-SPIDER, you can already achieve $O(\frac{d^1/2}{(n\epsilon)^2/3)$. Thus, the theoretical results are very limited.

2. The algorithm itself is also unsurprising. It uses the projection on subspaces. However, such an idea has been widely studied in the DP deep learning literature. But there is no comparison between them.

3. The experiments are not convincing. For example, Auto-S/NSGD is mainly designed for better clipping. Its performance actually is comparable to the original DP-SGD.  I would like to see whether the method can achieve SOTA for CIFAR 10 for private pertaining rather than fine-tuning. DP-PSAC is also for another clipping and it is not SOTA. If the author provides a new clipping method, then comparing with these two methods is sufficient. It does not. So it is very strange to compare with these methods.

**Questions:**

This outer loop in algorithm 1 is the mistake made by the author? The input does not specify E and subsequent algorithm does not use c. from high level understanding, seems like only the inner loop for T is required?

When g_t^{tail} shows up, I do not know what it means. The presentation for algorithm is confusing. Which makes it hard to check the soundness of the privacy guarantee.

The privacy proof for trace sorting is too complicated; could the author simply do it?

---

> ### Author Response · Authors · 2024-11-13
>
> We thank the reviewer for the detailed comments. We will further explain our contributions in theory and compare our methods with existing subspace-based DPSGD work.
>
> $\textbf{Weaknesses}$.
>
> Point 1:
> The mentioned work based on DP-SPIDER, such as [1-2]. Their bounds are indeed $\mathcal{O}({\frac{d^{\frac{2}{3}}}{n\epsilon}})$ and the better $\mathcal{O}({\frac{d^{\frac{2}{3}}}{(n\epsilon)^{\frac{2}{3}}}\textasciicircum\frac{d^{\frac{1}{6}}}{n\epsilon}})$, but the condition for their bounds to be better than our stated SOTA result $\mathcal{O}({\frac{d^{\frac{1}{4}}}{(n\epsilon)^{\frac{1}{2}}}})$ is $n\epsilon>d$, which is difficult to hold when the model dimension is large. In our work, we focus more on the general case, where the SOTA DPSGD bound holds without the condition $n\epsilon>d$, where $\mathcal{O}({\frac{d^{\frac{1}{4}}}{(n\epsilon)^{\frac{1}{2}}}})$ is the current best result. Furthermore, our contribution is not limited to the logarithmic terms; we are the first to propose a high-probability bound for DPSGD, where the logarithmic terms are close to the current SOTA high-probability bounds for SGD, which is a very tight bound. Optimizing the logarithmic terms in high-probability bounds is not an easy task. Additionally, our DC-DPSGD (Theorem 5.4) improves the original DPSGD bound (Theorem 4.1), with an exponential improvement in both the logarithmic terms and the broken probability $\delta$, which represents a significant contribution in high-probability bounds.
>
> [1] Private (stochastic) nonconvex optimization revisited: Second-order stationary points and excess risks.
>
> [2] How to Make the Gradients Small Privately: Improved Rates for Differentially Private Non-Convex Optimization.
>
> Point 2: We clarify the distinction from subspace-based DPSGD. They use public data to generate a low-rank subspace, thereby reducing the DP noise to low-rank dependencies and avoiding the severe dependence on model parameters' $d$-dimensionality. Our approach, on the other hand, uses subspace classification to differentiate light-tail and heavy-tail per-sample gradients and improve the gradient clipping in DPSGD instead of the dimension dependence of DP noise, without introducing additional public data. Both theoretically and practically, our approach is orthogonal to subspace-based DPSGD. We will further validate through experiments that our algorithm outperforms subspace-based DPSGD.
>
> Point 3: We will provide additional experiments regarding subspace-based DPSGD.
>
> $\textbf{Questions}$.
>
> Q1: Our analysis is strictly based on $T$ iterations, but it can be easily extended to $E$ epochs, or simply by substituting $T=T*E$.
>
> Q2: The symbol $\mathbf{g}^{tail}$ means the heavy-tailed gradient with the larger (top-$p\%$) sorted trace, and we employ the large clipping threshold $c_1$ to clip them, instead of using the single common clipping threshold $c_2$ for the normal light body gradient.
>
> Q3: The DP sorting we designed is similar to the functionality of DP top-k and DP SVT, that is, sorting a set of values $V=[v1,v2,...,v_{n}]$
>   after adding noise. Since their divergence or sensitivity depends on how the sorting result changes when one data point is modified, the proof process needs to consider all such scenarios. Our key aim is to prove that \textbf{TraceSorting$\circ$GradientPerturbation(GP)} satisfys DP. We will appropriately simplify the proof and provide the core parts for your reference.

---

> > ### Author Response · Authors · 2024-11-13
> >
> > Supplement to Question 3:
> >
> > We define the clipping threshold vector $c$ for per-sample gradient by TraceSorting, for example, with the batch gradients with batch size $B=3$ and the heavy-tailed proportion $p=1/3$, if heavy tailed indicator $\lambda=[1,0,0] \text{then}~c=[c_1,c_2,c_2]$, which means the first sample is a heavy-tailed gradient $\mathbf{g}^{tail}$ and would be clipped with the larger clipping threshold $c_1$.
> > $$ \mathbb{P}[M(D)=Y]=\mathbb{P}[\text{TraceSorting=index} i \text{AND GP}|D] $$
> >
> > $$ =  \int_{-\infty}^{\infty}\mathbb{P}[i|D,r_{-i}]\cdot\mathbb{P}[\text{GP with heavy tailed samples} i]dr  $$
> >
> > $$ = \int_{-\infty} ^{\infty} \int_{-\infty}^{\infty}\mathbb{P}[i|D,r_{-i}] \cdot\mathbb{P}[\frac{1}{B}(\sum^{B\in D}_jg_j+c_j\zeta_j)=Y|c]drd\zeta$$
> >
> > $$ = \int_{-\infty}^{\infty}\int_{-\infty}^{\infty}\mathbb{P}[i|D,r_{-i}]\cdot\mathbb{P}[f(D)=Y|c]\cdot\mathbb{P}[\zeta=c_j\zeta_j/B]drd\zeta=*$$
> >
> > where $r\sim\text{Gauss}(1/\epsilon_{\mathrm{tr}})$ and $\zeta\sim\text{Gauss}(1/\epsilon_{\mathrm{dp}})$. We define $f(\cdot)=$~GradientDiscent and $\Delta f=\Vert f(D)-f(D')\Vert_2=\frac{1}{B}(pBc_1+(1-p)Bc_2)=pc_1+(1-p)c_2$.
> > With $1-(\delta_{\mathrm{tr}}+\delta_{\mathrm{dp}})$, we have
> >
> > $$ *= \int_{-\infty}^{\infty}\int_{-\infty}^{\infty}\exp(\epsilon_{\mathrm{tr}})\mathbb{P}[i|D',r_{-i}]\cdot\mathbb{P}[\frac{1}{B}(\sum^{B\in D'}_jg_j+c_j\zeta_j)=Y|c]drd\zeta $$
> >
> > $$= \int_{-\infty}^{\infty}\int_{-\infty}^{\infty}\exp(\epsilon_{\mathrm{tr}})\mathbb{P}[i|D',r_{-i}]\cdot\mathbb{P}[f(D')+c_j\zeta_j/B=Y+\Delta f|c]drd\zeta $$
> >
> > $$= \int_{-\infty}^{\infty}\int_{-\infty}^{\infty}\exp(\epsilon_{\mathrm{tr}})\mathbb{P}[i|D',r_{-i}]\cdot\mathbb{I}[f(D')=Y]\cdot\mathbb{P}[\zeta=c_j\zeta_j/B-\Delta f|c]drd\zeta $$
> >
> > $$\leq\int_{-\infty}^{\infty}\int_{-\infty}^{\infty}\exp(\epsilon_{\mathrm{tr}})\mathbb{P}[i|D',r_{-i}]\cdot\mathbb{I}[f(D')=Y]\cdot\exp(\epsilon_{\mathrm{dp}})\mathbb{P}[\zeta=c_j\zeta_j/B|c]drd\zeta $$
> >
> > $$\leq\exp(\epsilon_{\mathrm{tr}}+\epsilon_{\mathrm{dp}})\mathbb{P}[M(D')=Y].  $$
> >
> > Then, $\textbf{TraceSorting$\circ$GradientPerturbation(GP)}$ satisfying ($\epsilon,\delta$)-DP is hold.
> >
> > We sincerely hope the reviewer would appreciate our contributions and consider raising the score if satisfied.

---

> > ### Comment · Reviewer_2ETm · 2024-12-02
> >
> > I am unsatisfactory with the author's response. There is no previous theoretical paper that says d^1/4/(n\epsilon)^1/2  is the best result in theory! If \sqrt{d}>n\epsilon, then your bound will be constant and meaningless. Please show your results is optimal.

---

> > > ### Author Response · Authors · 2024-12-03
> > >
> > > Reasonable response to the reviewer：
> > >
> > > For clipped DPSGD, the bound of $\mathcal{O}(\frac{d^{1/4}}{(n\epsilon)^{1/2}})$ represents the best-known result [ref1] in the context of DPSGD combined with gradient clipping, without requiring any additional assumptions beyond smoothness and Lipschitz continuity (e.g., no assumptions like gradient symmetry used in [ref2-3]). Moreover, the known bounds for clipped SGD further validate that our bound for clipped DPSGD is both tight and optimal. Specifically, with the setting of $T=\frac{d^{1/2}}{n\epsilon}$, we find that the high-probability bound of $\Vert \nabla L_S(\mathbf(w)_t) \Vert^2 \leq \mathcal{O}(\frac{d^{1/4}}{(n\epsilon)^{1/2}})$ can be convert into $\Vert \nabla L_S(\mathbf(w)_t) \Vert \leq \mathcal{O}(\frac{1}{T^{1/4}})$, which coincides with the non-convex expected upper bound provided for clipped SGD by Zhang et al. (Theorem 2) [ref4], under the assumption of smoothness and bounded variance, and matches the corresponding lower bound (Theorem 6).
> > >
> > > Additionally, we emphasize that DP-SPIDER and clipped DPSGD fundamentally differ in nature. Unlike clipped DPSGD, DP-SPIDER does not involve explicit gradient clipping operation, and therefore, the bounds derived from DP-SPIDER cannot be directly compared to the conclusions about clipped DPSGD. And actually, the upper bound of DP-SPIDER is $(\frac{\sqrt{d}}{n\epsilon})^{2/3}$ (Theorem 4.2) [ref5], but not $\frac{d^{1/2}}{(n\epsilon)^{2/3}}$ you mentioned, thus, if $\sqrt{d}\geq n\epsilon$, DP-SPIDER is also constant and meaningless, and many previous bounds for DPSGD are meaningless [ref6-8].
> > >
> > > [1] Normalized/Clipped SGD with Perturbation for Differentially Private Non-Convex Optimization
> > >
> > > [2] Understanding Gradient Clipping in Private SGD: A Geometric Perspective
> > >
> > > [3] Automatic Clipping: Differentially Private Deep Learning Made Easier and Stronger
> > >
> > > [4] Why are Adaptive Methods Good for Attention Models?
> > >
> > > [5] Faster Rates of Convergence to Stationary Points in Differentially Private Optimization
> > >
> > > [6] Efficient private erm for smooth objectives
> > >
> > > [7] Private stochastic convex optimization with optimal rates
> > >
> > > [8] Private stochastic convex optimization: optimal rates in linear time

---

### Official Review · Reviewer_GmbT · 2024-11-03

**Soundness:** 3
**Presentation:** 3
**Contribution:** 2
**Rating:** 6
**Confidence:** 3

**Summary:**

The paper proposes a new DP-SGD like training algorithm, the key idea is to clip gradient using different threshold for body and tail distributions. The proposed algorithm first identify if a sample belongs to body or tail using subspace identification, then clip and add noise correspondingly. Convergence analyses are conducted to show the algorithm is guaranteed to converge with comparable or improved rates with over algorithms. Experiments show the algorithm can have improved performance in practice.

**Strengths:**

1. The proposed algorithm is studied in both theory and practice. Yielding better results in both cases. The idea of clipping different samples using different threshold is interesting, with the actual algorithm also being solid with guarantees.
2. The idea of clipping samples with different thresholds worth further exploring by the community. Clipping with a single threshold is known to have issues for class-imbalanced data and may create fairness issues, using different clipping threshold could be a way to mitigate this though it is not touched in this work.

**Weaknesses:**

1. The proposed algorithm seems to have significantly higher computation cost compared with standard DP-SGD
2. The algorithm introduced more hyperparameters to tune in practice.

**Questions:**

1. In practice what is a good way to balance privacy budget for subspace identification and privacy budget for gradient noise?
2. Have the authors considered label-based clipping threshoulds?

---

> ### Author Response · Authors · 2024-11-13
>
> Thank you for appreciating our work. We will answer the weaknesses and insightful questions.
>
> $\textbf{Weaknesses}$.
>
> Point 1: We believe that the main additional computational complexity arises from the calculation of the projection subspace. Generally, this can be mitigated by using power PCA techniques to reduce computational overhead and improve the quality of projection reconstruction. In our experiments, we primarily used the SVD package provided by PyTorch for the computation. By adjusting the `full matrices' parameter, we can significantly alleviate the computation time, reducing it from the original time dependence on $\mathcal{O}(d^2)$ to $\mathcal{O}(d*B)$, where $d$ is model parameters and $B$ is the batch size.
>
> Point 2: We introduced a larger $c_1$
>   compared to the original $c_2$
>   as the heavy-tailed clipping threshold. We provide a linear perspective that links the tuning of $c_1$ with $c_2$, (i.e., $c_1=10c_2$), meaning that only $c_2$
>   needs to be adjusted, similar to the original DPSGD. For other hyper-parameters, such as $\theta$ and $p$, their impact on the algorithm is limited and relatively insensitive within a certain range, as discussed in Experiment 6.2 and Appendix G.2.
>
> $\textbf{Questions}$.
>
> Q1: We believe that, based on a limited privacy budget, equal distribution is often a one-size-fits-all solution in experiments. However, upon closer theoretical analysis, there may be slight advantages in using the privacy budget more strategically in DPSGD perturbation. This is because the dependence on noise of DPSGD is $d$-dimensional, while the dependence on subspace identification is at a constant level.
>
> Q2: We believe that label-based clipping thresholds could yield interesting results in the field of fairness. Discriminative clipping can assign more weight to sparse labels, helping to mitigate the issue of unfairness. We would be eager to explore this further.
>
>  We would like to thank the reviewer again for the comments.

---

### Official Review · Reviewer_agpJ · 2024-11-05

**Soundness:** 2
**Presentation:** 2
**Contribution:** 3
**Rating:** 3
**Confidence:** 4

**Summary:**

This paper proposes Discriminative Clipping (DC), a novel algorithm for gradient clipping when gradient norms follow a heavy tailed distribution. Current techniques assume the norms follow subgaussian distributions, so these methods can incur in high utility drop for heavier tailed distributions.

DC identifies the subspace containing the gradients to separate the tail, and after that performs clipping with different thresholds. For a subWeibull distribution, DC reduces empirical risk. Their experiments show improvements up to 10% over baselines.

**Strengths:**

The paper studies a relevant problem in the privacy and optimization community. Typically, DP-SGD degrades model performance, with respect to the non-private model. This paper proposes an interesting idea of setting two different clipping values. If the distribution of gradients is heavy tailed, then it should help performance by reducing the bias introduced by overclipping.

**Weaknesses:**

**Unclear relation to previous work:**
- I read Bu et al. 2024 and could not find “the small gradient assumption”, only the observation that in the range [0.1, 1000], the smaller clipping values worked better when training GPT2 and Imagenet.

**Unclear claims and terminology:**
- Throughout the paper symbols are mixed, undefined, and sentences are ambiguous, making it hard to validate the correctness of theorems and proofs. Specifically, this ambiguity could break the privacy guarantee (e.g. incorrect noise calibration due to undefined noise variables), a fundamental aspect of the paper. Similarly, the description of the algorithm could be made clearer. Below I provide some examples.

  - Comment for Line 35: clipping is performed to bound the maximum divergence rather than “obtaining”.


  - Comment for Line 36: Gradient clipping can introduce bias so the estimation is not unbiased. I am not sure if saying “gradient noise” is appropriate, but rather sampling.

  - Theorem 4.1 does not introduce what “c” is. Is it the clipping value? And the bound does not depend on it so it is unclear why it is introduced. Further, I think  $\delta$ from the privacy guarantee and $\delta$ from the high probability bound share the same symbol? How is the noise defined in this version of DP-SGD? The version in the appendix specifies it,  but still depends on two constants m_1, m_2 that are also not introduced in the theorem statement.

  - Similarly, theorem 5.1. uses constants that have not been defined, specifically q, T, B.

  - Lines 297-299 sketch the proof for the privacy guarantee but it is hard to follow, yet a crucial aspect of the paper:

    “According to the results of trace sorting,…” What results?
    “we apply two clipping thresholds for gradient perturbation, making it essential to reanalyze the unified privacy
    guarantees of our composition mechanism.” I found this sentence unclear.

  - Section 5.1. is hard to follow due to several unjustified statements.

  - Algorithm 1 and the corresponding adjacent could be more precise (see questions)

**Informal privacy claims**
- Throughout the manuscript the privacy parameters are mixed with high probability bound parameters disrupting the flow of the paper and clarity about statements. Further, it is hard to parse and validate that the privacy claims can be derived given the gaps.

**Experimental setting is not clearly explained**
- Details about the baselines are missing, making it difficult to assess why the proposed method is outperforming all other methods. For example, details on the hyperparameter tuning process for both DC and the baselines are needed for the results on table 1.

**Minor:**
- Some citations are missing parenthesis.
- I would recommend introducing the definition of heavy tailed index earlier in the introduction.
- Section 3.1 has several typos
- What do the authors mean by “private batch size” in the input of algorithm 1?

**Questions:**

1. How can one verify in practice if one or two thresholds should be used?

2. Section 5.1. Introduces subspace identification. The authors mention the similarity between gradients and a subspace. How is this similarity defined?

3. Can the authors clarify this sentence? “Due to the high-dimensional nature of gradients, their normalized versions act as mutually orthogonal eigenvectors”. Are the authors claiming that  all gradients are orthogonal? Or that body and tail gradients are orthogonal? What is the intuition behind this?

4. On Algorithm 1:

    - How are c1 and c2 tuned?

    - How is the batch generated in line 5 of the algorithm? Poisson sampling, without replacement, cyclically traversing the dataset? This has an impact on the privacy bound.

    - How is the sub-Weibull distribution specified in line 6 of algorithm 1? How are $g_t^tail$ and $g_t^body$ defined?

    - The authors state that gradients are “divided into the light body or heavy tail”. So the set of per-example gradients in a batch is split in two mutually exclusive groups? Or each gradient is split into two components? The algorithm suggests the latter but the description is unclear.

    - How are the clipping values for baselines defined? From the heatmaps it seems like large c_2 values always work best. So I would imagine that baselined also profited from a large (unique) clipping value.

---

> ### Author Response · Authors · 2024-11-13
>
> Thank you for your review. We will address the your comments point-by-point.
>
> $\textbf{Unclear relation to previous work}$. We argue that the work by Bu et al. focuses on small-norm gradients, as their optimization and analysis are based on the assumption of light-tailed gradient noise. Moreover, their specific clipping scheme, $\mathbf{g}=\frac{\mathbf{g}}{\Vert \mathbf{g}\Vert+\gamma}$ ($\mathbf{g}$ denotes per-sample gradient and $\gamma>0$ is a hyper-parameter), normalize every per-sample gradient to about 1 and sacrifices the information of large-norm gradients, while amplifying the energy of small-norm gradients. Our work aims to ensure that large-norm gradients are not excessively sacrificed during clipping.
>
> $\textbf{Unclear claims and terminology}$. We thank the reviewer for careful reading.
>
> Point 1-2: we will refine the Line 35 and 36 for clear claims.
>
> Point 3: the symbol $c$ in Theorem 4.1 denotes clipping threshold. Our Theorem 4.1 specifically addresses classic DPSGD proposed by Abadi et al. (which employs Gaussian noise). Thus, in our bounds, we also interpret the privacy parameter $\delta$ as a high-probability form, as it essentially arises from the unbounded nature of Gaussian noise, and we share the symbol $\delta$ in our approach.
>
> Point 4: we define $B$ as the batch size in Algorithm.1, $q=\frac{B}{n}$ as the sample ratio and $T$ as iterations in the context.
>
> Point 5: the results of trace sorting are classified into light body or heavy tail gradients~(as detailed in Section 5.1). Then, based on the sorted and well classified gradients, we apply the large clipping threshold to the heavy tailed gradient and the small clipping threshold to the light body one.  In addition, we perform the discriminative clipping at the level of per-sample gradient, but not the eigen-components of one gradient.
>
> Point 6-7: we will address them in the Questions.
>
> $\textbf{Informal privacy claims}$. We consider the symbol $\delta$ in the privacy parameter to be a type of high-probability bound parameter, as Gaussian-based DPSGD entails a certain failure rate. In the case of non-pure DP in Definition 3.1, the broken probability $1-\delta$ arises from the unbounded nature of Gaussian noise.
>
> $\textbf{Experimental setting is not clearly explained}$. Due to the space limitations, we place the specific experimental details in Supplementary Material G.1, where the parameter settings of the baselines follows follow those provided in the original paper, while our parameters are based on extensive experimental searches and provide corresponding empirical guidance.
>
> $\textbf{Minor}$. Thank you for the reviewer’s comments. We will address them one by one. Regarding the fourth point, we refer to $B$ as the batch size for the private dataset.

---

> > ### Author Response · Authors · 2024-11-13
> >
> > $\textbf{Questions}$. Thank you to the reviewer for raising these valuable questions and we will answer them thoroughly.
> >
> > Q1: In deep learning, the gradients of even the simplest datasets (MNIST, FMNIST) tend to exhibit heavy-tailed behaviors.
> > Therefore, it is common, general, and reasonable to use two clipping thresholds with more complex data environments in practice.
> >
> > Q2: Subspace similarity refers to the fact that heavy-tailed gradients become more active in heavy-tailed subspaces, and their trace increases in subspaces that share similar characteristics, that is, the linear transformation product $v^T_k\mathbf{g}$ of the subspace eigenvector $v_k$ and the gradient $\mathbf{g}$. The larger the value, the more similar the gradient is to the heavy-tailed subspace, making it more likely to be a heavy-tailed gradient.
> >
> > Q3: The idea of high-dimensional gradients being approximately orthogonal stems from the fact that, due to the vast data space and high-dimensional matrix analysis, high-dimensional random vectors tend to exhibit near-orthogonal properties. The intuition behind using subspaces is that high-dimensional per-sample gradients can be treated as potential eigen-vectors of the entire dataset's distribution. We then apply a heavy-tailed distribution and perform a linear transformation on these high-dimensional gradients to calculate their similarity, i.e., the trace. If the trace is small, it suggests that the gradients are likely orthogonal to the heavy-tailed subspace, meaning the gradient is probably far from the heavy-tailed subspace distribution. In contrast, if the trace is large, it indicates that the gradient "responds" to the heavy-tailed subspace, meaning it is likely not orthogonal to the heavy-tailed subspace, and thus is likely a heavy-tailed gradient.
> >
> > Q4-1: We emphasize that the choice of $c_2 $ is based on extensive experimental exploration and follows the settings used in existing DPSGD, with $ c_2 = [0.0001, 0.001, 0.01, 0.1, 1, 10] $ and learning rate $= [100, 10, 1, 0.1, 0.01, 0.001] $ For $c_1$, we first provide the guideline $c_1 = 10c_2 $in our theoretical analysis in Theorem 5.3 and in Section 6.3, and in conjunction with extensive experimental searches, we use $c_1 = [1, 10, 100, 100]c_2$. This setting achieves nearly optimal results in our experiments.
> >
> > Q4-2: For the data sampling, we adopt the uniform sampling without replacement as in most DPSGD works.
> >
> > Q4-(3-4): The sub-Weibull distribution is characterized mainly by the heavy-tailed parameter $\theta$. In emprical setting, we conduct the experiments for the choose of the parameter in Section 6.2. As shown in Figure 2, we divide batch gradients into two categories: tail and body, and define the per-sample gradient with the larger sorted trace (top-$p\%$ per-sample gradients in the batch) as $\mathbf{g}^{tail}$ and the smaller one as $\mathbf{g}^{body}$.
> >
> > Q4-5: For the heatmaps (Figure 5), we would like to emphasize that the theoretical guidance $c_1=10c_2$ is also supported by our experimental results, as the optimal value for our approach is achieved under this setting. The baseline results are based on the parameters provided in their original paper and further refined through extensive experimental exploration on our part. We are confident that, under the small/common clipping threshold we have proposed (with a single $c_2$), these baselines do not outperform the results obtained with our method.
> >
> > We hope we have addressed your comments and if not, we are happy to further discuss them.

---

> > > ### Comment · Reviewer_agpJ · 2024-12-02
> > > **Thanks!**
> > >
> > > Thanks for the response!
> > >
> > > > Q1: In deep learning, the gradients of even the simplest datasets (MNIST, FMNIST) tend to exhibit heavy-tailed behaviors. Therefore, it is common, general, and reasonable to use two clipping thresholds with more complex data environments in practice.
> > >
> > > In a private setting it is already hard to estimate one clipping threshold, how can one in practice verify these two thresholds efficiently?
> > >
> > > > We emphasize that the choice of $c_2$ is based on extensive experimental exploration.
> > >
> > > Wouldn't this violate privacy?
> > >
> > > > Q4-2: For the data sampling, we adopt the uniform sampling without replacement as in most DPSGD works.
> > >
> > > I think most papers use Poisson sampling for the analysis (e.g. Abadi et al) because it is easier to analyse, but practical implementations shuffle datasets and then traverse them cyclically. I think the authors are not supposed to solve this gap in the literature but they do have to clarify in their paper which version they're using and where. Especially if the experiments use a different smapling strategy since this will degrade the privacy guarantee.
> > >
> > > I thank the authors for taking the time to discuss my comments. I will keep my score.

---

> > ### Comment · Reviewer_agpJ · 2024-12-02
> >
> > > We argue that the work by Bu et al. focuses on small-norm gradients
> >
> > I think the authors are using interchangeably the terms "small norm gradients" with "concentrated norm distribution", and these are not the same.
> >
> > > We consider the symbol in the privacy parameter to be a type of high-probability bound parameter
> >
> > This interpretation corresponds to Probabilistic DP, not approximate DP. See, e.g., [https://arxiv.org/pdf/1906.01337.](https://arxiv.org/pdf/1906.01337.)
> >
> > > Due to the space limitations,
> >
> > I think a short description that allows interpretation of results is necessary. Additional details can be left to the appendix.

---

### Note · Authors · 2024-12-10

I have read and agree with the venue's withdrawal policy on behalf of myself and my co-authors.